# Regret Lower Bounds for Decentralized Multi-Agent Stochastic Shortest Path Problems

**Utkarsh U. Chavan**
Department of Mechanical Engineering
Indian Institute of Technology Bombay
Mumbai India 400076
utkarshuchavan007@gmail.com

**Prashant Trivedi**
Computer Science and Information Systems
BITS Pilani, Pilani Campus
Rajasthan 333031
iitb.pt@gmail.com

**Nandyala Hemachandra**
Industrial Engineering and Operations Research
Indian Institute of Technology Bombay
Mumbai India 400076
nh@iitb.ac.in

## Abstract

Multi-agent systems (MAS) are central to applications such as swarm robotics and traffic routing, where agents must coordinate in a decentralized manner to achieve a common objective. Stochastic Shortest Path (SSP) problems provide a natural framework for modeling decentralized control in such settings. While the problem of learning in SSP has been extensively studied in single-agent settings, the decentralized multi-agent variant remains largely unexplored. In this work, we take a step towards addressing that gap. We study decentralized multi-agent SSPs (Dec-MASSPs) under linear function approximation, where the transition dynamics and costs are represented using linear models. Applying novel symmetry-based arguments, we identify the structure of optimal policies. Our main contribution is the first regret lower bound for this setting based on the construction of hard-to-learn instances for any number of agents, $n$. Our regret lower bound of $\Omega(\sqrt{K})$, over $K$ episodes, highlights the inherent learning difficulty in Dec-MASSPs. These insights clarify the learning complexity of decentralized control and can further guide the design of efficient learning algorithms in multi-agent systems.

## 1 Introduction

The Stochastic Shortest Path (SSP) problem is a foundational model for goal-oriented decision-making under uncertainty, extending classical shortest path formulations by incorporating stochastic transitions and random costs [6, 4]. In an SSP, an agent sequentially selects actions in a probabilistic environment with the objective of reaching a designated terminal state while minimizing the expected cumulative cost. By explicitly modeling uncertainty in outcomes, SSP serves as a core abstraction in many areas, including robotics, automated planning and reinforcement learning. While planning in fully known SSPs is well understood [5], the focus has increasingly shifted to the learning setting, where the agent must learn the optimal policy through interaction with the environment. The quality of learning algorithms is typically measured using the notion of *regret*, which quantifies the difference between the cost incurred by following a learning algorithm and the optimal policy [11, 3, 2].

Given its broad applicability, the problem of learning optimal policies in tabular SSP environments has garnered significant attention, leading to algorithms with near-optimal regret guarantees [20, 18, 21, 7]. However, in many real-world applications, the state and action spaces are humongous, making tabular

39th Conference on Neural Information Processing Systems (NeurIPS 2025).

approaches computationally infeasible. To overcome this, recent works have assumed linear structure in the cost function or transition dynamics [25, 14, 24]. These approaches have led to algorithms that achieve sublinear regret in SSPs with either known or unknown cost settings, along with tight upper and lower bounds on regret [20, 18, 7, 14, 24], making the single-agent SSP well understood.

Despite significant progress in single agent SSPs, many real-world applications such as robotics, traffic routing, and distributed control naturally involve multiple agents interacting in a shared environment [9, 16, 1, 13]. These multi-agent systems are often decentralized, meaning agents cannot or choose not to share private information (e.g., their actions or incurred costs). However, they can exchange certain parameters (e.g., cost or transition parameter estimates) over a predefined communication graph. Motivated by this, [23] introduced the decentralized multi-agent stochastic shortest path (Dec-MASSP) problem with linear function approximations of transition dynamics and costs. They propose a learning algorithm and report a sub-linear regret upper bound of $\widetilde{O}(B^{*1.5}d\sqrt{nK/c_{\min}})$, where $n$ is the number of agents, $K$ is the number of episodes, $d$ is the individual feature dimension, $B^*$ is the maximum expected cost-to-go, and $c_{\min} > 0$ is a lower bound on per-period cost. While it provides the first regret guarantees for Dec-MASSPs, optimality of their algorithm remains open leading us to a central question:

*What are the fundamental limits of learning in Dec-MASSPs with linear function approximations?*

To address this question, we establish tight regret lower bounds for the decentralized MASSP setting by constructing hard-to-learn instances. Extending single-agent SSP lower bound techniques [14, 18] to decentralized MASSPs presents several key challenges, which we overcome using novel methods.

(i) **Exponential state-action space.** Even with just two nodes in the network, the global state space grows as $2^n$ with the number of agents $n$, which complicates both instance construction and lower bound analysis. To address this in the linear setting, we introduce a novel feature design for the transition probabilities specific to our MASSP instances. These features lead to a valid transition probability distribution and can serve as a foundational approach for feature design in other decentralized multi-agent reinforcement learning (MARL) applications.

(ii) **Coupled costs and dynamics.** In the MASSP setting, both transition probabilities and cost functions are inherently coupled across agents, depending on the global state-action space. To construct hard-to-learn instances and establish a meaningful regret lower bound, we first demonstrate that it suffices to restrict the analysis to uniform cost. We derive closed-form expressions for the transition probabilities using our novel feature design. This explicit formulation allows us to identify the structure of optimal actions, making our approach both tractable and novel.

(iii) **Intractable value functions.** Unlike the single-agent setting with two nodes [14], where a closed-form expression for the optimal value of the non-goal state can be exactly derived, obtaining similar expressions for the many non-goal states in multi-agent systems is significantly challenging due to exponentially many states. To address this, we observe that this huge state space can be partitioned based on the number of agents present at each node. This key insight allows us to establish a monotonicity property of the value function, which in turn is sufficient to derive regret lower bounds without requiring its explicit closed-form.

(iv) **Bounding KL divergence.** A standard approach for deriving regret lower bound relies on information-theoretic techniques, particularly bounding the Kullback–Leibler (KL) divergence between two distributions. In the SSP setting [14], the KL divergence reduces to just two analytically tractable terms. However, in the MASSP setting, the KL divergence involves an exponential number of terms, making exact analysis intractable. To overcome this, we exploit the non-negativity of the KL divergence and leverage the symmetry in our problem instances, which allows us to bound the KL divergence efficiently depending on the number of agents present at each node.

Building on the challenges outlined above and the novel methods we use to address them, we now summarize the major contributions of our work.

- **Regret lower bounds:** We establish the first $\Omega(\sqrt{K})$ regret lower bound for decentralized MASSPs under linear function approximation, which matches the previously reported upper bound up to constant and poly-logarithmic factors [23], thereby providing a precise characterization of the fundamental limits of learning in these settings. Notably, our result also recovers the lower bound of [14] as a special case of $n = 1$, which matches the corresponding upper bound in $K$.

- **Design of hard-to-learn MASSP instances:** The above regret bound is based on a proposed family of decentralized MASSP instances with linearly parameterized costs and transitions that are provably hard-to-learn. As scalability with $n$ is a key factor in evaluating the efficiency of MARL algorithms, our increasingly hard-to-learn instances (with respect to $n$) are suitable to assess the performances of such algorithms.

- **First analysis of instances with exponential state-action space**: Our lower bound analysis is the first to handle exponential state and action space in MASSP settings. These lower bounds are based on the identification of optimal policies and value functions in two-node MASSPs. These optimal policies have the same values for all states with identical number of agents at each node.

## 1.1 Related work

**Stochastic shortest path (SSP) problems and learning in tabular setting.** The stochastic shortest path (SSP) problem is a classical framework introduced in early work by [8]. [6] later extended deterministic shortest path theory to stochastic environments, formalizing the SSP setting within dynamic programming. More recently, the focus has shifted toward learning in SSPs under unknown dynamics, in the tabular settings. Authors in [20] introduced Upper Confidence SSP (UC-SSP), the first no-regret algorithm for tabular SSPs, with regret $\widetilde{O}(DS\sqrt{ADK/c_{\min}})$, where $D$ is the expected hitting time and $c_{\min}$ the minimum per-period cost, $S$ and $A$ denote the cardinality of state and action space and $K$ is the number of episodes. Follow-up work by [18] improved this bound by removing the $c_{\min}$ dependence and gave a lower bound of $\Omega(B^*\sqrt{SAK})$, where $B^*$ captures the maximum optimal cost-to-go, showing near-optimality. [21] further refined SSP learning through Exploration Bonus for SSP (EB-SSP), a model-based algorithm with regret $\widetilde{O}(B^*\sqrt{SAK})$, that doesn't require prior knowledge of $B^*$. [7] introduced a minimax-optimal algorithm under stochastic costs with matching bounds by reducing the SSP to a finite-horizon MDP.

**Function approximation in SSPs.** To address scalability beyond the tabular regime, recent efforts have explored SSPs under linear function approximation. [24] designed a model-free algorithm with sub-linear regret and is computationally efficient under minimal assumptions, using stationary policies. Parallelly, [14] proposed two algorithms, each based on Hoeffding-type and Bernstien-type sets. The Bernstien-type confidence set based algorithm has a better regret upper bound of $\widetilde{O}(dB^*\sqrt{K/c_{\min}})$. Additionally, the paper provides a lower bound $\Omega(dB^*\sqrt{K})$ in this setting.

**Decentralized multi-agent SSP.** The extension of SSP learning to decentralized multi-agent systems is still in its early stages. Authors in [23] introduced the MASSP framework with linearly parameterized costs and transitions, and proposed MACCM- a decentralized learning algorithm, reporting a regret upper bound of $\widetilde{O}(B^{*1.5}d\sqrt{nK/c_{\min}})$, where $n$ is the number of agents. However, their work does not establish any lower bounds, leaving open the fundamental questions regarding inherent difficulty and optimal regret rates for decentralized MASSP learning. We close this gap by establishing the first regret lower bounds for decentralized MASSPs (with any number of agents, $n$) under linear transition kernel.

Table 1 summarizes the relevant results.

# 2 Preliminaries and problem formulation

In this section, we introduce the Multi-Agent Stochastic Shortest Path (MASSP) problem, and also present the formal setup and notations that underpin the decentralized learning framework.

## 2.1 Notations and preliminaries

A MASSP problem is defined by the tuple $(\mathcal{V}, \mathcal{N}, \mathbb{P}, \mathcal{A})$, where $\mathcal{V} = \{v_1, v_2, \ldots, v_q\}$ is the set of nodes and $\mathcal{N} = \{1, 2, \ldots, n\}$ is the set of agents. Each agent occupies exactly one node at any given time, leading to a global system state $\boldsymbol{s} = (s_1, s_2, \ldots, s_n)$ in the global state space $\mathcal{S} = \mathcal{V}^n$; so $|\mathcal{S}| = q^n$. We assume that all agents start at the same node $s$ and aim to reach the common goal node $g$, resulting in initial and goal states $\boldsymbol{s}_{\text{init}} = (s, \ldots, s)$ and $\boldsymbol{g} = (g, \ldots, g)$, respectively. Each agent $i \in \mathcal{N}$ has a finite action space $\mathcal{A}_i$ at every node, leading to a joint action $\boldsymbol{a} = (a_1, \ldots, a_n) \in \mathcal{A} = \mathcal{A}_1 \times \cdots \times \mathcal{A}_n$. Once an action $\boldsymbol{a}$ is taken in the state $\boldsymbol{s}$, the system evolves

| Reference | Setting | Upper Bound | Lower Bound |
|---|---|---|---|
| [20] | • | $\widetilde{O}(DS\sqrt{ADK/c_{\min}})$ | – |
| [18] | • | $\widetilde{O}(B^*S\sqrt{AK})$ | $\Omega(B^*\sqrt{SAK})$ |
| [7] when $B^* > 1$ | • | $\widetilde{O}(B^*\sqrt{SAK})$ | – |
| [7] when $B^* < 1$ | • | $\widetilde{O}(\sqrt{B^*SAK})$ | $\Omega(\sqrt{B^*SAK})$ |
| [14] | ⋆ | $\widetilde{O}(dB^*\sqrt{K/c_{\min}})$ | $\Omega(dB^*\sqrt{K})$ |
| [24] | ⋆ | $\widetilde{O}(dB^*\sqrt{dB^*K/c_{\min}} + B^{*2}d^3/c_{\min})$ | – |
| [23] | ‡ | $\widetilde{O}(B^{*1.5}d\sqrt{nK/c_{\min}})$ | – |
| **This work** | ‡ | – | $\Omega\left(\dfrac{d\sqrt{KB^*/n}}{2^n}\right)$ |

Table 1: Comparison of related work in SSP learning and decentralized multi-agent SSPs. Here $S$ and $A$ denote the number of states and actions, respectively; $K$ is the number of episodes; $c_{\min} > 0$ is a lower bound on the minimum per-period cost to an agent; $B^*$ is the SSP diameter, defined as the maximum (over all states) of the expected cost to reach goal under the optimal policy; $D$ is the expected hitting time defined, as the maximum over all states, of the minimum (over policies of) expected time to reach goal state starting from this state. Symbols: • = tabular SSP (no function approximation), ⋆ = linear mixture SSP (with linear function approximation), ‡ = decentralized MA-SSP (with linear function approximation).

to next state $s'$ according to a Markovian transition kernel $\mathbb{P}(\cdot \mid s, a)$ and each agent $i \in \mathcal{N}$ incurs an instantaneous cost $c_i(s, a) \in [c_{\min}, 1]$, where $c_{\min} > 0$.

The cost function $c_i(s, a)$ captures complex dependencies of individual costs on both the positions and actions of all agents. For example, the cost can be modeled by taking $c_i(s, a) = x_i(s, a) \cdot K_i(s, a)$ where $x_i(s, a) \coloneqq \sum_{j=1}^n \mathbb{1}_{\{a_j = a_i\}}$ is the *congestion* (number of agents choosing the same action) and $K_i(s, a)$ encodes the cost due to private efficiency and all other complex dependencies (except congestion). The global cost is defined as the average of individual costs: $c(s, a) = \frac{1}{n} \sum_{i=1}^n c_i(s, a)$.

A stationary Markov deterministic policy is a mapping $\pi : \mathcal{S} \to \mathcal{A}$. Given the stationary Markovian dynamics $\mathbb{P}$ and cost $c$, an optimal policy lies in the class $\Pi_{\mathrm{MD}}$ of Markov deterministic policies [17]. We further focus on the class of proper policies $\Pi_p \subseteq \Pi_{\mathrm{MD}}$, i.e., the policies for which each agent reaches the goal node, $g$ from any initial state with probability one [20, 21, 7, 14]. Given any proper policy $\pi \in \Pi_p$, the value function (or cost-to-go function) is defined as $V^\pi(s) = \mathbb{E}\left[\sum_{t=1}^{\tau^\pi(s)} c(s^t, \pi(s^t)) \,\middle|\, s^1 = s\right]$, where the expectation is over all sample paths due to interaction of policy $\pi$ and transition kernel $\mathbb{P}$. $\tau^\pi(s)$ is the random time to reach the goal state $g$ under policy $\pi$ starting at $s$. The objective is to find a policy that minimizes the expected cumulative global cost from the initial state $s_{\mathrm{init}}$ to the goal state $g$. The optimal value function with the corresponding optimal policy $\pi^*$ is thus

$$V^*(s) = \min_{\pi \in \Pi_p} V^\pi(s). \tag{1}$$

We further define the diameter of MASSP as $B^* = \max_{s \in \mathcal{S}} V^*(s)$.

## 2.2 Problem formulation

When transition kernel $\mathbb{P}$ and cost $c(\cdot, \cdot)$ are known, the objective of MASSP problem is to find an optimal policy in the space of proper policies, as in Equation (1). However, in the learning setting with unknown transition kernel $\mathbb{P}$ or cost $c(\cdot, \cdot)$, agents must actively *explore* to learn an optimal policy. Though the goal is to minimize the cumulative global cost to reach $g$, agents in our setup act independently and lack knowledge of others' actions or costs. A natural solution is to employ a central controller that aggregates signals from all agents to coordinate learning [26]. However, such centralization poses several challenges: (i) poor scalability with the number of agents due to increased communication overhead [12] (ii) vulnerability to single-point failures and adversarial attacks [27], and (iii) privacy concerns, as agents may be unwilling to reveal their individual costs. These limitations motivate decentralized learning, where the agents do not explicitly share the actions

or private costs among themselves via a central controller. This setting falls within the purview of Decentralized Multi-Agent Reinforcement Learning (Dec-MARL).

To make decentralized learning tractable, prior works (e.g., [27, 22]) adopt linear cost approximations as $c(\boldsymbol{s}, \boldsymbol{a}) = \langle \psi(\boldsymbol{s}, \boldsymbol{a}), \boldsymbol{w} \rangle$, where $\psi(\boldsymbol{s}, \boldsymbol{a}) \in \mathbb{R}^{nd}$ is a known feature map, and $\boldsymbol{w} \in \mathbb{R}^{nd}$ is an unknown parameter, with $nd << |\mathcal{S}||\mathcal{A}|$. This allows agents to coordinate via shared parameter estimates rather than raw cost values. Given the large state-action space, we also assume a linear model of transitions, commonly used in recent literature. [14, 23]

**Assumption 1.** *For every $(\boldsymbol{s}, \boldsymbol{a}, \boldsymbol{s}') \in \mathcal{S} \times \mathcal{A} \times \mathcal{S}$, there exists known features $\phi(\boldsymbol{s}'|\boldsymbol{s}, \boldsymbol{a}) \in \mathbb{R}^{nd}$ and unknown parameter $\theta \in \mathbb{R}^{nd}$ with $d \geq 2$ such that $\mathbb{P}(\boldsymbol{s}'|\boldsymbol{s}, \boldsymbol{a}) = \langle \phi(\boldsymbol{s}'|\boldsymbol{s}, \boldsymbol{a}), \theta \rangle$.*

We design the features $\phi(\boldsymbol{s}'|\boldsymbol{s}, \boldsymbol{a})$ specific to our setting in the next section. We refer to this setting as the Linear Mixture Multi-Agent Stochastic Shortest Path (LM-MASSP).

**Algorithm.** Analogous to [10, 15], we define a general (possibly randomized) Dec-MARL algorithm as a sequence of deterministic functions $\pi = \{\pi^t\}_{t=1}^{\infty}$, where at each time step $t$, the algorithm specifies a policy $\pi^t = (\pi_1^t, \ldots, \pi_n^t)$. Here $\pi_i^t : \mathcal{H}_i^t \to \Delta(\mathcal{A}_i)$ maps the history observed by agent $i$ to a distribution over its action space $\mathcal{A}_i$, from which the agent samples its action. Each history $\mathcal{H}_i^t$ consists of the agent's past observations- own actions and rewards, along with any information it may have received from other agents in the network. To maintain generality, we do not impose any specific communication protocol; thus, our analysis and results hold for a broad range of settings, from full information sharing to completely communication free (Note that in communication free system, our lower bounds are trivial since the global optimal policy cannot be learned [27]).

**Performance criteria.** Let $(\boldsymbol{s}^{k,h}, \boldsymbol{a}^{k,h})$ denote the state-action pair encountered at step $h$ of episode $k$ under a given algorithm $\pi$ and model parameter $\theta$. Let $h_k$ represent the (random) length of episode $k$. The expected cumulative regret after $K$ episodes is defined as

$$\mathbb{E}_{\theta, \pi}[R(K)] = \mathbb{E}_{\theta, \pi}\left[ \sum_{k=1}^{K} \sum_{h=1}^{h_k} c(\boldsymbol{s}^{k,h}, \boldsymbol{a}^{k,h}) \right] - K \cdot V^*(\boldsymbol{s}_{\text{init}}). \tag{2}$$

The above regret measures the difference between the total expected cost by following the algorithm $\pi$ instead of (unknown) optimal policy. A desirable algorithm achieves sublinear regret in $K$, indicating that it *eventually* learns an optimal policy. In the setting described above, [23] propose an algorithm with expected cumulative regret $\widetilde{\mathcal{O}}(\sqrt{K})$. However, the optimality of this algorithm has not yet been established raising questions such as: Are better learning algorithms possible in this setting? If so, in what aspects can they improve compared to the existing one? To answer these questions, it is essential to derive tight regret lower bounds. Hence, we construct hard-to-learn instances with tractable optimal policies and analyze the regret of any algorithm. Following [2], we consider deterministic algorithms, though similar analysis extends to randomized algorithms, only at the expense of notational overhead.

## 3 Our approach

In this section, we construct a family of provably hard instances for LM-MASSP. These are designed so that the optimal policy is analytically tractable, where any deviation leads to measurable regret.

### 3.1 Construction of hard-to-learn LM-MASSP instances

We consider a minimal yet expressive network with only two nodes: $\mathcal{V} = \{s, g\}$ with $n$ agents. This induces a global state space $\mathcal{S} = \{s, g\}^n$ of size $|\mathcal{S}| = 2^n$, where any agent can either be at $s$ or $g$. We fix the initial state as $\boldsymbol{s}_{\text{init}} = (s, s, \ldots, s)$ and the goal state as $\boldsymbol{g} = (g, g, \ldots, g)$. We assume that once an agent reaches node $g$, it is impossible to go back to node $s$. Each agent $i \in [n]$ has the same action set $\mathcal{A}_i = \{-1, 1\}^{d-1}$, for some $d \geq 2$. Consequently, the global action space is $\mathcal{A} = \{-1, 1\}^{n(d-1)}$ with $|\mathcal{A}| = 2^{n(d-1)}$. We parameterize transition probabilities according to Assumption 1. Next, we define features $\phi(\boldsymbol{s}'|\boldsymbol{s}, \boldsymbol{a})$, which may be of independent interest to the MARL community, as it offers practical guidance for designing valid feature representations[1]. We represent the concatenation of

---

[1]While there are many possible ways to design features, any valid feature construction must satisfy the conditions specified in Lemma 1 below.

vectors $x$ and $y$ by $(x, y)$.

$$\phi(s' \mid s, a) = \begin{cases} \left(\phi_1(s_1'|s_1, a_1)^\top, \ldots, \phi_n(s_n'|s_n, a_n)^\top\right)^\top, & \text{if } s \neq g \text{ and } s_i' = g \text{ if } s_i = g \,\forall\, i \in [n] \\ \mathbf{0}_{nd}, & \text{if } s_i = g, \ s_i' = s \text{ for any } i \in [n] \\ (\mathbf{0}_{nd-1}^\top, 1)^\top, & \text{if } s = g, \ s' = g \end{cases}$$

(3)

where for every agent $i \in \mathcal{N}$ the feature map $\phi_i(s_i' \mid s_i, a_i)$ is defined as follows

$$\phi_i(s_i'|s_i, a_i) = \begin{cases} \left(-a_i^\top, \frac{1-\delta}{n \cdot 2^{r-1}}\right)^\top, & \text{if } s_i = s_i' = s \\ \left(a_i^\top, \frac{\delta}{n \cdot 2^{r-1}}\right)^\top, & \text{if } s_i = s, \ s_i' = g \\ \left(\mathbf{0}_{d-1}^\top, \frac{1}{n \cdot 2^r}\right)^\top, & \text{if } s_i = g, \ s_i' = g. \end{cases}$$

(4)

In the above, $r$ denotes the number of agents at node $s$ in the current global state $s \in \mathcal{S}$. The parameters $\delta$ and $\Delta$ are chosen such that $\delta \in \left(\frac{2}{5}, \frac{1}{2}\right)$ and $\Delta < 2^{-n}\left(\frac{1-2\delta}{1+n+n^2}\right)$ (details are available in the next section). Furthermore, the transition model is parameterized by a vector $\theta \in \mathbb{R}^{nd}$ of the form $\theta = (\theta_1, 1, \theta_2, 1, \ldots, \theta_n, 1)^\top$, where each $\theta_i \in \left\{-\frac{\Delta}{n(d-1)}, \frac{\Delta}{n(d-1)}\right\}^{d-1}$. Let the set of all such $\theta$ be $\Theta$. Thus, $|\Theta| = 2^{n(d-1)}$. For any $i \in [n]$ and $p \in [d-1]$, we denote the $p^{th}$ component of $a_i$ and $\theta_i$ as $a_{i,p}$ and $\theta_{i,p}$, respectively. These parameters along with the features $\phi(.|.,.)$ define the transition probabilities as in Assumption 1. Observe that the first case in Eqn (3) captures general transitions, the second captures disallowed transitions and the third corresponds to the only *certain* transition.

**Instances.** With the above construction, we define hard-to-learn LM-MASSP instances by varying key problem parameters while maintaining analytical tractability of the optimal policy. In particular, we define each instance by a fixed tuple $(n, \delta, \Delta, \theta)$ as defined above.

**Lemma 1.** *For any instance $(n, \delta, \Delta, \theta)$, features defined in Equations (3) and (4) yield valid transition probabilities, i.e.,*

1. *We have, $\langle \phi(s'|s, a), \theta \rangle \geq 0 \,\forall\, (s, a, s')$, and $\sum_{s' \in \mathcal{S}} \langle \phi(s'|s, a), \theta \rangle = 1 \,\forall\, (s, a) \in \mathcal{S} \times \mathcal{A}$.*

2. *For any $(s, a) \in \mathcal{S} \times \mathcal{A}$, if transition to some $s' \in \mathcal{S}$ is impossible, we have $\langle \phi(s'|s, a), \theta \rangle = 0$; on the other hand, if the transition is certain, then $\langle \phi(s'|s, a), \theta \rangle = 1$.*

The detailed proof of this lemma is given in Appendix C.1. Analysis of a general SSP instance is complicated due to trade-offs between policies that quickly reach the goal state (minimizing time) and those that take longer but are cost-efficient [20]. These make optimal policies, in general, intractable. To avoid this, we adopt a uniform cost structure i.e., $\forall\, a$, $c(s, a) = 1 \,\forall\, s \neq g$ and $c(g, a) = 0$. Thus, the optimal policy is the one that minimizes the expected time to reach goal and depends only on transition dynamics. Note that the above cost structure is not merely hypothetical; in our instance design, for all $i \in [n]$, $a \in \mathcal{A}$, and $s \neq g$, we define $K^i(s, a) = \frac{1}{x_i(s, a)}$ and set $K^i(g, a) = 0$.

## 3.2 Transition probabilities

Using above features and the transition model in Assumption 1, we now derive a general expression for transition probabilities for any instance $(n, \delta, \Delta, \theta)$ in the lemma below. We partition the state space $\mathcal{S}$ based on the number of agents at node $s$. Let $\mathcal{S}_r$ be the subset of state space with $r$ agents at node $s$, i.e., $\mathcal{S}_r := \left\{(s_1, s_2, \ldots, s_n) \in \mathcal{S} : \sum_{i \in \mathcal{N}} \mathbb{1}_{\{s_i = s\}} = r\right\}$. For any $r \in \{0\} \cup [n]$, we say that a state $s$ is of *type* $r$ if $s \in \mathcal{S}_r$. Clearly, $\cup_{r=0}^n \mathcal{S}_r = \mathcal{S}$ and $\mathcal{S}_{r_1} \cap \mathcal{S}_{r_2} = \emptyset$ for any $r_1 \neq r_2$. Thus, $\{\mathcal{S}_0, \mathcal{S}_1, \ldots, \mathcal{S}_n\}$ partitions $\mathcal{S}$. Further, for any $s$, we define $\mathcal{S}(s) := \{s' \in \mathcal{S} : \mathbb{P}(s'|s, a) > 0, \text{ for some } a \in \mathcal{A}\}$ as the set of all states that are *reachable* from $s$ in the next step. Also let $\mathcal{S}_r(s)$ be the set of all states of *type* $r$ that are *reachable* in the next step from $s$, i.e., $\mathcal{S}_r(s) := \mathcal{S}_r \cap \mathcal{S}(s)$.

**Lemma 2.** *For a given state $s \in \mathcal{S} \setminus g$, let $\mathcal{I}$ denote the set of agents at node $s$. Consider a transition to $s' \in \mathcal{S}(s)$ under joint action $a \in \mathcal{A}$. Let $\mathcal{T} \subseteq \mathcal{I}$ be the set of agents moving from $s$ to $g$. Define $r = |\mathcal{I}|$, $r' = |\mathcal{I} \setminus \mathcal{T}|$ and $\mathcal{T}' = \mathcal{N} \setminus \mathcal{T}$. Then, under our proposed feature construction and Assumption 1 on the instances, we have*

$$\mathbb{P}(s'|s, a) = \frac{r' + (r - 2r')\delta}{n \cdot 2^{r-1}} + \frac{n-r}{n \cdot 2^r} + \frac{\Delta}{n}(r - 2r')$$

$$+ \frac{2\Delta}{n(d-1)} \sum_{p=1}^{d-1} \Big[ \sum_{i \in \mathcal{I} \cap \mathcal{T}'} \mathbb{1}\{sgn(a_{i,p}) \neq sgn(\theta_{i,p})\} - \sum_{t \in \mathcal{T}} \mathbb{1}\{sgn(a_{t,p}) \neq sgn(\theta_{t,p})\} \Big].$$

**Corollary 1.** *Following Lemma 2, if each agent $i \in \mathcal{N}$ selects its action according to $a_{i,j} = sgn(\theta_{i,j}) \; \forall \; j \in [d-1]$, the resulting transition probability is independent of $\mathcal{I}$ and $\mathcal{T}$, and it only depends on $r$ and $r'$. We denote the corresponding global action by $\boldsymbol{a}_\theta$ and resulting transition probabilities by $p^*_{r,r'}$.*

We observe the following property regarding behavior of $p^*_{r,r'}$ with $r$ and $r'$. This helps us identify optimal policies (Theorem 1) as well as derive regret lower bounds (Theorem 2).

**Lemma 3.** *For every $r \in [n-1]$, the following holds: (a) for $r' \in \{0, \ldots, \lfloor \frac{r+1}{2} \rfloor\}$, we have $\binom{r+1}{r'} p^*_{r+1,r'} < \binom{r}{r'} p^*_{r,r'}$ and (b) for $r' \in \{\lfloor \frac{r+1}{2} \rfloor + 1, \ldots, r\}$, we have $\binom{r+1}{r'} p^*_{r+1,r'} > \binom{r}{r'} p^*_{r,r'}$.*

**Why do we expect tight lower bounds for these instances?** In Theorem 1 we will establish that, for any instance $(n, \delta, \Delta, \theta)$, the optimal policy selects action $\boldsymbol{a}_\theta$ in every state. Now consider a sub-optimal policy that differs from $\boldsymbol{a}_\theta$ in only one non-goal state and exactly at one component of global action. Due to the small $\Delta (< 2^{-n})$, the transition probability (from Lemma 2) for any $(\boldsymbol{s}, \boldsymbol{a}, \boldsymbol{s}')$ under this sub-optimal policy is nearly identical to that under optimal policy, implying the value of such a sub-optimal policy is only slightly worse. There are $(2^n - 1) \times n(d-1)$ such policies. Similarly, consider any sub-optimal policy that differs from the optimal at exactly one state but in two different components of global action. Again, due to a small $\Delta$, we can expect the transition probabilities to differ slightly and the value functions for various states to be only marginally worse. At each state, there are $\binom{n(d-1)}{2}$ such sub-optimal policies. Thus, there are $\binom{n(d-1)}{2} \times (2^n - 1)$ such policies. We can similarly obtain that the number of policies differing from the optimal in any two states at any one component of global action is even higher $\binom{2^n-1}{2} \times n(d-1) \times n(d-1)$. So many near-optimal policies make learning the optimal policy extremely challenging for any algorithm.

# 4 Theoretical results

We now present our main theoretical findings. Our goals are twofold: (i) to characterize the structure of optimal policies and value functions in our LM-MASSP instances and (ii) to establish a tight regret lower bound for any decentralized learning algorithm in this setting. Intuitively, any state with less agents at the initial node $s$ is expected to be *closer* to the *goal* state than any other state with more agents at $s$. Thus, it is desirable to have the state with less agents at $s$ have a relatively lower cumulative cost-to-go. Such monotonicity allows us to reason about the value function at a coarser level depending on the number of agents at node $s$. Leveraging this structure, we characterize the optimal policy and value function in the following theorem.

**Theorem 1** (Structure of the optimal value and policy). *For any instance $(n, \delta, \Delta, \theta)$,*

*(a) the policy that selects action $\boldsymbol{a}_\theta$ (as given in Corollary 1) in every state $\boldsymbol{s} \in \mathcal{S}$ is optimal.*

*(b) the optimal value is same for all states of same type, i.e., for every $r \in [n]$, we have, $V^*(\boldsymbol{s}) = V^*_r$ for every $\boldsymbol{s} \in \mathcal{S}_r$, where $V^*_r$ depends only on $r$. We define $V^*_0 = 0$.*

*(c) the optimal value function is strictly increasing with $r$, i.e., $0 = V^*_0 < V^*_1 < \cdots < V^*_n = B^*$.*

The detailed proof is deferred to Appendix A. The theorem reveals that our instances admit a simple optimal policy depending only on a parameter $\theta$. Despite exponential state space, the optimal value function is fully characterized by its *type*. We emphasize that this result is critical, enabling our regret analysis and may be of independent interest in broader classes of symmetric decentralized multi-agent settings.

We now present our main result: regret lower bound for decentralized learning in LM-MASSPs. It matches the upper bound of [14] (for $n = 1$) and reported upper bound in [23] for a general $n$ in number of episodes, $K$.

**Theorem 2** (Lower bound on regret). *Let $\pi$ be any decentralized learning algorithm. For every $n \geq 1$, $\delta \in (2/5, 1/2)$, and $\Delta < 2^{-n}(\frac{1-2\delta}{1+n+n^2})$, there exists an LM-MASSP instance ($\theta \in \Theta$) such*

that, after $K > \frac{n(d-1)^2 \cdot \delta}{2^{10} B^* \left( \frac{1-2\delta}{1+n+n^2} \right)^2}$ episodes, the expected cumulative regret incurred by $\pi$ satisfies

$$\mathbb{E}_{\theta,\pi}[R(K)] \geq \frac{d \cdot \sqrt{\delta} \cdot \sqrt{K \cdot B^*/n}}{2^{n+9}}. \tag{29}$$

The proof is provided in Appendix B. This theorem highlights that even with strong structure and symmetry in the instances, any decentralized learning algorithm suffers from fundamental information bottlenecks, and no learning algorithm can escape the fundamental $\Omega(\sqrt{K})$ regret. Moreover, our lower bound recovers the previously established single-agent result [14] as a special case at $n = 1$.

**Remark 1.** *The exponential decay of regret lower bound with $n$ may not be misinterpreted as weakening of the bound with increasing $n$. To analyze regret scaling with $n$, one must fix parameters $(\delta, \Delta, \theta)$. Since $\Delta < 2^{-n}(\frac{1-2\delta}{1+n+n^2})$, it inherently fixes the range on $n$. For any valid instance, there exist exponentially many near-optimal policies (see discussion after Lemma 3). In these instances, as $n$ increases within this range, more suboptimal policies get closer to optimal, making learning significantly harder. The corresponding decay of the lower bound reflects the increased difficulty.*

## 5 Outline of proofs

### 5.1 Proof sketch of Theorem 1

*Proof.* We use the Mathematical induction on $r$, the number of agents at node $s$ in any $s \in \mathcal{S}$. All the claims trivially hold for $r = 0$. So, we consider $r = 1$ as the base case.

**Base case ($r = 1$).** For any state $s \in \mathcal{S}_1$, the only *reachable* next state is either $s$ or $g$. For any action $a \in \mathcal{A}$, value of the policy $\pi_a$ that always selects action $a$ at $s$ is given by $V^{\pi_a}(s) = \frac{1}{1-\mathbb{P}(s|s,a)}$. Using Lemma 2, one observes that the self-transition probability $\mathbb{P}(s|s,a)$ is minimized at $a = a_\theta$. This establishes the first claim. The above arguments hold for every $s \in \mathcal{S}_1$ and using corollary 1, the value is same for every state in $\mathcal{S}_1$. This completes the second claim. The third claim holds as $V_1^* > 0$.

**Inductive Step.** Assume the three claims of the theorem hold for all $r \in \{1, 2, \ldots, k\}$. We now show they also hold for $r = k + 1$, in the same order. Consider any state $s \in \mathcal{S}_{k+1}$. To prove the first claim, we evaluate the value $V^{\pi_a}(s)$ under a policy $\pi_a$ that always takes action $a$ at $s$ and optimal action $a_\theta$ in other *reachable* states. By the inductive hypothesis, values and optimal actions are known for all states in $\cup_{i=0}^{k} \mathcal{S}_i$. Since all states reachable from $s$ lie in this set, i.e., $\cup_{i \in [k] \cup \{0\}} \mathcal{S}_i(s) \subseteq \cup_{i \in [k] \cup \{0\}} \mathcal{S}_i$, the value $V^{\pi_a}(s)$ is given by

$$V^{\pi_a}(s) = 1 + \mathbb{P}(s|s,a)V^{\pi_a}(s) + \mathbb{P}(g|s,a)V^*(g) + \sum_{r' \in [k]} \sum_{s' \in \mathcal{S}_{r'}(s)} \mathbb{P}(s'|s,a)V_{r'}^*. \tag{5}$$

Using the inductive hypothesis and Lemma 2, we show that $V^{\pi_a}(s)$ is minimized when $a = a_\theta \; \forall s \in \mathcal{S}_{k+1}$, proving the first claim. The second claim follows directly from Corollary 1 and Eqn (5). For the third claim, we use Lemma 3 along with the following result (see appendix C.5).

**Lemma 4.** *Let $r \in [n-1]$. Suppose that for every $r' \in \{0, 1, \ldots, \lfloor \frac{r+1}{2} \rfloor\}$, we have $\binom{r+1}{r'} p_{r+1,r'}^* < \binom{r}{r'} p_{r,r'}^*$, and for every $r' \in \{\lfloor \frac{r+1}{2} \rfloor + 1, \ldots, r\}$, we have $\binom{r+1}{r'} p_{r+1,r'}^* > \binom{r}{r'} p_{r,r'}^*$. Further, if $V_r^* > V_{r-1}^* > \cdots > V_1^* > V_0^* = 0$, then $V_{r+1}^* > V_r^*$.*

This proves the final claim for $r = k + 1$. By induction, the claim holds for all $r = \{0, 1, \ldots, n\}$. $\square$

### 5.2 Proof sketch of Theorem 2

*Proof.* Consider instances $(n, \delta, \Delta, \theta)$ where $n \geq 1$, $\delta \in (2/5, 1/2)$ and $\Delta < 2^{-n}(\frac{1-2\delta}{1+n+n^2})$ are fixed, while $\theta$ varies across instances with corresponding optimal policy $\pi_\theta^*$ as described in Theorem 1. Let $s^t$ and $a^t$ be the global state and action at time $t$, respectively. The expected regret of an algorithm $\pi$ in the first episode is

$$\mathbb{E}_{\theta,\pi}[R_1] = V^\pi(s^1) - V^{\pi_\theta^*}(s^1)$$

$$= \mathbb{E}_{\boldsymbol{a}^1 \sim \pi(\boldsymbol{s}^1),\, \boldsymbol{s}^2 \sim \mathbb{P}(\cdot|\boldsymbol{s}^1,\boldsymbol{a}^1)} \left[ V^\pi(\boldsymbol{s}^2) - V^{\pi_\theta^*}(\boldsymbol{s}^2) \right] + \mathbb{E}_{\boldsymbol{a}^1 \sim \pi(\boldsymbol{s}^1)} \left[ Q^{\pi_\theta^*}(\boldsymbol{s}^1,\boldsymbol{a}^1) \right] - V^{\pi_\theta^*}(\boldsymbol{s}^1) \tag{6}$$

where in the second step we add and subtract the expectation of optimal $Q$-function [19] and use its definition. We emphasize that unlike the value of a state under any Markov stationary policy, the value of any intermediate state $\boldsymbol{s}^t$ for any learning algorithm depends, in general, on the past history till $t$. This implies that the $V^\pi(\boldsymbol{s}^2)$ above is not independent of the history for any fixed $\boldsymbol{s}^2$. The first expectation in Eqn (6) suggests a recurrence. Using definition of $Q^{\pi_\theta^*}(.,.)$ and Lemma 2,

$$\mathbb{E}_{\boldsymbol{a}^1 \sim \pi(\boldsymbol{s}^1)} \left[ Q^{\pi_\theta^*}(\boldsymbol{s}^1,\boldsymbol{a}^1) \right] - V^{\pi_\theta^*}(\boldsymbol{s}^1) = \mathbb{E}_{\boldsymbol{a}^1 \sim \pi(\boldsymbol{s}^1)} \Bigg\{ \sum_{\boldsymbol{s}^2 \neq \boldsymbol{s}^1} \left[ \mathbb{P}(\boldsymbol{s}^2|\boldsymbol{s}^1,\boldsymbol{a}^1) - \mathbb{P}(\boldsymbol{s}^2|\boldsymbol{s}^1,\boldsymbol{a}^{1^*}) \right] V^{\pi_\theta^*}(\boldsymbol{s}^2)$$

$$+ \frac{2\Delta}{n(d-1)} \sum_{i=1}^{n} \sum_{p=1}^{d-1} \mathbb{1}\{\text{sgn}(a_{i,p}^1) \neq \text{sgn}(\theta_{i,p})\} \cdot V^{\pi_\theta^*}(\boldsymbol{s}^1) \Bigg\}.$$

We further lower bound the above using the following lemma (see appendix C.6).

**Lemma 5.** *For any of our instances, $\forall\, (\boldsymbol{s},\boldsymbol{a}) \in \mathcal{S} \times \mathcal{A}$, $\sum_{\boldsymbol{s}' \neq \boldsymbol{s}} \{\mathbb{P}(\boldsymbol{s}'|\boldsymbol{s},\boldsymbol{a}) - \mathbb{P}(\boldsymbol{s}'|\boldsymbol{s},\boldsymbol{a}^*)\} V^*(\boldsymbol{s}') \geq 0$.*

Further, unrolling the relationship in Eqn (6) recursively over all time steps $h$ in the first episode and applying the third claim in Theorem 1 to lower bound $V^{\pi_\theta^*}(\boldsymbol{s}')$ for each non-goal state $\boldsymbol{s}'$, we get

$$\mathbb{E}_{\theta,\pi}[R_1] \geq \frac{2\Delta V_1^*}{n(d-1)} \sum_{h=1}^{\infty} \mathbb{E}_{\theta,\pi} \left[ \sum_{i \in \mathcal{I}_h} \sum_{p=1}^{d-1} \mathbb{1}\{\boldsymbol{s}^h \neq \text{goal}\} \cdot \mathbb{1}\{\text{sgn}(a_{i,p}^h) \neq \text{sgn}(\theta_{i,p})\} \right]. \tag{7}$$

where $\mathcal{I}_h$ denotes the set of agents at node $s$ at time step $h$. Extending this over all the $K$ episodes, the lower bound can be expressed as

$$\mathbb{E}_{\theta,\pi}[R(K)] \geq \frac{2\Delta V_1^*}{n(d-1)} \mathbb{E}_{\theta,\pi} \Big\{ \sum_{i=1}^{n} \sum_{j=1}^{d-1} N_{i,j}(\theta) \Big\}, \tag{8}$$

where $N_{i,j}(\theta) = \sum_{t=1}^{\infty} \mathbb{1}\{s_i^t \neq g\} \cdot \mathbb{1}\{\text{sgn}(a_{i,j}^t) \neq \text{sgn}(\theta_{i,j})\}$ for each $i \in [n], j \in [d-1]$. It is important to note that we cannot use the standard Pinsker's inequality here to bound the regret, as $N_{i,j}(\theta)$ can be unbounded for example, in case of trivial algorithms. To handle this, we define the truncated sums up to a predefined $T$ as $N_i^- = \sum_{t=1}^{T} \mathbb{1}\{s_i^t \neq g\}$, $N^- = \max_{i \in [n]} N_i^-$ and $N_{i,j}^-(\theta) = \sum_{t=1}^{T} \mathbb{1}\{s_i^t \neq g\} \cdot \mathbb{1}\{\text{sgn}(a_{i,j}^t) \neq \text{sgn}(\theta_{i,j})\}$ for any agent $i \in [n]$ and $j \in [d-1]$. Thus, the regret upto $K$ episodes can be further lower bounded as

$$\mathbb{E}_{\theta,\pi}[R(K)] \geq \frac{2\Delta V_1^*}{n(d-1)} \mathbb{E}_{\theta,\pi} \left[ \sum_{i=1}^{n} \sum_{j=1}^{d-1} N_{i,j}^-(\theta) \right]. \tag{9}$$

For every $\theta = (\theta_1, 1, \theta_2, 1 \ldots, \theta_n, 1) \in \Theta$ and $j \in [d-1]$, let $\theta^j = (\theta_1', 1, \theta_2', 1, \ldots, \theta_n', 1)$, where for every $i \in [n]$, $\theta_i'$ is identical to $\theta_i$ except at the $j^{\text{th}}$ coordinate, where $\theta_{i,j}' = -\theta_{i,j}$. Thus, we have

$$2 \sum_{\theta \in \Theta} \mathbb{E}_{\theta,\pi}[R(K)] \geq \frac{2\Delta V_1^*}{n(d-1)} \sum_{\theta \in \Theta} \sum_{i=1}^{n} \sum_{j=1}^{d-1} \left[ \mathbb{E}_{\theta,\pi}[N_{i,j}^-(\theta)] + \mathbb{E}_{\theta^j,\pi}[N_{i,j}^-(\theta^j)] \right] \tag{10}$$

$$= \frac{2\Delta V_1^*}{n(d-1)} \sum_{\theta \in \Theta} \sum_{i=1}^{n} \sum_{j=1}^{d-1} \left[ \mathbb{E}_{\theta,\pi}[N_i^-] + \mathbb{E}_{\theta,\pi}[N_{i,j}^-(\theta)] - \mathbb{E}_{\theta^j,\pi}[N_{i,j}^-(\theta)] \right]. \tag{11}$$

The first step also uses the fact that $V_1^*$ is independent of $\theta$ and the last step follows since $N_i^- = N_{i,j}^-(\theta) + N_{i,j}^-(\theta^j)$ for any $i \in [n]$, $j \in [d-1]$ along with noting that the map $\theta \to \theta^j$ is bijective for any $j \in [d-1]$. To simplify the first expectation, we generalize Lemma C.2 in [18] as follows

**Lemma 6.** *For any instance $(n, \delta, \Delta, \theta)$, algorithm $\pi$, and $T \geq 2KV_1^*$, we have $\mathbb{E}_{\theta,\pi}[N_i^-] \geq \frac{KV_1^*}{4}$ for any $i \in [n]$.*

This bounds the first term in Eqn (11). The remaining terms capture the difference in expectations under different model parameters. Applying the Pinsker's inequality (Lemma E.3 in [14]), we get

$$2\sum_{\theta\in\Theta}\mathbb{E}_{\theta,\pi}[R(K)] \geq \frac{2\Delta V_1^*}{n(d-1)}\sum_{\theta\in\Theta}\sum_{i=1}^{n}\sum_{j=1}^{d-1}\left\{\frac{KV_1^*}{4} - T\sqrt{\frac{\log 2}{2}}\cdot\sqrt{\mathrm{KL}(\mathbb{P}_\theta^\pi\|\mathbb{P}_{\theta^j}^\pi)}\right\} \qquad (12)$$

In the above, $\mathbb{P}_\theta^\pi$ represents the distribution over $T$-length state sample paths due to interaction of algorithm $\pi$ and the transition dynamics described by parameter $\theta$. Bounding KL divergence in the multi-agent setting is challenging due to exponential number of next states and actions. Since existing techniques ( [11], [14]) do not directly apply, we leverage symmetries in our instances to derive an appropriate upper bound on the KL divergence.

**Lemma 7.** *For any instance $(n,\delta,\Delta,\theta)$ and algorithm $\pi$, for any $j\in[d-1]$, we have,*

$$\mathrm{KL}(\mathbb{P}_\theta^\pi\|\mathbb{P}_{\theta^j}^\pi) \leq 3\cdot 2^{2n}\cdot\frac{\Delta^2}{\delta(d-1)^2}\cdot\mathbb{E}_{\theta,\pi}[N^-]. \qquad (64)$$

The proof of this lemma is deferred to Appendix C.8. Using Lemma 7, $\mathbb{E}_{\theta,\pi}[N^-]\leq T = 2KV_1^*$ and picking $\Delta = \Delta^* = \frac{(d-1)\sqrt{\delta}}{2^{n+5}\sqrt{KV_1^*}}$ in Eqn (12) we get

$$\frac{1}{|\Theta|}\sum_{\theta\in\Theta}\mathbb{E}_{\theta,\pi}[R(K)] \geq \frac{(d-1)\cdot\sqrt{\delta}\cdot\sqrt{KV_1^*}}{2^{n+8}} \geq \frac{d\cdot\sqrt{\delta}\cdot\sqrt{KV_1^*}}{2^{n+9}} \geq \frac{d\cdot\sqrt{\delta}\cdot\sqrt{KB^*/n}}{2^{n+9}}, \quad (13)$$

where the last inequality holds due to the result $V_1^*\geq B^*/n$ (see C.11) for any of our instances. Since this is a lower bound on the average of expected regret over $\Theta$, there must exist at least one instance $\theta\in\Theta$ for which the lower bound in Eqn (13) holds. Finally, to ensure that the chosen $\Delta = \Delta^*$ is valid, i.e., $\Delta < 2^{-n}\left(\frac{1-2\delta}{1+n+n^2}\right)$, we obtain the following sufficient condition $K > \frac{n(d-1)^2\cdot\delta}{2^{10}\cdot B^*\cdot\left(\frac{1-2\delta}{1+n+n^2}\right)^2}$.

This completes the proof. □

## 6 Discussion

This work takes a first step towards closing the gap in our understanding of Dec-MASSPs under linear function approximations. We establish the first regret lower bounds for this setting, obtained by carefully constructed hard-to-learn instances whose complexity scales with the number of agents. By leveraging symmetry-based arguments, we identify the structure of optimal policies and their values. Our $\Omega(\sqrt{K})$ regret lower bound over $K$ episodes underscores the fundamental difficulty of decentralized learning in stochastic environments. These findings deepen the theoretical understanding of decentralized control and can provide a foundation for designing more efficient and scalable algorithms in MARL. It is important to recognize that our instances, created for the purpose of establishing fundamental limits of learning in the broad MASSP setting, may not be ubiquitous in practice. We also remark that, depending on the application, it may be possible to exploit specific information to design learning algorithms (but limited to the particular class of application) with sub-$\sqrt{K}$ regret. We also note that our lower bound theorem is an existence result i.e., Theorem 2 doesn't help identify which of the exponentially many instances is *hard*-enough for a given algorithm hence, complicating empirical validations. Furthermore, due to lack of multiple algorithms in this setting, a run-time comparison over the constructed hard instances is currently infeasible but would be an interesting direction to pursue when more algorithms are available in the future.

The design of our MASSP instances is based on linear function approximations [14, 24, 23]; thus, an important extension of this work is to obtain suitable regret lower bounds with non-linear function approximations. It would also be interesting to investigate how both the regret upper and lower bounds behave under model mis-specifications. Furthermore, in order to have widely applicable results, we did not restrict to a specific communication protocol among the learning agents. Yet, we managed to close the gap in number of episodes, $K$. This calls for a study investigating the effect of various levels of communication among the agents.

An important open direction is to formally understand the increased complexity of decentralized MASSPs compared to single-agent SSPs. In particular, demonstrating a separation where the regret lower bound for a Dec-MASSP instance exceeds the upper bound for single-agent settings would establish fundamental hardness of MASSPs over single-agent SSPs.

## Acknowledgments and Disclosure of Funding

The authors would like to express their sincere gratitude to their institutes for providing the necessary resources. They are grateful to the Reviewers and the Area Chair whose valuable feedback and constructive suggestions have greatly helped improve the quality of this work. Utkarsh acknowledges support from the NeurIPS Foundation for the Travel Award and from IIT Bombay-FedEx ALFA for the partial travel support. He also thanks his home Department of Mechanical Engineering at IIT Bombay for allowing him the flexibility of spending his final undergrad year at the Department of IEOR IIT Bombay as well as accepting this work as his undergraduate thesis. He also acknowledges the Teaching Assistantships provided by IIT Bombay in his final year. A portion of this work was carried out while Prashant was a Postdoctoral Scholar at the University of Central Florida, USA, under the mentorship of Dr. Amrit Singh Bedi. Prashant would like to thank him for providing the flexibility to work on this problem.

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

## Appendix

In this appendix, we provide additional notation, definitions, and proofs of the theorems and intermediate results that support the main findings presented in the paper.

More details of proofs of Theorem 1 and Theorem 2 are in Appendix A and Appendix B respectively. Also, proofs of some other intermediate results, used in Appendices A and B, are in Appendix C.

We first recall the intermediate results that we use to prove the main results, i.e., Theorem 1 and Theorem 2.

**Lemma 1.** *For any instance* $(n, \delta, \Delta, \theta)$*, features defined in Equations* (3) *and* (4) *yield valid transition probabilities, i.e.,*

1. *We have,* $\langle \phi(s'|s, a), \theta \rangle \geq 0 \ \forall \ (s, a, s')$*, and* $\sum_{s' \in \mathcal{S}} \langle \phi(s'|s, a), \theta \rangle = 1 \ \forall \ (s, a) \in \mathcal{S} \times \mathcal{A}$*.*

2. *For any* $(s, a) \in \mathcal{S} \times \mathcal{A}$*, if transition to some* $s' \in \mathcal{S}$ *is impossible, we have* $\langle \phi(s'|s, a), \theta \rangle = 0$*; on the other hand, if the transition is certain, then* $\langle \phi(s'|s, a), \theta \rangle = 1$*.*

**Lemma 2.** *For a given state* $s \in \mathcal{S} \setminus g$*, let* $\mathcal{I}$ *denote the set of agents at node* $s$*. Consider a transition to* $s' \in \mathcal{S}(s)$ *under joint action* $a \in \mathcal{A}$*. Let* $\mathcal{T} \subseteq \mathcal{I}$ *be the set of agents moving from* $s$ *to* $g$*. Define* $r = |\mathcal{I}|$*,* $r' = |\mathcal{I} \setminus \mathcal{T}|$ *and* $\mathcal{T}' = \mathcal{N} \setminus \mathcal{T}$*. Then, under our proposed feature construction and Assumption 1 on the instances, we have*

$$\mathbb{P}(s'|s, a) = \frac{r' + (r - 2r')\delta}{n \cdot 2^{r-1}} + \frac{n-r}{n \cdot 2^r} + \frac{\Delta}{n}(r - 2r')$$

$$+ \frac{2\Delta}{n(d-1)} \sum_{p=1}^{d-1} \Big[ \sum_{i \in \mathcal{I} \cap \mathcal{T}'} \mathbb{1}\{sgn(a_{i,p}) \neq sgn(\theta_{i,p})\} - \sum_{t \in \mathcal{T}} \mathbb{1}\{sgn(a_{t,p}) \neq sgn(\theta_{t,p})\} \Big].$$

**Corollary 1.** *Following Lemma 2, if each agent* $i \in \mathcal{N}$ *selects its action according to* $a_{i,j} = sgn(\theta_{i,j}) \ \forall \ j \in [d-1]$*, the resulting transition probability is independent of* $\mathcal{I}$ *and* $\mathcal{T}$*, and it only depends on* $r$ *and* $r'$*. We denote the corresponding global action by* $a_\theta$ *and resulting transition probabilities by* $p^*_{r,r'}$*.*

**Lemma 3.** *For every* $r \in [n-1]$*, the following holds: (a) for* $r' \in \{0, \ldots, \lfloor \frac{r+1}{2} \rfloor\}$*, we have* $\binom{r+1}{r'} p^*_{r+1,r'} < \binom{r}{r'} p^*_{r,r'}$ *and (b) for* $r' \in \{\lfloor \frac{r+1}{2} \rfloor + 1, \ldots, r\}$*, we have* $\binom{r+1}{r'} p^*_{r+1,r'} > \binom{r}{r'} p^*_{r,r'}$*.*

## A   Proof of Theorem 1

**Theorem 1** (Structure of the optimal value and policy)**.** *For any instance* $(n, \delta, \Delta, \theta)$*,*

*(a) the policy that selects action* $a_\theta$ *(as given in Corollary 1) in every state* $s \in \mathcal{S}$ *is optimal.*

*(b) the optimal value is same for all states of same type, i.e., for every* $r \in [n]$*, we have,* $V^*(s) = V^*_r$ *for every* $s \in \mathcal{S}_r$*, where* $V^*_r$ *depends only on* $r$*. We define* $V^*_0 = 0$*.*

*(c) the optimal value function is strictly increasing with* $r$*, i.e.,* $0 = V^*_0 < V^*_1 < \cdots < V^*_n = B^*$*.*

*Proof.* The simplest case is when all the agents are at the goal node, i.e., $r = 0$, i.e., $s \in \mathcal{S}_0$. Note that in this case $|\mathcal{S}_0| = 1$, that is when $r = 0$, there is only one state $g = (g, g, \ldots, g)$.

For this case, all the actions at goal node are optimal, this is because no matter what action is taken at the goal state $g$, the cost is same i.e., 0. Moreover, the next state is certainly $g$ (by construction of features). Hence, in particular, the action $a_\theta$ is also optimal. Hence the claim (a) of the theorem holds.

Next, since $|\mathcal{S}_0| = 1$, the claim (b) also holds trivially. That is, $V^*(s) = V^*_0 = 0$ for all $s \in \mathcal{S}_0$. Finally, the claim (c) also follows by assumption, as $V^*_0 = 0 < B^*$.

Now, we begin to prove the theorem for the interesting states, i.e., states in which at least one agent is at the node $s$. We will prove all the claims of this theorem using the principle of Mathematical induction on the type of states $r = 1, \ldots, n$. To this end, first consider the base case of $r = 1$.

**Base case, $r = 1$.** For any state $s \in \mathcal{S}_1$, we first find the optimal action in this state. Note that for this state, there are only two possible next global states, $s$ itself and the global goal state $g$. This may seem very similar to the single agent problem.

Suppose $\mathbb{P}(s|s, a)$ denotes the transition probability from $s$ to $s$ for any $a \in \mathcal{A}$. To determine the optimal action at this state, we consider any policy $\pi_a$ that takes action $a \in \mathcal{A}$ in the state $s$ and the optimal action at other states that are reachable from $s$. Note that, above we argued that the action $a_\theta$ is optimal at $g$ and hence the value of this policy at $s$ can be expressed as

$$V^{\pi_a}(s) = \mathbb{P}(s|s, a) \cdot (1 + V^{\pi_a}(s)) + (1 - \mathbb{P}(s|s, a)) \cdot (1 + V_0^*) \tag{14}$$

Simplifying the above expression using $V_0^* = 0$, we have

$$V^{\pi_a}(s) = \frac{1}{1 - \mathbb{P}(s|s, a)}. \tag{15}$$

Thus, the optimal action at $s$ is the one that minimizes $V^{\pi_a}(s)$ i.e., minimizes $\mathbb{P}(s|s, a)$. We now use the following corollary of Lemma 2; the proof of Corollary 2 is given in Appendix C.10.

**Corollary 2.** *For any non-goal state $s \in \mathcal{S}_r$ of type $r$, the self-transition probability under action $a$ is*

$$\mathbb{P}(s|s, a) = \frac{r(1 - \delta)}{n \cdot 2^{r-1}} + \frac{n - r}{n \cdot 2^r} - \frac{r \cdot \Delta}{n} + \frac{2\Delta}{n(d - 1)} \sum_{i \in \mathcal{I}} \sum_{p=1}^{d-1} \mathbb{1}\{sgn(a_{i,p}) \neq sgn(\theta_{i,p})\}. \tag{16}$$

*This is minimized when $a = a_\theta$ and the resulting probability only depends on $r$.*

Using the above, we have that $a_\theta$ is optimal action in this state. Notice, we chose $s \in \mathcal{S}_1$ arbitrarily. Hence the above holds for all states in $\mathcal{S}_1$. This proves claim (a) of the theorem for the base case of $r = 1$.

Now from Corollary 1, we have that

$$\mathbb{P}(s|s, a^*) = p_{1,1}^*, \quad \forall s \in \mathcal{S}_1 \tag{17}$$

where $p_{1,1}^*$ is the probability that is independent of state itself, rather only depends on the number of agents at $s$. Thus, from Equation (15), we have that $V^*(s) = V_1^*$, $\forall s \in \mathcal{S}_1$. This implies that value is same for all states in $\mathcal{S}_1$. Thus claim (b) of the theorem also holds for the base case.

Finally, we have that $V_1^* = \frac{1}{1 - \mathbb{P}(s|s, a_\theta)} > 0 = V_0^*$. This completes claim (c) of the theorem for the base case. Thus, all three claims of Theorem 1 hold for the base case.

**Inductive Step.** Next, let us assume that all three claims of the theorem hold for $k = 1, 2, \ldots, r$ for any $r < n$. We will prove that the claims hold for $k = r + 1$. In particular, we assume that for all $k \in \{1, \ldots, r\}$, it holds that

1. The optimal action $a^* = a_\theta$ for any state in $\mathcal{S}_k$.

    This implies from Corollary 1 that, the transition probabilities from any $s \in \mathcal{S}_k$ to any $s' \in \mathcal{S}(s)$ can be expressed as $p_{k,k'}^*$, for any $k' \in \{0, 1, 2, \ldots, k\}$ under the optimal policy.

2. $V_k^*(s) = V_k^*$ for all $s \in \mathcal{S}_k$

3. $V_k^* > V_{k-1}^*$

We will now show that all the three claims hold for $k = r + 1$.

Consider any $s \in \mathcal{S}_{r+1}$. We again want to know the optimal action in this state. The optimal action in all the other states that are *reachable* from $s$ are known. Hence, we consider any policy $\pi_a$ that takes $a \in \mathcal{A}$ at this state and optimal actions at all other states that are *reachable* from here. The value of this policy can be expressed as follows

$$V^{\pi_a}(s) \tag{18}$$

$$= \sum_{s' \in \mathcal{S}(s) \setminus \{s, g\}} \mathbb{P}(s'|s, a)(1 + V^*(s')) + \mathbb{P}(s|s, a)(1 + V^{\pi_a}(s)) + \mathbb{P}(g|s, a) \cdot (1 + V^*(g)) \tag{19}$$

$$= 1 + \mathbb{P}(\boldsymbol{s}|\boldsymbol{s},\boldsymbol{a})V^{\pi_a}(\boldsymbol{s}) + \mathbb{P}(\boldsymbol{g}|\boldsymbol{s},\boldsymbol{a})V^*(\boldsymbol{g}) + \sum_{r'=1}^{r}\sum_{\boldsymbol{s}'\in\mathcal{S}_{r'}(\boldsymbol{s})}\mathbb{P}(\boldsymbol{s}'|\boldsymbol{s},\boldsymbol{a})V_{r'}^* \tag{20}$$

We will simplify the last term in the above equation using Lemma 2 as follows.

$$\sum_{r'=1}^{r}\sum_{\boldsymbol{s}'\in\mathcal{S}_{r'}(\boldsymbol{s})}\mathbb{P}(\boldsymbol{s}'|\boldsymbol{s},\boldsymbol{a})V_{r'}^*$$

$$= \sum_{r'=1}^{r}\sum_{\boldsymbol{s}'\in\mathcal{S}_{r'}(\boldsymbol{s})}\left[\frac{r'+(r+1-2r')\delta}{n\cdot 2^r} + \frac{n-r-1}{n2^{r+1}} + \frac{\Delta}{n}(r+1-2r')\right.$$
$$\left. + \frac{2\Delta}{n(d-1)}\sum_{p=1}^{d-1}\left(\sum_{i\in\mathcal{I}\cap\mathcal{T}'}\mathbb{1}\{\mathrm{sgn}(a_{i,p})\neq\mathrm{sgn}(\theta_{i,p})\} - \sum_{t\in\mathcal{T}}\mathbb{1}\{\mathrm{sgn}(a_{t,p})\neq\mathrm{sgn}(\theta_{t,p})\}\right)\right]V_{r'}^* \tag{21}$$

$$= \sum_{r'=1}^{r}\left[\left(\frac{r'+(r+1-2r')\delta}{n\cdot 2^r} + \frac{n-r-1}{n2^{r+1}} + \frac{\Delta}{n}(r+1-2r')\right)\cdot\binom{r+1}{r'}\right.$$
$$+ \frac{2\Delta}{n(d-1)}\sum_{p=1}^{d-1}\sum_{\boldsymbol{s}'\in\mathcal{S}_{r'}(\boldsymbol{s})}\left(\sum_{i\in\mathcal{I}\cap\mathcal{T}'}\mathbb{1}\{\mathrm{sgn}(a_{i,p})\neq\mathrm{sgn}(\theta_{i,p})\}\right.$$
$$\left.\left. - \sum_{t\in\mathcal{T}}\mathbb{1}\{\mathrm{sgn}(a_{t,p})\neq\mathrm{sgn}(\theta_{t,p})\}\right)\right]V_{r'}^* \tag{22}$$

The Equation (22) follows by observing that for fixed $r'$, the inner summation over $\boldsymbol{s}'$ in the first line is just the sum of $|\mathcal{S}_{r'}(\boldsymbol{s})| = \binom{r+1}{r'}$ terms of same value.

In the below, we alternatively represent the summation over sets $\mathcal{I}\cap\mathcal{T}'$ and $\mathcal{T}$ as an appropriate summation over $\mathcal{I}$ and in the next step, we take the $V^*(\cdot)$ factor inside and take the summation over two terms:

$$\sum_{r'=1}^{r}\sum_{\boldsymbol{s}'\in\mathcal{S}_{r'}(\boldsymbol{s})}\mathbb{P}(\boldsymbol{s}'|\boldsymbol{s},\boldsymbol{a})V_{r'}^*$$

$$= \sum_{r'=1}^{r}\left[\left(\frac{r'+(r+1-2r')\delta}{n\cdot 2^r} + \frac{n-r-1}{n2^{r+1}} + \frac{\Delta}{n}(r+1-2r')\right)\cdot\binom{r+1}{r'}\right.$$
$$+ \frac{2\Delta}{n(d-1)}\sum_{p=1}^{d-1}\sum_{\boldsymbol{s}'\in\mathcal{S}_{r'}(\boldsymbol{s})}\sum_{i\in\mathcal{I}}\left(\mathbb{1}\{i\in\mathcal{I}\cap\mathcal{T}'\}\cdot\mathbb{1}\{\mathrm{sgn}(a_{i,p})\neq\mathrm{sgn}(\theta_{i,p})\}\right.$$
$$\left.\left. - \mathbb{1}\{i\in\mathcal{T}\}\cdot\mathbb{1}\{\mathrm{sgn}(a_{i,p})\neq\mathrm{sgn}(\theta_{i,p})\}\right)\right]V_{r'}^* \tag{23}$$

$$= \sum_{r'=1}^{r}\left[\left(\frac{r'+(r+1-2r')\delta}{n\cdot 2^r} + \frac{n-r-1}{n2^{r+1}} + \frac{\Delta}{n}(r+1-2r')\right)\cdot\binom{r+1}{r'}V_{r'}^*\right]$$
$$+ \frac{2\Delta}{n(d-1)}\sum_{p=1}^{d-1}\sum_{i\in\mathcal{I}}\sum_{r'=1}^{r}\sum_{\boldsymbol{s}'\in\mathcal{S}_{r'}(\boldsymbol{s})}\left(\mathbb{1}\{i\in\mathcal{I}\cap\mathcal{T}'\}\cdot\mathbb{1}\{\mathrm{sgn}(a_{i,p})\neq\mathrm{sgn}(\theta_{i,p})\}\right.$$
$$\left. - \mathbb{1}\{i\in\mathcal{T}\}\mathbb{1}\{\mathrm{sgn}(a_{i,p})\neq\mathrm{sgn}(\theta_{i,p})\}\right)V_{r'}^* \tag{24}$$

$$= \sum_{r'=1}^{r}\left[\left(\frac{r'+(r+1-2r')\delta}{n\cdot 2^r} + \frac{n-r-1}{n2^{r+1}} + \frac{\Delta}{n}(r+1-2r')\right)\cdot\binom{r+1}{r'}V_{r'}^*\right]$$

$$+ \frac{2\Delta}{n(d-1)} \sum_{p=1}^{d-1} \sum_{i \in \mathcal{I}} \sum_{r'=1}^{r} \left[ \binom{r}{r'-1} \cdot \mathbb{1}\{\mathrm{sgn}(a_{i,p}) \neq \mathrm{sgn}(\theta_{i,p})\} \right.$$

$$\left. - \binom{r}{r'} \cdot \mathbb{1}\{\mathrm{sgn}(a_{i,p}) \neq \mathrm{sgn}(\theta_{i,p})\} \right] V_{r'}^*. \tag{25}$$

In the last step, we identified that for every agent $i \in \mathcal{I}$ for a fixed $r'$ the number of states in $\mathcal{S}_{r'}(s)$ in which the agent $i$ stays at node $s$ itself are $\binom{r}{r'-1}$ and those in which it transits to node $g$ are $\binom{r}{r'}$.

In the immediate below step, we use the following property: for any function $f(\cdot)$ and natural number $m$, we have $2 \cdot \sum_{a \in [m]} f(a) = \sum_{a \in [m]} f(a) + \sum_{a \in [m]} f(1 + m - a)$. In our case, $f$ is the terms inside the summations over $p$, $i$ of Equation (25), $a$ is $r'$ and $m$ is $r$. In the next few steps, we simplify the expression using properties of binomial coefficients:

$$\sum_{r'=1}^{r} \sum_{s' \in \mathcal{S}_{r'}(s)} \mathbb{P}(s'|s,a) V_{r'}^*$$

$$= \sum_{r'=1}^{r} \left[ \left( \frac{r' + (r+1-2r')\delta}{n \cdot 2^r} + \frac{n-r-1}{n2^{r+1}} + \frac{\Delta}{n}(r+1-2r') \right) \cdot \binom{r+1}{r'} \cdot V_{r'}^* \right]$$

$$+ \frac{\Delta}{n(d-1)} \left[ \sum_{p=1}^{d-1} \sum_{i \in \mathcal{I}} \sum_{r'=1}^{r} \left\{ \left( \binom{r}{r'-1} - \binom{r}{r'} \right) \cdot \mathbb{1}\{\mathrm{sgn}(a_{i,p}) \neq \mathrm{sgn}(\theta_{i,p})\} \cdot V_{r'}^* \right\} \right.$$

$$\left. + \sum_{p=1}^{d-1} \sum_{i \in \mathcal{I}} \sum_{r'=1}^{r} \left\{ \left( \binom{r}{r+1-r'-1} - \binom{r}{r+1-r'} \right) \mathbb{1}\{\mathrm{sgn}(a_{i,p}) \neq \mathrm{sgn}(\theta_{i,p})\} V_{r+1-r'}^* \right\} \right] \tag{26}$$

$$= \sum_{r'=1}^{r} \left[ \left( \frac{r' + (r+1-2r')\delta}{n \cdot 2^r} + \frac{n-r-1}{n2^{r+1}} + \frac{\Delta}{n}(r+1-2r') \right) \cdot \binom{r+1}{r'} \cdot V_{r'}^* \right]$$

$$+ \frac{\Delta}{n(d-1)} \sum_{p=1}^{d-1} \sum_{i \in \mathcal{I}} \sum_{r'=1}^{r} \left[ \left( \binom{r}{r'-1} - \binom{r}{r'} \right) \cdot \mathbb{1}\{\mathrm{sgn}(a_{i,p}) \neq \mathrm{sgn}(\theta_{i,p})\} \cdot V_{r'}^* \right]$$

$$+ \frac{\Delta}{n(d-1)} \sum_{p=1}^{d-1} \sum_{i \in \mathcal{I}} \sum_{r'=1}^{r} \left[ \left( \binom{r}{r'} - \binom{r}{r'-1} \right) \cdot \mathbb{1}\{\mathrm{sgn}(a_{i,p}) \neq \mathrm{sgn}(\theta_{i,p})\} \cdot V_{r+1-r'}^* \right] \tag{27}$$

$$= \sum_{r'=1}^{r} \left[ \left( \frac{r' + (r+1-2r')\delta}{n \cdot 2^r} + \frac{n-r-1}{n2^{r+1}} + \frac{\Delta}{n}(r+1-2r') \right) \cdot \binom{r+1}{r'} \cdot V_{r'}^* \right]$$

$$+ \frac{\Delta}{n(d-1)} \sum_{p=1}^{d-1} \sum_{i \in \mathcal{I}} \sum_{r'=1}^{r} \left[ \left( \binom{r}{r'-1} - \binom{r}{r'} \right) \cdot \mathbb{1}\{\mathrm{sgn}(a_{i,p}) \neq \mathrm{sgn}(\theta_{i,p})\} \cdot (V_{r'}^* - V_{r+1-r'}^*) \right] \tag{28}$$

Observe that the term inside the summation is always non negative since when $r' \geq r/2$ the binomial difference $\binom{r}{r'-1} - \binom{r}{r'} \geq 0$ as well as $V_{r'}^* \geq V_{r+1-r'}^*$ while for $r' \leq r/2$, $\binom{r}{r'-1} - \binom{r}{r'} \leq 0$ as well as $V_{r'}^* \leq V_{r+1-r'}^*$ due to property of Binomial coefficients and the Induction assumption on monotonicity of $V^*$. Note that the above argument is for $r' \in [r]$.

To minimize $V^{\pi_a}(s)$, we need to minimize $\mathbb{P}(s|s,a)$ as well (can be seen by rearranging Equation (20) along with the fact that $V^*(g) = 0$ as it will appear in denominator as $1 - \mathbb{P}(s|s,a)$).

Clearly, the action $a_\theta$ minimizes $\mathbb{P}(s|s,a)$ as well as makes the sum of non-negative terms the minimum i.e., 0. Hence, it minimizes $V^{\pi_a}(s)$. Hence this action $a_\theta$ is the optimal action at $s$. Note we had chosen $s \in \mathcal{S}_{r+1}$ arbitratily. Hence, the above arguments hold for any $s \in \mathcal{S}_{r+1}$. This proves claim (a) of the theorem for $r+1$.

Again, by Corollary 1, for any $s \in \mathcal{S}_{r+1}$ and any $s' \in \mathcal{S}_{r'}(s)$, we have $\mathbb{P}(s'|s,a_\theta) = p_{r+1,r'}^*$ for any $r' \in [r] \cup \{0\}$.

Thus, using $\boldsymbol{a} = \boldsymbol{a}_\theta$ in Equation (19), we have $V^*(\boldsymbol{s}) = V^*_{r+1}$ for any $\boldsymbol{s} \in \mathcal{S}_{r+1}$. This proves the claim (b) for $r+1$.

Using the induction assumption combined with the above results, Lemma 3 and the following lemma we have $V^*_{r+1} > V^*_r$, completing the claim (c) for $r+1$.

**Lemma 4.** *Let $r \in [n-1]$. Suppose that for every $r' \in \{0, 1, \ldots, \lfloor \frac{r+1}{2} \rfloor\}$, we have $\binom{r+1}{r'}p^*_{r+1,r'} < \binom{r}{r'}p^*_{r,r'}$, and for every $r' \in \{\lfloor \frac{r+1}{2} \rfloor + 1, \ldots, r\}$, we have $\binom{r+1}{r'}p^*_{r+1,r'} > \binom{r}{r'}p^*_{r,r'}$. Further, if $V^*_r > V^*_{r-1} > \cdots > V^*_1 > V^*_0 = 0$, then $V^*_{r+1} > V^*_r$.*

The proof of Lemma 4 is given in Appendix C.5. Thus, by induction, all the three claims of theorem hold. Note that $V^*_n = B^*$ as it is the maximum optimal value over all states. $\qquad\square$

# B    Proof of Theorem 2

**Theorem 2** (Lower bound on regret). *Let $\pi$ be any decentralized learning algorithm. For every $n \geq 1$, $\delta \in (2/5, 1/2)$, and $\Delta < 2^{-n}(\frac{1-2\delta}{1+n+n^2})$, there exists an LM-MASSP instance $(\theta \in \Theta)$ such that, after $K > \frac{n(d-1)^2 \cdot \delta}{2^{10}B^*\left(\frac{1-2\delta}{1+n+n^2}\right)^2}$ episodes, the expected cumulative regret incurred by $\pi$ satisfies*

$$\mathbb{E}_{\theta,\pi}[R(K)] \geq \frac{d \cdot \sqrt{\delta} \cdot \sqrt{K \cdot B^*/n}}{2^{n+9}}. \tag{29}$$

*Proof.* The proof builds on techniques used in the single-agent SSP setting by [14] and the two-state RL setting in [11]. However, extending these ideas to the Multi-Agent SSP (MASSP) setting introduces several technical challenges. These include: designing novel features, constructing instances that generalize to any number of agents $n$ while admitting a tractable optimal policy, the intractability of expressing the optimal value function in closed form due to exponential growth of the state space, and deriving tight upper bounds on the KL divergence. Some of these challenges are addressed in Theorem 1, while the remaining ones are tackled in the below analysis.

We construct and analyze hard to learn instances based on the discussion in Section 3. In Theorem 1, we characterized the optimal policy. Recall that for our instances, the cost of any global action in any non-goal state is equal to 1 and that in goal state equal to 0 (see discussion at the end of Section 3.1).

Consider any instance parameters $(n \geq 1, \delta \in (2/5, 1/2), \Delta < 2^{-n} \cdot \frac{1-2\delta}{1+n+n^2})$. We want to analyze regret of any algorithm $\pi$ for instances that are defined by fixed parameters $(n, \delta, \Delta)$ and all $\theta \in \Theta$.

Consider any instance $\theta \in \Theta$. Let $\pi^*_\theta$ denote the optimal policy for this instance as in Theorem 1. Suppose in episode $k$, agents take actions according to $\pi$. Let, $\boldsymbol{s}^t$ represent the global state and $\boldsymbol{a}^t$ the global action at time $t$. Note that as we are analyzing any algorithm, it is important to track the time step at which any state is reached and the history till then. Then, the expected regret for this instance in the $1^{st}$ episode is:

$$\mathbb{E}_{\theta,\pi}[R_1] = V^\pi(\boldsymbol{s}^1) - V^{\pi^*_\theta}(\boldsymbol{s}^1) \tag{30}$$

$$= V^\pi(\boldsymbol{s}^1) - \mathbb{E}_{\boldsymbol{a}^1 \sim \pi(\boldsymbol{s}^1)}[Q^{\pi^*_\theta}(\boldsymbol{s}^1, \boldsymbol{a}^1)] + \mathbb{E}_{\boldsymbol{a}^1 \sim \pi(\boldsymbol{s}^1)}[Q^{\pi^*_\theta}(\boldsymbol{s}^1, \boldsymbol{a}^1)] - V^{\pi^*_\theta}(\boldsymbol{s}^1) \tag{31}$$

In the above, we add and subtract $\mathbb{E}_{\boldsymbol{a}^1 \sim \pi(\boldsymbol{s}^1)}[Q^{\pi^*_\theta}(\boldsymbol{s}^1, \boldsymbol{a}^1)]$. Now using the definitions of $V^\pi(\cdot)$ and $Q^{\pi^*_\theta}(\cdot, \cdot)$ and rearranging the terms we have:

$$\mathbb{E}_{\theta,\pi}[R_1] = \mathbb{E}_{\boldsymbol{a}^1 \sim \pi(\boldsymbol{s}^1)}\{c(\boldsymbol{s}^1, \boldsymbol{a}^1) + \mathbb{E}_{\boldsymbol{s}^2 \sim \mathbb{P}(.|\boldsymbol{s}^1, \boldsymbol{a}^1)}[V^\pi(\boldsymbol{s}^2)]\}$$
$$- \mathbb{E}_{\boldsymbol{a}^1 \sim \pi(\boldsymbol{s}^1)}\{c(\boldsymbol{s}^1, \boldsymbol{a}^1) + \mathbb{E}_{\boldsymbol{s}^2 \sim \mathbb{P}(.|\boldsymbol{s}^1, \boldsymbol{a}^1)}[V^{\pi^*_\theta}(\boldsymbol{s}^2)]\} + \mathbb{E}_{\boldsymbol{a}^1 \sim \pi(\boldsymbol{s}^1)}[Q^{\pi^*_\theta}(\boldsymbol{s}^1, \boldsymbol{a}^1)] - V^{\pi^*_\theta}(\boldsymbol{s}^1)$$
$$\tag{32}$$

$$= \mathbb{E}_{\boldsymbol{a}^1 \sim \pi(\boldsymbol{s}^1), \boldsymbol{s}^2 \sim \mathbb{P}(.|\boldsymbol{s}^1, \boldsymbol{a}^1)}[V^\pi(\boldsymbol{s}^2) - V^{\pi^*_\theta}(\boldsymbol{s}^2)] + \mathbb{E}_{\boldsymbol{a}^1 \sim \pi(\boldsymbol{s}^1)}[Q^{\pi^*_\theta}(\boldsymbol{s}^1, \boldsymbol{a}^1)] - V^{\pi^*_\theta}(\boldsymbol{s}^1) \tag{33}$$

In the above expression, there appears to be a recursive relation for the first term however, it is unclear how to deal with the remaining terms. To this end, we expand the second term below using the definition of $Q^*(\cdot, \cdot) = Q^{\pi^*_\theta}$, and $V^*(\cdot) = V^{\pi^*_\theta}$

$$\mathbb{E}_{\boldsymbol{a}_1 \sim \pi(\boldsymbol{s}^1)}[Q^*(\boldsymbol{s}^1, \boldsymbol{a}^1)] - V^*(\boldsymbol{s}^1) = \mathbb{E}_{\boldsymbol{a}^1 \sim \pi(\boldsymbol{s}^1)}[c(\boldsymbol{s}^1, \boldsymbol{a}^1) + \mathbb{E}_{\boldsymbol{s}^2 \sim \mathbb{P}(.|\boldsymbol{s}^1, \boldsymbol{a}^1)}\{V^*(\boldsymbol{s}^2)\}] - V^*(\boldsymbol{s}^1) \tag{34}$$

Above, we use the definition of $Q^*(\cdot, \cdot)$. Next, we use the cost $c(\boldsymbol{s}, \boldsymbol{a}) \equiv 1, \forall (\boldsymbol{s}, \boldsymbol{a}) \in \mathcal{S}\backslash\{\boldsymbol{g}\} \times \mathcal{A}$ and then expand the expectation as

$$\mathbb{E}_{\boldsymbol{a}^1 \sim \pi(\boldsymbol{s}^1)}[Q^*(\boldsymbol{s}^1, \boldsymbol{a}^1)] - V^*(\boldsymbol{s}^1)$$

$$= \mathbb{E}_{\boldsymbol{a}^1 \sim \pi(\boldsymbol{s}^1)}\left[1 + \mathbb{E}_{\boldsymbol{s}^2 \sim \mathbb{P}(\cdot|\boldsymbol{s}^1, \boldsymbol{a}^1)}[V^*(\boldsymbol{s}^2)] - V^*(\boldsymbol{s}^1)\right] \tag{35}$$

$$= \mathbb{E}_{\boldsymbol{a}^1 \sim \pi(\boldsymbol{s}^1)}\left[1 + \mathbb{P}(\boldsymbol{s}^1|\boldsymbol{s}^1, \boldsymbol{a}^1)V^*(\boldsymbol{s}^1) + \sum_{\boldsymbol{s}^2 \neq \boldsymbol{s}^1} \mathbb{P}(\boldsymbol{s}^2|\boldsymbol{s}^1, \boldsymbol{a}^1)V^*(\boldsymbol{s}^2) - V^*(\boldsymbol{s}^1)\right] \tag{36}$$

To simplify above, we use the general transition probability expression from Lemma 2 with $\boldsymbol{s}^1 = (s, s, \dots, s)$, i.e., $r = n$, and then rearrange

$$\mathbb{E}_{\boldsymbol{a}^1 \sim \pi(\boldsymbol{s}^1)}[Q^*(\boldsymbol{s}^1, \boldsymbol{a}^1)] - V^*(\boldsymbol{s}^1)$$

$$= \mathbb{E}_{\boldsymbol{a}^1 \sim \pi(\boldsymbol{s}^1)}\left[1 + \left(\frac{1-\delta}{2^{n-1}} - \Delta + \frac{2\Delta}{n(d-1)}\sum_{i=1}^{n}\sum_{p=1}^{d-1}\mathbb{1}\{\mathrm{sgn}(a_{i,p}^1) \neq \mathrm{sgn}(\theta_{i,p})\}\right)V^*(\boldsymbol{s}^1)\right.$$

$$\left. + \sum_{\boldsymbol{s}^2 \neq \boldsymbol{s}^1} \mathbb{P}(\boldsymbol{s}^2|\boldsymbol{s}^1, \boldsymbol{a}^1)V^*(\boldsymbol{s}^2) - V^*(\boldsymbol{s}^1)\right] \tag{37}$$

$$= \mathbb{E}_{\boldsymbol{a}^1 \sim \pi(\boldsymbol{s}^1)}\left[1 + \left(\frac{1-\delta}{2^{n-1}} - \Delta\right)V^*(\boldsymbol{s}^1) + \sum_{\boldsymbol{s}^2 \neq \boldsymbol{s}^1}\mathbb{P}(\boldsymbol{s}^2|\boldsymbol{s}^1, \boldsymbol{a}^1)V^*(\boldsymbol{s}^2) - V^*(\boldsymbol{s}^1)\right]$$

$$+ \mathbb{E}_{\boldsymbol{a}^1 \sim \pi(\boldsymbol{s}^1)}\left[\frac{2\Delta}{n(d-1)}\sum_{i=1}^{n}\sum_{p=1}^{d-1}\mathbb{1}\{\mathrm{sgn}(a_{i,p}^1) \neq \mathrm{sgn}(\theta_{i,p})\} \cdot V^*(\boldsymbol{s}^1)\right] \tag{38}$$

$$= \mathbb{E}_{\boldsymbol{a}^1 \sim \pi(\boldsymbol{s}^1)}\left[1 + \mathbb{P}(\boldsymbol{s}^1|\boldsymbol{s}^1, a^{1^*})V^*(\boldsymbol{s}^1) + \sum_{\boldsymbol{s}^2 \neq \boldsymbol{s}^1}\mathbb{P}(\boldsymbol{s}^2|\boldsymbol{s}^1, \boldsymbol{a}^1)V^*(\boldsymbol{s}^2) - V^*(\boldsymbol{s}^1)\right]$$

$$+ \mathbb{E}_{\boldsymbol{a}^1 \sim \pi(\boldsymbol{s}^1)}\left[\frac{2\Delta}{n(d-1)}\sum_{i=1}^{n}\sum_{p=1}^{d-1}\mathbb{1}\{\mathrm{sgn}(a_{i,p}^1) \neq \mathrm{sgn}(\theta_{i,p})\}V^*(\boldsymbol{s}^1)\right] \tag{39}$$

The last step follows from transition probability expression in Lemma 2. Now, we expand $V^*(\boldsymbol{s}^1)$ and simplify

$$\mathbb{E}_{\boldsymbol{a}^1 \sim \pi(\boldsymbol{s}^1)}[Q^*(\boldsymbol{s}^1, \boldsymbol{a}^1)] - V^*(\boldsymbol{s}^1)$$

$$= \mathbb{E}_{\boldsymbol{a}^1 \sim \pi(\boldsymbol{s}^1)}\left[1 + \mathbb{P}(\boldsymbol{s}^1|\boldsymbol{s}^1, \boldsymbol{a}^{1^*})V^*(\boldsymbol{s}^1) + \sum_{\boldsymbol{s}^2 \neq \boldsymbol{s}^1}\mathbb{P}(\boldsymbol{s}^2|\boldsymbol{s}^1, \boldsymbol{a}^1)V^*(\boldsymbol{s}^2) - \left(1 + \mathbb{P}(\boldsymbol{s}^1|\boldsymbol{s}^1, \boldsymbol{a}^{1^*})V^*(\boldsymbol{s}^1)\right.\right.$$

$$\left.\left. + \sum_{\boldsymbol{s}^2 \neq \boldsymbol{s}^1}\mathbb{P}(\boldsymbol{s}^2|\boldsymbol{s}^1, \boldsymbol{a}^{1^*})V^*(\boldsymbol{s}^2)\right)\right] + \mathbb{E}_{\boldsymbol{a}^1 \sim \pi(\boldsymbol{s}^1)}\left[\frac{2\Delta}{n(d-1)}\sum_{i=1}^{n}\sum_{p=1}^{d-1}\mathbb{1}\{\mathrm{sgn}(a_{i,p}^1) \neq \mathrm{sgn}(\theta_{i,p})\}V^*(\boldsymbol{s}^1)\right] \tag{40}$$

$$= \mathbb{E}_{\boldsymbol{a}^1 \sim \pi(\boldsymbol{s}^1)}\left[\sum_{\boldsymbol{s}^2 \neq \boldsymbol{s}^1}[\mathbb{P}(\boldsymbol{s}^2|\boldsymbol{s}^1, \boldsymbol{a}^1) - \mathbb{P}(\boldsymbol{s}^2|\boldsymbol{s}^1, \boldsymbol{a}^{1^*})]V^*(\boldsymbol{s}^2)\right]$$

$$+ \mathbb{E}_{\boldsymbol{a}^1 \sim \pi(\boldsymbol{s}^1)}\left[\frac{2\Delta}{n(d-1)}\sum_{i=1}^{n}\sum_{p=1}^{d-1}\mathbb{1}\{\mathrm{sgn}(a_{i,p}^1) \neq \mathrm{sgn}(\theta_{i,p})\}V^*(\boldsymbol{s}^1)\right]. \tag{41}$$

Next, we have the following lemma (proof in Appendix C.6)

**Lemma 5.** *For any of our instances,* $\forall (\boldsymbol{s}, \boldsymbol{a}) \in \mathcal{S} \times \mathcal{A}, \sum_{\boldsymbol{s}' \neq \boldsymbol{s}}\{\mathbb{P}(\boldsymbol{s}'|\boldsymbol{s}, \boldsymbol{a}) - \mathbb{P}(\boldsymbol{s}'|\boldsymbol{s}, \boldsymbol{a}^*)\}V^*(\boldsymbol{s}') \geq 0.$

Using the above lemma, we have the following bound:

$$\mathbb{E}_{\boldsymbol{a}^1\sim\pi(\boldsymbol{s}^1)}[Q^*(\boldsymbol{s}^1,\boldsymbol{a}^1)] - V^*(\boldsymbol{s}^1) \geq \mathbb{E}_{\boldsymbol{a}^1\sim\pi(\boldsymbol{s}^1)}\left[\frac{2\Delta}{n(d-1)}\sum_{i=1}^{n}\sum_{p=1}^{d-1}\mathbb{1}\{\mathrm{sgn}(a_{i,p}^1)\neq\mathrm{sgn}(\theta_{i,p})\}V^*(\boldsymbol{s}^1)\right]$$

(42)

Using Equations (30), (42) and (33), we have that

$$\mathbb{E}_{\theta,\pi}[R_1] = V^\pi(\boldsymbol{s}^1) - V^{\pi_\theta^*}(\boldsymbol{s}^1) \tag{43}$$

$$\geq \mathbb{E}_{\boldsymbol{a}^1\sim\,\pi(\boldsymbol{s}^1),\boldsymbol{s}^2\sim\,\mathbb{P}(.|\boldsymbol{s}^1,\boldsymbol{a}^1)}[V^\pi(\boldsymbol{s}^2) - V^*(\boldsymbol{s}^2)]$$

$$+ \mathbb{E}_{\boldsymbol{a}^1\sim\pi(\boldsymbol{s}^1)}\left[\frac{2\Delta}{n(d-1)}\sum_{i=1}^{n}\sum_{p=1}^{d-1}\mathbb{1}\{\boldsymbol{s}^1\neq goal\}\mathbb{1}\{\mathrm{sgn}(a_{i,p}^1)\neq\mathrm{sgn}(\theta_{i,p})\}V^*(\boldsymbol{s}^1)\right].$$

(44)

This introduces a recursive structure in the regret: the difference at time 1 is written in terms of the expectation of the same difference at time 2, plus another term. Although $\boldsymbol{s}^1$ is deterministic, $\boldsymbol{s}^2$ is stochastic due to transitions under $\pi$. Taking any sample path of length $H$ (i.e., global goal $\boldsymbol{g}$ is reached at step $H$), and unrolling the recursion across all steps (similar to the analysis between Equations (30) and (44)), we obtain

$$V^\pi(\boldsymbol{s}^1) - V^{\pi_\theta^*}(\boldsymbol{s}^1) \geq \mathbb{E}_1\left\{\sum_{h=1}^{H}\frac{2\Delta}{n(d-1)}\times\sum_{i\in\mathcal{I}_h}\sum_{p=1}^{d-1}\mathbb{1}\{\boldsymbol{s}^h\neq goal\}\mathbb{1}\{sgn(a_{i,p}^h)\neq sgn(\theta_{i,p})\}V^*(\boldsymbol{s}^h)\right\}$$

(45)

where $I_h$ represents the random set of all agents at node $s$ in global state $\boldsymbol{s}^h$ and where the expectation is taken with respect to trajectory induced in the first episode due to the interaction of $\pi$ and transition kernel $\mathbb{P}_\theta$. Note that in Equation (44), we have the summation over all agents in $[n]$ since $\mathcal{I}_1 = [n]$ as $\boldsymbol{s}^1 = \boldsymbol{s}_{\mathrm{init}}$ and, this $\mathcal{I}_h$ differs for different times based on sample paths. To fully convince why $\mathcal{I}_h$ occurs in the above expression, we encourage the reader to try carrying out the steps between Equations (30) and (44) for the second time step and then observe the pattern for arbitrary $h$.

Now, observe that as per our model, once an agent reaches the goal node, it remains there for the rest of the episode. For all $h > H$ in this episode, the global state is thought to be the goal state, by construction, making the indicator term $\mathbb{1}\{\boldsymbol{s}^h\neq goal\} = 0$. This allows us to safely extend the summation to infinity without altering its value. Specifically, we apply the following identity:

$$\mathbb{E}\left[\sum_{t=1}^{\tau}x_t\right] = \mathbb{E}\left[\sum_{t=1}^{\infty}x_t\mathbb{1}\{t\leq\tau\}\right]$$

(46)

where $\tau$ and $x_t$ are random variables. Using this, we get

$$V^\pi(\boldsymbol{s}^1) - V^{\pi_\theta^*}(\boldsymbol{s}^1) \geq \mathbb{E}_1\left\{\sum_{h=1}^{\infty}\frac{2\Delta}{n(d-1)}\sum_{i\in\mathcal{I}_h}\sum_{p=1}^{d-1}\mathbb{1}\{\boldsymbol{s}^h\neq goal\}\mathbb{1}\{sgn(a_{i,p}^h)\neq sgn(\theta_{i,p})\}V^*(\boldsymbol{s}^h)\right\}$$

(47)

Using the fact that $V^*(\boldsymbol{s}) \geq V_1^*$ for any $\boldsymbol{s} \in \mathcal{S}\backslash\boldsymbol{g}$, we have that

$$\mathbb{E}_{\theta,\pi}[R_1] \geq \frac{2\Delta V_1^*}{n(d-1)}\mathbb{E}_1\left[\sum_{h=1}^{\infty}\sum_{i\in\mathcal{I}_h}\sum_{p=1}^{d-1}\mathbb{1}\{\boldsymbol{s}^h\neq goal\}\mathbb{1}\{\mathrm{sgn}(a_{i,p}^h)\neq\mathrm{sgn}(\theta_{i,p})\}\right],$$

(48)

Now, the total expected regret over $K$ episodes is

$$\mathbb{E}_{\theta,\pi}[R(K)] \geq \frac{2\Delta V_1^*}{n(d-1)}\mathbb{E}_{\theta,\pi}\left[\sum_{t=1}^{\infty}\sum_{i\in\mathcal{I}_t}\sum_{p=1}^{d-1}\mathbb{1}\{\boldsymbol{s}^t\neq goal\}\mathbb{1}\{\mathrm{sgn}(a_{i,p}^t)\neq\mathrm{sgn}(\theta_{i,p})\}\right]$$

(49)

We get rid of the random set $\mathcal{I}_t$ by introducing an indicator as below, where $s_i^t$ denotes the $i^{th}$ index of state $\boldsymbol{s}^t$ i.e., node at which agent $i$ is present at $t$:

$$\mathbb{E}_{\theta,\pi}[R(K)] \geq \frac{2\Delta V_1^*}{n(d-1)} \mathbb{E}_{\theta,\pi}\left[\sum_{t=1}^{\infty}\sum_{i\in[n]}\mathbb{1}\{s_i^t \neq g\}\sum_{p=1}^{d-1}\mathbb{1}\{\boldsymbol{s}^t \neq goal\}\mathbb{1}\{\text{sgn}(a_{i,p}^t) \neq \text{sgn}(\theta_{i,p})\}\right] \tag{50}$$

$$= \frac{2\Delta V_1^*}{n(d-1)} \mathbb{E}_{\theta,\pi}\left[\sum_{t=1}^{\infty}\sum_{i\in[n]}\sum_{p=1}^{d-1}\mathbb{1}\{s_i^t \neq g\}\mathbb{1}\{\text{sgn}(a_{i,p}^t) \neq \text{sgn}(\theta_{i,p})\}\right] \tag{51}$$

where the simplification follows since $\mathbb{1}\{s_i^t \neq g\} \cdot \mathbb{1}\{\boldsymbol{s}^t \neq goal\} = \mathbb{1}\{s_i^t \neq g\}$ for any $i \in [n]$ and $t \in \mathbb{N}$.

For any $i \in [n]$, let $N_i$ be the random variable capturing total number of time steps over $K$ episodes in which agent $i$ stays at node $s$,

$$N_i = \sum_{t=1}^{\infty}\mathbb{1}\{s_i^t \neq g\} \tag{52}$$

Further, for any agent $i \in [n]$ and $j \in [d-1]$, let $N_{i,j}(\theta)$ be the (random) number of times in the entire duration of $K$ episodes that the agent $i$ chose $j^{th}$ component of its action as not the same as that of the optimal action for that instance while being at node $s$. Formally, it is given by

$$N_{i,j}(\theta) = \sum_{t=1}^{\infty}\mathbb{1}\{s_i^t \neq g\}\mathbb{1}\{\text{sgn}(a_{i,j}^t) \neq \text{sgn}(\theta_{i,j})\} \tag{53}$$

Using the above notations, we have Equation (51) rewritten as,

$$\mathbb{E}_{\theta,\pi}[R(K)] \geq \frac{2\Delta V_1^*}{n(d-1)} \mathbb{E}_{\theta,\pi}\left[\sum_{i\in[n]}\sum_{j=1}^{d-1}N_{i,j}(\theta)\right]. \tag{54}$$

To use the standard Pinsker's inequality, we need $N_{i,j}(\theta)$ to be bounded almost surely which doesn't hold (for example, in poorly designed or trivial algorithms) similar to the single agent SSP [14, 7]. Hence, we use the capping trick introduced by [7] and also used by [14]. In this trick, we cap the learning process to a predetermined $T$.

For any $i \in [n]$, we define the following:

$$N_i^- = \sum_{t=1}^{T}\mathbb{1}\{s_i^t \neq g\}, \tag{55}$$

and for any $i \in [n]$ and $j \in [d-1]$,

$$N_{i,j}^-(\theta) = \sum_{t=1}^{T}\mathbb{1}\{s_i^t \neq g\} \cdot \mathbb{1}\{\text{sgn}(a_{i,j}^t) \neq \text{sgn}(\theta_{i,j})\}. \tag{56}$$

Thus,

$$N^- = \max_{i\in[n]} N_i^- \tag{57}$$

denotes the truncated length over the $K$ episodes.

If the learning process (i.e. $K^{th}$ episode) ends before this $T$, the agents are taken to continue to remain in the *goal* state $\boldsymbol{g}$. In this case, the summations for the actual process and the capped process are the same. On the other hand, when $K$ episodes end beyond $T$, we have a possibly truncated sum. Due to the non-negativity of indicator functions, the summations for the capped process lower bounds the actual sum.

For any $i \in [n]$, we observe that (since any component of action, which is either $+1$ or $-1$, has to match corresponding sign of $\theta$ or $\theta^j$) for any $j \in [d-1]$,

$$N_i^- = N_{i,j}^-(\theta) + N_{i,j}^-(\theta^j) \tag{58}$$

where for every $\theta \in \Theta$, $\theta^j \in \Theta$ for $j \in [d-1]$ represents the vector that only differs from $\theta$ only at the $j^{th}$ index of every $\theta_i$ in $\theta = \{\theta_1, 1, \theta_2, 1, \ldots, \theta_n, 1\}$, where each $\theta_i \in \{\frac{-\Delta}{n(d-1)}, \frac{\Delta}{n(d-1)}\}^{d-1}$. Note that since any component can take only one of the two values $\left\{\frac{\Delta}{n(d-1)}, -\frac{\Delta}{n(d-1)}\right\}$, the differing components are exactly negative of each other. Thus, we can further lower bound Equation (54) as:

$$\mathbb{E}_{\theta,\pi}[R(K)] \geq \frac{2\Delta V_1^*}{n(d-1)} \mathbb{E}_{\theta,\pi}\left[\sum_{i \in [n]} \sum_{j=1}^{d-1} N_{i,j}^-(\theta)\right] \tag{59}$$

Notice that for any $j \in [d-1]$, $\theta \to \theta^j$ map is bijective. This allows us to write the following

$$2\sum_{\theta \in \Theta} \mathbb{E}_{\theta,\pi}[R(K)] \geq \frac{2\Delta}{n(d-1)} \sum_{\theta \in \Theta} V_1^* \sum_{i \in [n]} \sum_{j=1}^{d-1} \left[\mathbb{E}_{\theta,\pi}[N_{i,j}^-(\theta)] + \mathbb{E}_{\theta^j,\pi}[N_{i,j}^-(\theta^j)]\right] \tag{60}$$

$$= \frac{2\Delta V_1^*}{n(d-1)} \sum_{\theta \in \Theta} \sum_{i \in [n]} \sum_{j=1}^{d-1} \left[\mathbb{E}_{\theta,\pi}[N_{i,j}^-(\theta)] + \mathbb{E}_{\theta^j,\pi}[N_i^- - N_{i,j}^-(\theta)]\right] \tag{61}$$

$$= \frac{2\Delta V_1^*}{n(d-1)} \sum_{\theta \in \Theta} \sum_{i \in [n]} \sum_{j=1}^{d-1} \left[\mathbb{E}_{\theta,\pi}[N_i^-] + \mathbb{E}_{\theta,\pi}[N_{i,j}^-(\theta)] - \mathbb{E}_{\theta^j,\pi}[N_{i,j}^-(\theta)]\right] \tag{62}$$

The Equation (61) follows from Equation (58) and noticing that $V_1^* (= \frac{1}{1-p_{1,1}^*})$ only depends on $(n, \delta, \Delta)$ i.e., it is the same for all $\theta \in \Theta$ in this analysis. Also, Equation (62) is due to the fact that for any $j \in [d-1]$, for any function $f(\cdot)$, we have $\sum_{\theta \in \Theta} f(\theta^j) = \sum_{\theta \in \Theta} f(\theta)$. To deal with the first expectation in the above equation we use the following lemma (proof in Appendix C.7)

**Lemma 6.** *For any instance $(n, \delta, \Delta, \theta)$, algorithm $\pi$, and $T \geq 2KV_1^*$, we have $\mathbb{E}_{\theta,\pi}[N_i^-] \geq \frac{KV_1^*}{4}$ for any $i \in [n]$.*

For the remaining terms, we use Lemma E.3 from [14] which is a standard result using Pinskers' inequality. We re-iterate the lemma here.

**Lemma E.3 in [14].** *If $f : S^T \to [0, D]$ then, $\mathbb{E}_{\mathbb{P}_1}(f(s)) - \mathbb{E}_{\mathbb{P}_2}(f(s)) \leq D\sqrt{\frac{log(2)}{2}} \cdot \sqrt{KL(\mathbb{P}_2||\mathbb{P}_1)}$.*

Using Lemma 6 and Lemma E.3 from [14] in Equation (62), we obtain

$$2\sum_{\theta \in \Theta} \mathbb{E}_{\theta,\pi}[R(K)] \geq \frac{2\Delta V_1^*}{n(d-1)} \sum_{\theta \in \Theta} \sum_{i \in [n]} \sum_{j=1}^{d-1} \left[KV_1^*/4 - \{T\sqrt{log(2)/2}\sqrt{KL(\mathbb{P}_\theta^\pi||\mathbb{P}_{\theta^j}^\pi)}\}\right] \tag{63}$$

where $\mathbb{P}_\theta^\pi$ and $\mathbb{P}_{\theta^j}^\pi$ are the distributions over T-length state sample paths generated due to the interaction between $\pi$ and the transition kernels $\mathbb{P}_\theta$ and $\mathbb{P}_{\theta^j}$ respectively. Next, to bound the KL-divergence term appearing in Equation (63), we have the following lemma (proof in Appendix C.8)

**Lemma 7.** *For any instance $(n, \delta, \Delta, \theta)$ and algorithm $\pi$, for any $j \in [d-1]$, we have,*

$$KL(\mathbb{P}_\theta^\pi||\mathbb{P}_{\theta^j}^\pi) \leq 3 \cdot 2^{2n} \cdot \frac{\Delta^2}{\delta(d-1)^2} \cdot \mathbb{E}_{\theta,\pi}[N^-]. \tag{64}$$

Using above, we get the following

$$2\sum_{\theta \in \Theta} \mathbb{E}_{\theta,\pi}[R(K)] \geq \frac{2\Delta V_1^*}{n(d-1)} \sum_{\theta \in \Theta} \sum_{i \in [n]} \sum_{j=1}^{d-1} \left[KV_1^*/4 - \left(\frac{T}{\sqrt{2}}\sqrt{3.2^{2n}\frac{\Delta^2}{\delta(d-1)^2}\mathbb{E}_{\theta,\pi}[N^-]}\right)\right] \tag{65}$$

$$= \frac{2\Delta V_1^*}{n(d-1)} \sum_{\theta \in \Theta} \sum_{i \in [n]} \sum_{j=1}^{d-1} \left[ KV_1^*/4 - \left\{ T\sqrt{3/2} \frac{2^n.\Delta}{\sqrt{\delta}(d-1)} \cdot \sqrt{\mathbb{E}_{\theta,\pi}[N^-]} \right\} \right] \tag{66}$$

$$\geq \frac{2\Delta V_1^*}{n(d-1)} \sum_{\theta \in \Theta} \sum_{i \in [n]} \sum_{j=1}^{d-1} \left[ KV_1^*/4 - \left\{ T\sqrt{3/2} \frac{2^n.\Delta}{\sqrt{\delta}(d-1)} \cdot \sqrt{T} \right\} \right] \tag{67}$$

$$= \frac{2\Delta V_1^*}{n(d-1)} \sum_{\theta \in \Theta} \sum_{i \in [n]} \sum_{j=1}^{d-1} \left[ KV_1^*/4 - (2KV_1^*)^{3/2}\sqrt{3/2} \frac{2^n.\Delta}{\sqrt{\delta}(d-1)} \right] \tag{68}$$

where in the last second step we used the fact that $N^- \leq T$ and in the last step, we use $T = 2KV_1^*$. Further, notice that the term inside the bracket is independent of $\theta, i$ and $j$. Thus, we have

$$\sum_{\theta \in \Theta} \mathbb{E}_{\theta,\pi}[R(K)] \geq \Delta V_1^* \cdot |\Theta| \cdot \left[ KV_1^*/4 - (2KV_1^*)^{3/2}\sqrt{3/2} \frac{2^n.\Delta}{\sqrt{\delta}(d-1)} \right] \tag{69}$$

Rearranging and using the fact that $\sqrt{3} < 2$ we get

$$\frac{1}{|\Theta|} \sum_{\theta \in \Theta} \mathbb{E}_{\theta,\pi}[R(K)] \geq V_1^* \left[ \Delta KV_1^*/4 - (KV_1^*)^{3/2} \frac{2^{n+2}.\Delta^2}{\sqrt{\delta}(d-1)} \right]. \tag{70}$$

Now choosing

$$\Delta = \frac{(d-1)\sqrt{\delta}}{2^{n+5}\sqrt{KV_1^*}}, \tag{71}$$

we have that

$$\frac{1}{|\Theta|} \sum_{\theta \in \Theta} \mathbb{E}_{\theta,\pi}[R(K)] \geq V_1^* \left[ \frac{(d-1)\sqrt{\delta}}{2^{n+5}\sqrt{KV_1^*}} \cdot \frac{KV_1^*}{4} - \frac{(KV_1^*)^{3/2} \cdot 2^{n+2}}{\sqrt{\delta}(d-1)} \cdot \left\{ \frac{(d-1)\sqrt{\delta}}{2^{n+5}\sqrt{KV_1^*}} \right\}^2 \right] \tag{72}$$

$$= V_1^* \left[ \frac{(d-1)\sqrt{\delta \cdot K \cdot V_1^*}}{2^{n+7}} - \frac{(d-1)\sqrt{\delta \cdot K \cdot V_1^*}}{2^{n+8}} \right] \tag{73}$$

$$\geq \frac{(d-1) \cdot \sqrt{\delta} \cdot \sqrt{KV_1^*}}{2^{n+8}} \tag{74}$$

$$\geq \frac{d \cdot \sqrt{\delta} \cdot \sqrt{KV_1^*}}{2^{n+9}} \tag{75}$$

$$\geq \frac{d \cdot \sqrt{\delta} \cdot \sqrt{KB^*/n}}{2^{n+9}}, \tag{76}$$

where in the last second line, we use that $V_1^* > 1$ and in the last line, we use $d-1 \geq d/2$ for any natural number $d \geq 2$ which holds for our instances. The final step follows from the result $V_1^* > B^*/n$ (details in Appendix C.11). Since the average over entire set is lower bounded by the RHS, there must exist some parameter in the set for which the RHS is definitely a valid lower bound.

Now, what remains to check is that the $\Delta$ chosen above is valid. Hence, we need to ensure that

$$\Delta = \frac{(d-1)\sqrt{\delta}}{2^{n+5}\sqrt{KV_1^*}} < 2^{-n} \cdot \frac{1-2\delta}{1+n+n^2}. \tag{77}$$

Thus, we need

$$K > \frac{(d-1)^2 \cdot \delta}{2^{10} \cdot V_1^* \cdot \left( \frac{1-2\delta}{1+n+n^2} \right)^2}. \tag{78}$$

The condition in the theorem ensures the above because $V_1^* > B^*/n$

$$K > \frac{n \cdot (d-1)^2 \cdot \delta}{2^{10} \cdot B^* \cdot \left( \frac{1-2\delta}{1+n+n^2} \right)^2} > \frac{(d-1)^2 \cdot \delta}{2^{10} \cdot V_1^* \cdot \left( \frac{1-2\delta}{1+n+n^2} \right)^2}. \tag{79}$$

So the lower bound is valid and it gives a bound on the number of episodes as well. This completes the proof. $\square$

# C  Proof of Intermediate Lemmas

## C.1  Proof of Lemma 1

**Lemma 1.** *For any instance $(n, \delta, \Delta, \theta)$, features defined in Equations (3) and (4) yield valid transition probabilities, i.e.,*

1. *We have, $\langle \phi(s'|s, a), \theta \rangle \geq 0 \; \forall \; (s, a, s')$, and $\sum_{s' \in \mathcal{S}} \langle \phi(s'|s, a), \theta \rangle = 1 \; \forall \; (s, a) \in \mathcal{S} \times \mathcal{A}$.*

2. *For any $(s, a) \in \mathcal{S} \times \mathcal{A}$, if transition to some $s' \in \mathcal{S}$ is impossible, we have $\langle \phi(s'|s, a), \theta \rangle = 0$; on the other hand, if the transition is certain, then $\langle \phi(s'|s, a), \theta \rangle = 1$.*

*Proof.* For any instance $(n, \delta, \Delta, \theta)$, we prove the lemma through partitions of the set of all global transitions $(s, a, s') \in \mathcal{S} \times \mathcal{A} \times \mathcal{S}$.

Fix any action $a \in \mathcal{A}$. We consider various transitions and argue that the analysis below for this fixed action holds for any action.

**When $s = g$**

All the agents are at the goal node. Using the features, we have, for $s' = g$ (this corresponds to third case in Equation (3)),

$$
\begin{aligned}
\mathbb{P}(g|g, a) &= \langle \phi(g|g, a), \theta \rangle \\
&= \langle (\mathbf{0}_{nd-1}, 1), (\theta_1, 1, \ldots, \theta_n, 1) \rangle \\
&= 1
\end{aligned}
\tag{80}
$$

Again, using the features, for any $s' \neq g$ (this corresponds to the second case in Equation (3)), we have,

$$
\begin{aligned}
\mathbb{P}(s'|g, a) &= \langle \phi(s'|g, a), \theta \rangle \\
&= \langle \mathbf{0}_{nd}, (\theta_1, 1, \ldots, \theta_n, 1) \rangle \\
&= 0
\end{aligned}
\tag{81}
$$

Thus, for the fixed $a$, for any $s' \in \mathcal{S}$, from above Equations (81) and (80), we have

$$
\langle \phi(s'|g, a), \theta \rangle \geq 0.
$$

Also using Equations (81) and (80) we have

$$
\sum_{s' \in \mathcal{S}} \langle \phi(s'|g, a), \theta \rangle = \langle \phi(g|g, a), \theta \rangle + \sum_{s' \in \mathcal{S} \setminus \{g\}} \langle \phi(s'|g, a), \theta \rangle = 1 + 0 = 1
\tag{82}
$$

Hence, the first part of the lemma is proved for this case.

Now, recall from the construction of instances as given in Section 3.1, the impossible transitions in this case are from $g$ to any $s' \neq g$ and the only certain transition is from $g$ to $g$. Hence, Equations (80) and (81) ensure that the second part of the lemma is also satisfied for this case.

**When $s \neq g$**

These represent the most general transitions. For any such $s$, let $\mathcal{I} = \{i_1, i_2, \ldots, i_r\}$ be the set of all agents that are at node $s$ in state $s$. Let $\mathcal{J} = \{j_1, j_2, \ldots, j_{n-r}\}$ be the set of all agents that are at node $g$ in state $s$. We partition the state space here based on $s$.

Consider the set of all states in $\mathcal{S}$ in which at least one of the agents in $\mathcal{J}$ is not at node $g$. We represent this set as $\bar{\mathcal{S}}(s)$. Recall that these are the states not *reachable* from $s$. While all the other states that are *reachable* from $s$ are denoted by $\mathcal{S}(s)$.

Using the features, for any $s' \in \bar{\mathcal{S}}(s)$ (this corresponds to the second part of the second case in Equation (3)), we have

$$
\begin{aligned}
\mathbb{P}(s'|s, a) &= \langle \phi(s'|s, a), \theta \rangle \\
&= \langle \mathbf{0}_{nd}, (\theta_1, 1, \ldots, \theta_n, 1) \rangle \\
&= 0
\end{aligned}
\tag{83}
$$

These are the only impossible transitions and there's no certain transition for $s$. Thus second part of the lemma holds.

For any $s' \in \mathcal{S}(s)$, we derive the expression of transition probability in closed form in Appendix C.2. Here, we will just show that second condition of the lemma is also satisfied.

From Equation (83) we have

$$\sum_{s' \in \mathcal{S}} \mathbb{P}(s'|s, a) = \sum_{s' \in \mathcal{S}(s)} \mathbb{P}(s'|s, a) + \sum_{s' \in \bar{\mathcal{S}}(s)} \mathbb{P}(s'|s, a) = \sum_{s' \in \mathcal{S}(s)} \mathbb{P}(s'|s, a) \tag{84}$$

Using Assumption 1 and features in Equation (3), we have

$$\sum_{s' \in \mathcal{S}(s)} \mathbb{P}(s' \mid s, a) = \sum_{s' \in \mathcal{S}(s)} \langle \phi(s' \mid s, a), (\theta_1, 1, \ldots, \theta_n, 1) \rangle \tag{85}$$

$$= \sum_{s' \in \mathcal{S}(s)} \langle \phi((s'_1, \ldots, s'_n)|(s_1, \ldots, s_n), (a_1, \ldots, a_n)), (\theta_1, 1, \ldots, \theta_n, 1) \rangle \tag{86}$$

$$= \sum_{s' \in \mathcal{S}(s)} \langle ((\phi_1(s'_1 \mid s_1, a_1), \ldots, \phi_n(s'_n \mid s_n, a_n)), (\theta_1, 1, \ldots, \theta_n, 1) \rangle \tag{87}$$

$$= \sum_{s' \in \mathcal{S}(s)} \sum_{i=1}^{n} \langle \phi_i(s'_i \mid s_i, a_i), (\theta_i, 1) \rangle, \tag{88}$$

In the above, Equation (86) is obtained by explicitly expanding the joint state, action, and next-state tuples into their individual components. In Equation (87) we further break down the joint feature vector into a concatenation of per-agent feature vectors $\phi_i(s'_i \mid s_i, a_i)$ according to features corresponding to first case in Equation (3). Finally, in Equation (88) we use linearity of the inner product.

In the next step, we split the summation into two disjoint sets of agents, based on the nodes they are currently located at. This helps us exploit the properties of nested sums for further simplification.

$$\sum_{s' \in \mathcal{S}(s)} \mathbb{P}(s' \mid s, a) = \sum_{s' \in \mathcal{S}(s)} \sum_{i=1}^{n} \langle \phi_i(s'_i \mid s_i, a_i), (\theta_i, 1) \rangle \tag{89}$$

$$= \sum_{s' \in \mathcal{S}(s)} \left( \sum_{i \in \mathcal{I}} \langle \phi_i(s'_i \mid s_i, a_i), (\theta_i, 1) \rangle + \sum_{j \in \mathcal{J}} \langle \phi_j(s'_j \mid s_j, a_j), (\theta_j, 1) \rangle \right) \tag{90}$$

$$= \sum_{i \in \mathcal{I}} \sum_{s' \in \mathcal{S}(s)} \langle \phi_i(s'_i \mid s_i, a_i), (\theta_i, 1) \rangle + \sum_{j \in \mathcal{J}} \sum_{s' \in \mathcal{S}(s)} \langle \phi_j(s'_j \mid s_j, a_j), (\theta_j, 1) \rangle \tag{91}$$

$$= \sum_{i \in \mathcal{I}} \left[ \sum_{s'_1 \in \mathcal{S}'_1} \cdots \sum_{s'_n \in \mathcal{S}'_n} \langle \phi_i(s'_i \mid s_i, a_i), (\theta_i, 1) \rangle \right]$$

$$+ \sum_{j \in \mathcal{J}} \left[ \sum_{s'_1 \in \mathcal{S}'_1} \cdots \sum_{s'_n \in \mathcal{S}'_n} \langle \phi_j(s'_j \mid s_j, a_j), (\theta_j, 1) \rangle \right] \tag{92}$$

where in the Equation (92) we fully unroll the nested sum, making it explicit that we are summing over the Cartesian product of each agent's next state space ($\mathcal{S}'_i = \{s, g\}$ if $i \in \mathcal{I}$ and $\mathcal{S}'_i = \{g\}$ if $i \in \mathcal{J}$). It emphasizes that even though the local inner product depends only on a particular agent fixed in the outer sum, it is repeated across all combinations of other agents' next states.

Since for any $j \in \mathcal{J}$, we have $s'_j \equiv g$, the transition is certain. Moreover, $|\mathcal{I}| = r$, and also note that for each fixed $i \in \mathcal{I}$, the term $\langle \phi(s'_i \mid s_i, a_i), (\theta_i, 1) \rangle$ does not depend on the values of other agents' next states in $\mathcal{I}$. Similarly, for each $j \in \mathcal{J}$, the corresponding inner product term is independent of the next states of any agents in $\mathcal{I}$. This allows us to factor out these terms from the inner sums. We now refine the expression from Equation (92) by leveraging this structure as follows:

$$\sum_{s' \in \mathcal{S}(s)} \mathbb{P}(s' \mid s, a) = \sum_{i \in \mathcal{I}} \{ 2^{r-1} \cdot \sum_{s'_i} \langle \phi(s'_i \mid s_i, a_i), (\theta_i, 1) \rangle \} + \sum_{j \in \mathcal{J}} \{ 2^r \cdot \langle \phi(s'_j \mid s_j, a_j), (\theta_j, 1) \rangle \}$$

$$\tag{93}$$

Now, substituting the feature representations defined in Equation (4), we have

$$
\sum_{s' \in \mathcal{S}(s)} \mathbb{P}(s' \mid s, a) = \sum_{i \in \mathcal{I}} \left( 2^{r-1} \cdot \left\langle (0_{d-1}, \frac{1}{n \cdot 2^{r-1}}), (\theta_i, 1) \right\rangle \right)
$$

$$
+ \sum_{j \in \mathcal{J}} \left( 2^r \cdot \left\langle (0_{d-1}, \frac{1}{n \cdot 2^r}), (\theta_j, 1) \right\rangle \right)
$$

$$
= \sum_{i \in \mathcal{I}} \frac{1}{n} + \sum_{j \in \mathcal{J}} \frac{1}{n}
$$

$$
= \frac{r}{n} + \frac{n-r}{n}
$$

$$
= 1
$$

Now, what remains is to verify that all remaining individual transition probabilities are all non-negative i.e., for all $s' \in \mathcal{S}(s)$, we have

$$
\mathbb{P}(s' \mid s, a) \geq 0. \tag{94}
$$

Recall for any of our instance, $n \geq 1$, $\delta \in (2/5, 1/2)$, $\Delta < 2^{-n} \cdot \frac{1-2\delta}{1+n+n^2}$ and $\theta \in \Theta$. Suppose $s$ is of type $r \in [n]$. For any $r' \in [r] \cup \{0\}$, we analyze the non-negativity of the transition probability as follows.

For any $s' \in \mathcal{S}_{r'}(s)$, we let $\mathcal{T} = \{t_1, t_2, \ldots, t_{r-r'}\} \subseteq \mathcal{I}$ represent the set of agents that transit from node $s$ to node $g$ during the global transition $s$ to $s'$. Those who continue to stay at node $s$ constitute the set $\mathcal{I} \cap \mathcal{T}'$.
First, observe that

$$
\Delta < \frac{2^{-n}(1-2\delta)}{1+n+n^2} < \frac{2^{-n} \cdot 2\delta}{3}, \tag{95}
$$

where the second inequality holds because $1 - 2\delta < 2\delta$ for $\delta \in (2/5, 1/2)$ and $1 + n + n^2 \geq 3$ for all $n \geq 1$.

Next, using $r \leq n$, we get

$$
\delta > 3 \cdot 2^{n-1} \cdot \Delta > 3 \cdot 2^{r-1} \cdot \Delta \quad \implies \quad \frac{\delta}{2^{r-1}} - 3\Delta > 0. \tag{96}
$$

Observe that the above inequality is equivalent to

$$
\frac{(-0.5n)(1-2\delta) + 0.5n}{n \cdot 2^{r-1}} - 3\Delta > 0, \tag{97}
$$

Moreover, since $0 \leq r'$ and $r \leq n$, and $\delta \in (2/5, 1/2)$, it follows that $-0.5r \geq -0.5n$, and hence

$$
\frac{(r' - 0.5r)(1-2\delta) + 0.5n}{n \cdot 2^{r-1}} - 3\Delta > 0. \tag{98}
$$

Expanding and rearranging, we have

$$
\frac{r'(1-2\delta) + r(\delta - 0.5) + 0.5n}{n \cdot 2^{r-1}} - 3\Delta > 0. \tag{99}
$$

Since $r' \leq r \leq n$, we note that $\frac{\Delta}{n}(r - 2r') \geq -\Delta$. Hence, we have

$$
\frac{r' + (r - 2r')\delta}{n \cdot 2^{r-1}} + \frac{n-r}{n \cdot 2^r} + \frac{\Delta}{n}(r - 2r') - 2\Delta > 0. \tag{100}
$$

Now, note that each indicator $\mathbb{1}\{\mathrm{sgn}(a_{i,p}) \neq \mathrm{sgn}(\theta_{i,p})\}$ takes values in $0, 1$, so

$$
\sum_{i \in \mathcal{I} \cap \mathcal{T}'} \mathbb{1}\{\mathrm{sgn}(a_{i,p}) \neq \mathrm{sgn}(\theta_{i,p})\} - \sum_{t \in \mathcal{T}} \mathbb{1}\{\mathrm{sgn}(a_{t,p}) \neq \mathrm{sgn}(\theta_{t,p})\} \geq -(r - r') \tag{101}
$$

$$
\geq -r \tag{102}
$$

$$
\geq -n \tag{103}
$$

Using above results (Equations (100) and (103)) and the expression for general transition probability (from Lemma 2) we obtain

$$\mathbb{P}(s'|s,a) = \frac{r' + (r - 2r')\delta}{n \cdot 2^{r-1}} + \frac{n - r}{n \cdot 2^r} + \frac{\Delta}{n}(r - 2r')$$

$$+ \frac{2\Delta}{n(d-1)} \sum_{p=1}^{d-1} \left[ \sum_{i \in \mathcal{I} \cap \mathcal{T}'} \mathbb{1}\{\text{sgn}(a_{i,p}) \neq \text{sgn}(\theta_{i,p})\} - \sum_{t \in \mathcal{T}} \mathbb{1}\{\text{sgn}(a_{t,p}) \neq \text{sgn}(\theta_{t,p})\} \right] \tag{104}$$

$$> 0. \tag{105}$$

This shows that the constructed probability remains strictly positive for all feasible transitions. Thus, all transition probabilities $\mathbb{P}(s' \mid s, a)$ are non-negative under the proposed construction. Thus the first condition is also satisfied.

This completes the proof that the transition model and proposed feature design defines a valid probability distribution on any instance as defined in Section 3.1. $\qquad \square$

### C.2  Proof of Lemma 2

**Lemma 2.** *For a given state $s \in \mathcal{S} \setminus g$, let $\mathcal{I}$ denote the set of agents at node $s$. Consider a transition to $s' \in \mathcal{S}(s)$ under joint action $a \in \mathcal{A}$. Let $\mathcal{T} \subseteq \mathcal{I}$ be the set of agents moving from $s$ to $g$. Define $r = |\mathcal{I}|$, $r' = |\mathcal{I} \setminus \mathcal{T}|$ and $\mathcal{T}' = \mathcal{N} \setminus \mathcal{T}$. Then, under our proposed feature construction and Assumption 1 on the instances, we have*

$$\mathbb{P}(s'|s,a) = \frac{r' + (r - 2r')\delta}{n \cdot 2^{r-1}} + \frac{n - r}{n \cdot 2^r} + \frac{\Delta}{n}(r - 2r')$$

$$+ \frac{2\Delta}{n(d-1)} \sum_{p=1}^{d-1} \left[ \sum_{i \in \mathcal{I} \cap \mathcal{T}'} \mathbb{1}\{sgn(a_{i,p}) \neq sgn(\theta_{i,p})\} - \sum_{t \in \mathcal{T}} \mathbb{1}\{sgn(a_{t,p}) \neq sgn(\theta_{t,p})\} \right].$$

*Proof.* Consider any state $s \in \mathcal{S} \setminus \{g\}$ such that the agents in $\mathcal{I} := \{i_1, i_2, \ldots, i_r\}$ are at node $s$, and the remaining agents in $\mathcal{J} := \{j_1, j_2, \ldots, j_{n-r}\}$ are already at the node $g$. For any next state $s' \in \mathcal{S}(s)$, let $\mathcal{T} = \{t_1, t_2, \ldots, t_{r-r'}\} \subseteq \mathcal{I}$ be the set of agents transiting from node $s$ to node $g$ in the next state. Others in $\mathcal{I}$ that stay at node $s$ form the set $\mathcal{I} \cap \mathcal{T}'$. Then, $s \in \mathcal{S}_r$ and $s' \in \mathcal{S}_{r'}$, where $n \geq r \geq r' \geq 0$ and $n \geq 1$.

From Assumption 1, the transition probability is given by

$$\mathbb{P}(s'|s,a) = \langle \phi(s'|s,a), \theta \rangle \tag{106}$$

$$= \langle \phi\left((s_1', s_2', \ldots, s_n')|(s_1, s_2, \ldots, s_n), (a_1, a_2, \ldots, a_n)\right), (\theta_1, 1, \theta_2, 1, \ldots, \theta_n, 1) \rangle \tag{107}$$

$$= \sum_{i \in \mathcal{I} \cap \mathcal{T}'} \langle \phi_i(s \mid s, a_i), (\theta_i, 1) \rangle + \sum_{j \in \mathcal{J}} \langle \phi_j(g \mid g, a_j), (\theta_j, 1) \rangle + \sum_{t \in \mathcal{T}} \langle \phi_t(g \mid s, a_t), (\theta_t, 1) \rangle, \tag{108}$$

where the last inequality follows by decomposing the agents across sets $\mathcal{I} \cap \mathcal{T}'$, $\mathcal{J}$, and $\mathcal{T}$.

Now, using the feature definition from Equation (4), we compute the transition probability as follows:

$$\mathbb{P}(s'|s,a) = \sum_{i \in \mathcal{I} \cap \mathcal{T}'} \left\langle \left(-a_i, \frac{1-\delta}{n \cdot 2^{r-1}}\right), (\theta_i, 1) \right\rangle + \sum_{j \in \mathcal{J}} \left\langle \left(0_{d-1}, \frac{1}{n \cdot 2^r}\right), (\theta_j, 1) \right\rangle \tag{109}$$

$$+ \sum_{t \in \mathcal{T}} \left\langle \left(a_t, \frac{\delta}{n \cdot 2^{r-1}}\right), (\theta_t, 1) \right\rangle$$

$$= \sum_{i \in \mathcal{I} \cap \mathcal{T}'} \left(\frac{1-\delta}{n \cdot 2^{r-1}} + \langle -a_i, \theta_i \rangle\right) + \sum_{j \in \mathcal{J}} \frac{1}{n \cdot 2^r} + \sum_{t \in \mathcal{T}} \left(\langle a_t, \theta_t \rangle + \frac{\delta}{n \cdot 2^{r-1}}\right) \tag{110}$$

$$= r' \cdot \frac{1-\delta}{n \cdot 2^{r-1}} + \sum_{i \in \mathcal{I} \cap \mathcal{T}'} \langle -a_i, \theta_i \rangle + \frac{n-r}{n \cdot 2^r} + (r - r') \cdot \frac{\delta}{n \cdot 2^{r-1}} + \sum_{t \in \mathcal{T}} \langle a_t, \theta_t \rangle \tag{111}$$

$$= \frac{r' + (r - 2r')\,\delta}{n \cdot 2^{r-1}} + \sum_{i \in \mathcal{I} \cap \mathcal{T}'} \langle -a_i, \theta_i \rangle + \frac{n - r}{n \cdot 2^r} + \sum_{t \in \mathcal{T}} \langle a_t, \theta_t \rangle \tag{112}$$

$$= \frac{r' + (r - 2r')\delta}{n \cdot 2^{r-1}} + \frac{n - r}{n \cdot 2^r} - \sum_{i \in \mathcal{I} \cap \mathcal{T}'} \sum_{p=1}^{d-1} a_{i,p} \cdot \theta_{i,p} + \sum_{t \in \mathcal{T}} \sum_{p=1}^{d-1} a_{t,p} \cdot \theta_{t,p} \tag{113}$$

$$= \frac{r' + (r - 2r')\delta}{n \cdot 2^{r-1}} + \frac{n - r}{n \cdot 2^r}$$
$$- \sum_{i \in \mathcal{I} \cap \mathcal{T}'} \sum_{p=1}^{d-1} \frac{\Delta}{n(d-1)} \Big( \mathbb{1}\{\mathrm{sgn}(a_{i,p}) = \mathrm{sgn}(\theta_{i,p})\} - \mathbb{1}\{\mathrm{sgn}(a_{i,p}) \neq \mathrm{sgn}(\theta_{i,p})\} \Big)$$
$$+ \sum_{t \in \mathcal{T}} \sum_{p=1}^{d-1} \frac{\Delta}{n(d-1)} \Big( \mathbb{1}\{\mathrm{sgn}(a_{t,p}) = \mathrm{sgn}(\theta_{t,p})\} - \mathbb{1}\{\mathrm{sgn}(a_{t,p}) \neq \mathrm{sgn}(\theta_{t,p})\} \Big), \tag{114}$$

where in Equation (111) we use the cardinality of sets $\mathcal{I} \cap \mathcal{T}'$, $\mathcal{J}$ and $\mathcal{I}$. Equation (114) follows from the fact that for any $i \in [n]$, $a_i \in \{-1, 1\}^{d-1}$, and $\theta_i \in \left\{ \frac{-\Delta}{n(d-1)}, \frac{\Delta}{n(d-1)} \right\}^{d-1}$.

Taking the summation over $p$ inside,

$$\mathbb{P}(s'|s, a) = \frac{r' + (r - 2r')\delta}{n \cdot 2^{r-1}} + \frac{n - r}{n \cdot 2^r}$$
$$- \sum_{i \in \mathcal{I} \cap \mathcal{T}'} \frac{\Delta}{n(d-1)} \Big\{ d - 1 - 2 \sum_{p=1}^{d-1} \mathbb{1}\{\mathrm{sgn}(a_{i,p}) \neq \mathrm{sgn}(\theta_{i,p})\} \Big\}$$
$$+ \sum_{t \in \mathcal{T}} \frac{\Delta}{n(d-1)} \Big\{ d - 1 - 2 \sum_{p=1}^{d-1} \mathbb{1}\{\mathrm{sgn}(a_{t,p}) \neq \mathrm{sgn}(\theta_{t,p})\} \Big\} \tag{115}$$

$$= \frac{r' + (r - 2r')\delta}{n \cdot 2^{r-1}} + \frac{n - r}{n \cdot 2^r}$$
$$- \frac{\Delta}{n(d-1)} \Big\{ r'(d-1) - 2 \sum_{i \in \mathcal{I} \cap \mathcal{T}'} \sum_{p=1}^{d-1} \mathbb{1}\{\mathrm{sgn}(a_{i,p}) \neq \mathrm{sgn}(\theta_{i,p})\} \Big\}$$
$$+ \frac{\Delta}{n(d-1)} \Big\{ (r - r') \cdot (d-1) - 2 \sum_{t \in \mathcal{T}} \sum_{p=1}^{d-1} \mathbb{1}\{\mathrm{sgn}(a_{t,p}) \neq \mathrm{sgn}(\theta_{t,p})\} \Big\} \tag{116}$$

$$= \frac{r' + (r - 2r')\delta}{n \cdot 2^{r-1}} + \frac{n - r}{n \cdot 2^r}$$
$$+ \frac{\Delta}{n(d-1)} \Big\{ (r - 2r') \cdot (d-1) \Big\} + \frac{2\Delta}{n(d-1)} \cdot \sum_{i \in \mathcal{I} \cap \mathcal{T}'} \sum_{p=1}^{d-1} \mathbb{1}\{\mathrm{sgn}(a_{i,p}) \neq \mathrm{sgn}(\theta_{i,p})\}$$
$$- \frac{2\Delta}{n(d-1)} \cdot \sum_{t \in \mathcal{T}} \sum_{p=1}^{d-1} \mathbb{1}\{\mathrm{sgn}(a_{t,p}) \neq \mathrm{sgn}(\theta_{t,p})\} \Big\} \tag{117}$$

$$= \frac{r' + (r - 2r')\delta}{n \cdot 2^{r-1}} + \frac{n - r}{n \cdot 2^r} + \frac{\Delta}{n}(r - 2r')$$
$$+ \frac{2\Delta}{n(d-1)} \sum_{p=1}^{d-1} \Big\{ \sum_{i \in \mathcal{I} \cap \mathcal{T}'} \mathbb{1}\{\mathrm{sgn}(a_{i,p}) \neq \mathrm{sgn}(\theta_{i,p})\} - \sum_{t \in \mathcal{T}} \mathbb{1}\{\mathrm{sgn}(a_{t,p}) \neq \mathrm{sgn}(\theta_{t,p})\} \Big\}$$
$$\tag{118}$$

where Equation (116) is obtained by using the cardinality of the set $\mathcal{I} \cap \mathcal{T}'$. Finally, we simplify Equation (117) to obtain the final expression given in Equation (118).

This completes the proof. $\qquad\square$

## C.3 Proof of Corollary 1

**Corollary 1.** *Following Lemma 2, if each agent $i \in \mathcal{N}$ selects its action according to $a_{i,j} = sgn(\theta_{i,j}) \ \forall \ j \in [d-1]$, the resulting transition probability is independent of $\mathcal{I}$ and $\mathcal{T}$, and it only depends on $r$ and $r'$. We denote the corresponding global action by $\boldsymbol{a}_\theta$ and resulting transition probabilities by $p^*_{r,r'}$.*

*Proof.* Putting $\boldsymbol{a} = \boldsymbol{a}_\theta$, all indicators in the final equation from Lemma 2 proof vanish leaving probability only a function of $r$ and $r'$ for a given instance.

$$\mathbb{P}(\boldsymbol{s}'|\boldsymbol{s}, \boldsymbol{a}_\theta) = \frac{r' + (r - 2r')\delta}{n \cdot 2^{r-1}} + \frac{n-r}{n \cdot 2^r} + \frac{\Delta}{n}(r - 2r') = p^*_{r,r'} \tag{119}$$

$\square$

## C.4 Proof of Lemma 3

**Lemma 3.** *For every $r \in [n-1]$, the following holds: (a) for $r' \in \{0, \ldots, \lfloor \frac{r+1}{2} \rfloor\}$, we have $\binom{r+1}{r'} p^*_{r+1,r'} < \binom{r}{r'} p^*_{r,r'}$ and (b) for $r' \in \{\lfloor \frac{r+1}{2} \rfloor + 1, \ldots, r\}$, we have $\binom{r+1}{r'} p^*_{r+1,r'} > \binom{r}{r'} p^*_{r,r'}$.*

*Proof.* Consider any $r \in [n-1]$

**Part (a)** For any $r' \in \{0, 1, \ldots, \lfloor \frac{r+1}{2} \rfloor\}$.

For our instances, we have

$$\Delta < \frac{1 - 2\delta}{2^n(1 + n + n^2)} < \frac{1 - 2\delta}{2^n \left(1 + n + \frac{n^2}{8}\right)}. \tag{120}$$

Rearranging the above terms, we have

$$\Delta\left(\frac{n^2}{8} + n + 1\right) < \frac{1 - 2\delta}{2^n}. \tag{121}$$

Since $1 \le r \le n$, it follows that $\frac{n^2}{8} + n + 1 \ge \frac{r^2}{8} + r + 1$, and therefore

$$\Delta\left(\frac{r^2}{8} + r + 1\right) < \frac{1 - 2\delta}{2^n}. \tag{122}$$

Moreover, using the fact that $\frac{r+1}{2^{r+1}} \ge \frac{1}{2^n}$ for our range of $r$, we obtain

$$\frac{\Delta}{n}\left(\frac{r^2}{8} + r + 1\right) < (r+1) \cdot \frac{1 - 2\delta}{n \cdot 2^{r+1}}. \tag{123}$$

Rewriting the left-hand side, we get

$$\frac{\Delta}{n} \cdot \frac{r^2}{8} + (r+1) \cdot \frac{\Delta}{n} < (r+1) \cdot \frac{1 - 2\delta}{n \cdot 2^{r+1}}. \tag{124}$$

Next, we use the inequality $r'(r - 2r') \le \frac{r^2}{8}$, which holds for all $0 \le r' \le \lfloor \frac{r+1}{2} \rfloor$, to deduce

$$\frac{\Delta}{n} \cdot r'(r - 2r') + (r+1)\left(\frac{\delta - 0.5}{n \cdot 2^r} + \frac{\Delta}{n}\right) < 0. \tag{125}$$

Now, since $0 \le r' \le \lfloor \frac{r+1}{2} \rfloor$, we have that

$$\frac{2r' - (r+1)}{2} \le 0. \tag{126}$$

Also, since $0 \le r' \le r \le n$, and $\delta \in (2/5, 1/2)$, we have that

$$0.5n + (1 - 2\delta)(r' - 0.5r) \ge 0.5n + (1 - 2\delta)(-0.5r) \tag{127}$$

$$\ge 0.5n + (1 - 2\delta)(-0.5n) \tag{128}$$

$$= n \cdot \delta > 0. \tag{129}$$

Using Equations (125), (126) and (129), we have

$$\frac{2r' - (r+1)}{2} \cdot \frac{(r' - 0.5r)(1 - 2\delta) + 0.5n}{n \cdot 2^{r-1}} + \frac{\Delta}{n} r'(r - 2r') + (r+1)\left(\frac{\delta - 0.5}{n \cdot 2^r} + \frac{\Delta}{n}\right) < 0. \tag{130}$$

After appropriate rearrangement of terms in Equation (130), we obtain

$$(r+1) \cdot \left(\frac{r' + (r + 1 - 2r')\delta}{n \cdot 2^r} + \frac{n - r - 1}{n \cdot 2^{r+1}} + \frac{\Delta}{n}(r + 1 - 2r')\right)$$
$$< (r + 1 - r') \cdot \left(\frac{r' + (r - 2r')\delta}{n \cdot 2^{r-1}} + \frac{n - r}{n \cdot 2^r} + \frac{\Delta}{n}(r - 2r')\right). \tag{131}$$

The above inequality is equivalent to

$$\frac{r+1}{r+1-r'} \cdot p^*_{r+1,r'} < p^*_{r,r'}, \tag{132}$$

which in turn implies

$$\binom{r+1}{r'} \cdot p^*_{r+1,r'} < \binom{r}{r'} \cdot p^*_{r,r'}, \tag{133}$$

where we used the expression of transition probabilities $p^*_{r+1,r'}$ and $p^*_{r,r'}$ from Lemma 2. This completes the proof of part (a). We now proceed to the second case, where $r' > \lfloor \frac{r+1}{2} \rfloor$.

**Part (b):** Now consider the case where $r \geq r' > \lfloor \frac{r+1}{2} \rfloor$. Again, recall that for any of our instances, the parameters satisfy

$$\Delta < \frac{1 - 2\delta}{2^n(n^2 + n + 1)} < \frac{3\delta - 1}{2^n(n^2 + n + 1)}, \quad \text{with } \delta \in (2/5, 1/2). \tag{134}$$

The inequality follows because $\delta \in (2/5, 1/2)$ and hence we have $1 - 2\delta < 3\delta - 1$.
Now, using the above inequality, we have as $n \geq 1$ that

$$n > \frac{0.5 - \delta}{2\delta - 0.5} + \frac{\Delta \cdot 2^n(n^2 + n + 1)}{2\delta - 0.5}. \tag{135}$$

Rearranging the above terms we have

$$2n\delta - 0.5n + \delta - 0.5 > \Delta \cdot 2^n(n^2 + n + 1). \tag{136}$$

Using the fact that $n^2 + n + 1 \geq n^2 - n + 1 \geq r^2 - r + 1$ for $r \leq n$, we get

$$0.5n - n + 2n\delta - 0.5 + \delta > \Delta \cdot (r^2 - r + 1) \cdot 2^n \tag{137}$$

Rearranging the terms, we have

$$(-n - 0.5)(1 - 2\delta) + 0.5n > \Delta \cdot (r^2 - r + 1) \cdot 2^n > \Delta \cdot (r^2 - r + 1) \cdot 2^r \tag{138}$$

Again using $n > r$, we get

$$\frac{(-r - 0.5)(1 - 2\delta) + 0.5n}{n \cdot 2^r} > \frac{\Delta}{n}(r^2 - r - 1) \tag{139}$$

Further since $r \geq r' > \lfloor \frac{r+1}{2} \rfloor$, we have $r \cdot r' - 2r'^2 \geq -r^2$. Using this along with $\delta \in (2/5, 1/2)$ we get

$$\frac{(r' - r - 0.5)(1 - 2\delta) + 0.5n}{n \cdot 2^r} + \frac{\Delta}{n}(1 + r + r \cdot r' - 2r'^2) > 0 \tag{140}$$

Therefore

$$\frac{(r' - 0.5r)(1 - 2\delta) + 0.5n}{n \cdot 2^r} + \frac{\Delta}{n}(1 + r + r \cdot r' - 2r'^2) - 0.5(r+1)\left(\frac{1 - 2\delta}{n \cdot 2^r}\right) > 0 \tag{141}$$

Simplifying this further, we get

$$\frac{(r' - 0.5r)(1 - 2\delta) + 0.5n}{n \cdot 2^r} + \frac{\Delta}{n} \cdot r'(r - 2r') + (r+1)\left(\frac{\delta - 0.5}{n \cdot 2^r} + \frac{\Delta}{n}\right) > 0. \tag{142}$$

Noting that $r' > \lfloor \frac{r+1}{2} \rfloor$ implies $\frac{2r' - (r+1)}{2} \geq 1/2$. Hence, from the above inequality we get

$$\frac{2r' - (r+1)}{2} \cdot \frac{(r' - 0.5r)(1 - 2\delta) + 0.5n}{n \cdot 2^{r-1}} + \frac{\Delta}{n} \cdot r'(r - 2r') + (r+1)\left(\frac{\delta - 0.5}{n \cdot 2^r} + \frac{\Delta}{n}\right) > 0. \tag{143}$$

Simplifying this further, we have

$$(r+1) \cdot \left(\frac{(r' - 0.5r)(1 - 2\delta) + 0.5n}{n \cdot 2^r}\right) + (r+1)\left(\frac{\delta - 0.5}{n \cdot 2^r} + \frac{\Delta}{n}\right) + (r+1)\frac{\Delta}{n}(r - 2r')$$

$$> (r + 1 - r')\left(\frac{(r' - 0.5r)(1 - 2\delta) + 0.5n}{n \cdot 2^{r-1}}\right) + (r + 1 - r')\frac{\Delta}{n}(r - 2r') \tag{144}$$

The above can be further simplified as

$$(r+1) \cdot \left(\frac{(r' - 0.5r)(1 - 2\delta) + \delta - 0.5 + 0.5n}{n \cdot 2^r} + \frac{\Delta}{n}(r + 1 - 2r')\right)$$

$$> (r + 1 - r') \cdot \left(\frac{(r' - 0.5r)(1 - 2\delta) + 0.5n}{n \cdot 2^{r-1}} + \frac{\Delta}{n}(r - 2r')\right) \tag{145}$$

Therefore, we finally have

$$(r+1) \cdot \left(\frac{r' + (r + 1 - 2r')\delta}{n \cdot 2^r} + \frac{n - r - 1}{n2^{r+1}} + \frac{\Delta}{n}(r + 1 - 2r')\right)$$

$$> (r + 1 - r') \cdot \left(\frac{r' + (r - 2r')\delta}{n \cdot 2^{r-1}} + \frac{n - r}{n \cdot 2^r} + \frac{\Delta}{n}(r - 2r')\right) \tag{146}$$

The above inequality is equivalent to

$$\frac{r+1}{r + 1 - r'} \cdot p^*_{r+1,r'} > p^*_{r,r'}, \tag{147}$$

which in turn implies

$$\binom{r+1}{r'} \cdot p^*_{r+1,r'} > \binom{r}{r'} \cdot p^*_{r,r'}, \tag{148}$$

where we used the transition probabilities $p^*_{r+1,r'}$, and $p^*_{r,r'}$ from Lemma 2. This completes the proof. $\qquad\square$

## C.5 Proof of Lemma 4

**Lemma 4.** *Let $r \in [n-1]$. Suppose that for every $r' \in \{0, 1, \ldots, \lfloor \frac{r+1}{2} \rfloor\}$, we have $\binom{r+1}{r'}p^*_{r+1,r'} < \binom{r}{r'}p^*_{r,r'}$, and for every $r' \in \{\lfloor \frac{r+1}{2} \rfloor + 1, \ldots, r\}$, we have $\binom{r+1}{r'}p^*_{r+1,r'} > \binom{r}{r'}p^*_{r,r'}$. Further, if $V^*_r > V^*_{r-1} > \cdots > V^*_1 > V^*_0 = 0$, then $V^*_{r+1} > V^*_r$.*

*Proof.* Using the lemma hypothesis for every $r' \in \{0, 1, \ldots, r\}$, there exist constants $\delta_{r,r'} > 0$ such that

$$\binom{r+1}{r'}p^*_{r+1,r'} = \binom{r}{r'}p^*_{r,r'} - \delta_{r,r'}, \quad \forall \ r' \in \left\{0, 1, \ldots, \lfloor \frac{r+1}{2} \rfloor\right\}, \quad \text{and} \tag{149}$$

$$\binom{r+1}{r'}p^*_{r+1,r'} = \binom{r}{r'}p^*_{r,r'} + \delta_{r,r'}, \quad \forall \ r' \in \left\{\lfloor \frac{r+1}{2} \rfloor + 1, \ldots, r - 1, r\right\}. \tag{150}$$

Recall that from any state of *type* $r + 1$, there are $\binom{r+1}{r'}$ *reachable* states of *type* $r'$ for $r' \in \{0, 1, \dots, r + 1\}$. Suppose $\boldsymbol{a}_\theta$ is the action taken in this state, the sum of transition probabilities over the above mentioned states must add up to 1. Thus,

$$1 = \sum_{r'=0}^{r+1} \binom{r + 1}{r'} p_{r+1,r'}^* \tag{151}$$

$$= \sum_{r'=0}^{\lfloor \frac{r+1}{2} \rfloor} \binom{r + 1}{r'} p_{r+1,r'}^* + \sum_{r'=\lfloor \frac{r+1}{2} \rfloor+1}^{r} \binom{r + 1}{r'} p_{r+1,r'}^* + \binom{r + 1}{r + 1} p_{r+1,r+1}^* \tag{152}$$

$$= \sum_{r'=0}^{\lfloor \frac{r+1}{2} \rfloor} \left( \binom{r}{r'} p_{r,r'}^* - \delta_{r,r'} \right) + \sum_{r'=\lfloor \frac{r+1}{2} \rfloor+1}^{r} \left( \binom{r}{r'} p_{r,r'}^* + \delta_{r,r'} \right) + \binom{r + 1}{r + 1} p_{r+1,r+1}^* \tag{153}$$

$$= \sum_{r'=0}^{\lfloor \frac{r+1}{2} \rfloor} \binom{r}{r'} p_{r,r'}^* - \sum_{r'=0}^{\lfloor \frac{r+1}{2} \rfloor} \delta_{r,r'} + \sum_{r'=\lfloor \frac{r+1}{2} \rfloor+1}^{r} \binom{r}{r'} p_{r,r'}^* + \sum_{r'=\lfloor \frac{r+1}{2} \rfloor+1}^{r} \delta_{r,r'} + \binom{r + 1}{r + 1} p_{r+1,r+1}^* \tag{154}$$

This implies

$$p_{r+1,r+1}^* = \sum_{r'=0}^{\lfloor \frac{r+1}{2} \rfloor} \delta_{r,r'} - \sum_{r'=\lfloor \frac{r+1}{2} \rfloor+1}^{r} \delta_{r,r'} + 1 - \sum_{0 \le r' \le r} \binom{r}{r'} p_{r,r'}^* \tag{155}$$

$$= \sum_{r'=0}^{\lfloor \frac{r+1}{2} \rfloor} \delta_{r,r'} - \sum_{r'=\lfloor \frac{r+1}{2} \rfloor+1}^{r} \delta_{r,r'} \tag{156}$$

where, the last equation follows because $\sum_{0 \le r' \le r} \binom{r}{r'} p_{r,r'}^* = 1$. This can be interpreted as the sum of transition probabilities over all possible next states from a state of type $r$ under action $\boldsymbol{a}_\theta$.

Next, recall that $V_r^*$ is given by

$$V_r^* = 1 + \sum_{r'=0}^{r} \binom{r}{r'} p_{r,r'}^* V_{r'}^* \tag{157}$$

Using this we have that

$$V_{r+1}^* = 1 + p_{r+1,r+1}^* V_{r+1}^* + \sum_{r'=0}^{r} \binom{r + 1}{r'} p_{r+1,r'}^* V_{r'}^* \tag{158}$$

$$= 1 + p_{r+1,r+1}^* V_{r+1}^* + \sum_{r'=0}^{\lfloor \frac{r+1}{2} \rfloor} \binom{r + 1}{r'} p_{r+1,r'}^* V_{r'}^* + \sum_{r'=\lfloor \frac{r+1}{2} \rfloor+1}^{r} \binom{r + 1}{r'} p_{r+1,r'}^* V_{r'}^* \tag{159}$$

$$= 1 + p_{r+1,r+1}^* V_{r+1}^* + \sum_{r'=0}^{\lfloor \frac{r+1}{2} \rfloor} \left( \binom{r}{r'} p_{r,r'}^* - \delta_{r,r'} \right) V_{r'}^*$$

$$+ \sum_{r'=\lfloor \frac{r+1}{2} \rfloor+1}^{r} \left( \binom{r}{r'} p_{r,r'}^* + \delta_{r,r'} \right) V_{r'}^* \tag{160}$$

$$= 1 + p_{r+1,r+1}^* V_{r+1}^* + \sum_{r'=0}^{r} \binom{r}{r'} p_{r,r'}^* V_{r'}^* - \sum_{r'=0}^{\lfloor \frac{r+1}{2} \rfloor} \delta_{r,r'} V_{r'}^* + \sum_{r'=\lfloor \frac{r+1}{2} \rfloor+1}^{r} \delta_{r,r'} V_{r'}^* \tag{161}$$

$$= p_{r+1,r+1}^* V_{r+1}^* + V_r^* - \sum_{r'=0}^{\lfloor \frac{r+1}{2} \rfloor} \delta_{r,r'} V_{r'}^* + \sum_{r'=\lfloor \frac{r+1}{2} \rfloor+1}^{r} \delta_{r,r'} V_{r'}^* \tag{162}$$

$$> p^*_{r+1,r+1}V^*_{r+1} + V^*_r - \sum_{r'=0}^{\lfloor \frac{r+1}{2} \rfloor} \delta_{r,r'} V^*_{\lfloor \frac{r+1}{2} \rfloor +1} + \sum_{r'=\lfloor \frac{r+1}{2} \rfloor +1}^{r} \delta_{r,r'} V^*_{\lfloor \frac{r+1}{2} \rfloor +1} \tag{163}$$

$$= p^*_{r+1,r+1}V^*_{r+1} + V^*_r - \left( \sum_{r'=0}^{\lfloor \frac{r+1}{2} \rfloor} \delta_{r,r'} - \sum_{r'=\lfloor \frac{r+1}{2} \rfloor +1}^{r} \delta_{r,r'} \right) V^*_{\lfloor \frac{r+1}{2} \rfloor +1} \tag{164}$$

$$= p^*_{r+1,r+1}V^*_{r+1} + V^*_r - p^*_{r+1,r+1}V^*_{\lfloor \frac{r+1}{2} \rfloor +1} \tag{165}$$

$$> p^*_{r+1,r+1}V^*_{r+1} + V^*_r - p^*_{r+1,r+1}V^*_r \tag{166}$$

where in (162), we use the definition of $V^*_r$ from the Equation (157). (165) follows from Equation (156). In (163) and (166) we utilize the monotonicity assumption of $V^*$ with respect to $r$.

Thus, combining above with the fact that for our instances for any $r \in [n-1]$ and $p^*_{r+1,r+1} < 1$, we have that

$$V^*_{r+1} > p^*_{r+1,r+1}V^*_{r+1} + V^*_r - p^*_{r+1,r+1}V^*_r \tag{167}$$

This implies that

$$(1 - p^*_{r+1,r+1})(V^*_{r+1} - V^*_r) > 0 \implies V^*_{r+1} > V^*_r. \tag{168}$$

This completes the proof. $\qquad \square$

### C.6   Proof of Lemma 5

**Lemma 5.** *For any of our instances, $\forall\, (\boldsymbol{s},\boldsymbol{a}) \in \mathcal{S} \times \mathcal{A}, \sum_{s' \neq s} \{\mathbb{P}(\boldsymbol{s'}|\boldsymbol{s},\boldsymbol{a}) - \mathbb{P}(\boldsymbol{s'}|\boldsymbol{s},\boldsymbol{a}^*)\}V^*(\boldsymbol{s'}) \geq 0.$*

*Proof.* For $r = 0$, the statement trivially holds. For any $\boldsymbol{s} \in \mathcal{S}_r$ for any $r \in [n]$ and any $\boldsymbol{a} \in \mathcal{A}$, we have,

$$\sum_{s' \neq s}[\mathbb{P}(\boldsymbol{s'}|\boldsymbol{s},\boldsymbol{a}) - \mathbb{P}(\boldsymbol{s'}|\boldsymbol{s},\boldsymbol{a}^*)]V^*(\boldsymbol{s'}) = \sum_{r'=1}^{r-1} \sum_{s' \in \mathcal{S}_{r'}(s)} [\mathbb{P}(\boldsymbol{s'}|\boldsymbol{s},\boldsymbol{a}) - \mathbb{P}(\boldsymbol{s'}|\boldsymbol{s},\boldsymbol{a}^*)]V^*(\boldsymbol{s'}) \tag{169}$$

The above follows as there is only one state of *type* 0 i.e., $\boldsymbol{g}$ and $V^*(\boldsymbol{g}) = 0$. Also, all the other states *reachable* from $\boldsymbol{s}$ (apart from itself) lie in $\cup_{r' \in [r-1]}\mathcal{S}_{r'}(s)$. For other states that are not reachable, the probability is 0.

Using the transition probability obtained in Lemma 2 (along with the notation $\mathcal{I}, \mathcal{J}, \mathcal{T}$ that varies with individual transitions) and Theorem 1, the above can be further written as

$$\sum_{s' \neq s}[\mathbb{P}(\boldsymbol{s'}|\boldsymbol{s},\boldsymbol{a}) - \mathbb{P}(\boldsymbol{s'}|\boldsymbol{s},\boldsymbol{a}^*)]V^*(\boldsymbol{s'}) \tag{170}$$

$$= \sum_{r'=1}^{r-1} \sum_{s' \in \mathcal{S}_{r'}(s)} \frac{2\Delta}{n(d-1)} \sum_{p=1}^{d-1} \left( \sum_{i \in \mathcal{I} \cap \mathcal{T'}} \mathbb{1}\{\mathrm{sgn}(a_{i,p}) \neq \mathrm{sgn}(\theta_{i,p})\} \right.$$

$$\left. - \sum_{t \in \mathcal{T}} \mathbb{1}\{\mathrm{sgn}(a_{t,p}) \neq \mathrm{sgn}(\theta_{t,p})\} \right) V^*_{r'} \tag{171}$$

$$= \frac{2\Delta}{n(d-1)} \sum_{r'=1}^{r-1} V^*_{r'} \sum_{p=1}^{d-1} \left[ \sum_{s' \in \mathcal{S}_{r'}(s)} \left( \sum_{i \in I \cap T'} \mathbb{1}\{\mathrm{sgn}(a_{i,p}) \neq \mathrm{sgn}(\theta_{i,p})\} \right. \right.$$

$$\left. \left. - \sum_{t \in \mathcal{T}} \mathbb{1}\{\mathrm{sgn}(a_{t,p}) \neq \mathrm{sgn}(\theta_{t,p})\} \right) \right] \tag{172}$$

In the inner difference of summation of indicators, for any fixed $p \in [d-1]$ we observe that for any agent $i \in [n]$ the indicator has positive coefficient if the agent stays at node $s$ itself, whereas it has

negative sign if the agent transits to node $g$. Hence, we write the following along with swapping the summations:

$$\sum_{s' \neq s} [\mathbb{P}(s'|s, a) - \mathbb{P}(s'|s, a^*)]V^*(s') \tag{173}$$

$$= \frac{2\Delta}{n(d-1)} \sum_{r'=1}^{r-1} V_{r'}^* \sum_{p=1}^{d-1} \left( \sum_{i \in \mathcal{I}} \sum_{s' \in \mathcal{S}_{r'}(s)} [\mathbb{1}\{s_i' = s\} - \mathbb{1}\{s_i' = g\}]\mathbb{1}\{\mathrm{sgn}(a_{i,p}) \neq \mathrm{sgn}(\theta_{i,p})\} \right) \tag{174}$$

$$= \frac{2\Delta}{n(d-1)} \sum_{r'=1}^{r-1} V_{r'}^* \sum_{p=1}^{d-1} \sum_{i \in \mathcal{I}} \mathbb{1}\{\mathrm{sgn}(a_{i,p}) \neq \mathrm{sgn}(\theta_{i,p})\} \sum_{s' \in \mathcal{S}_{r'}(s)} (\mathbb{1}\{s_i' = s\} - \mathbb{1}\{s_i' = g\}) \tag{175}$$

$$= \frac{2\Delta}{n(d-1)} \sum_{r'=1}^{r-1} V_{r'}^* \sum_{p=1}^{d-1} \sum_{i \in \mathcal{I}} \mathbb{1}\{\mathrm{sgn}(a_{i,p}) \neq \mathrm{sgn}(\theta_{i,p})\} \left[ \binom{r-1}{r'-1} - \binom{r-1}{r'} \right] \tag{176}$$

The final steps follows from calculating how many states of type $r'$ that are *reachable* from $s$ are possible with the particular agent at node $s$ or node $g$ in the next state. Simplifying this further, we have that

$$\sum_{s' \neq s} [\mathbb{P}(s'|s, a) - \mathbb{P}(s'|s, a^*)]V^*(s') \tag{177}$$

$$= \frac{2\Delta}{n(d-1)} \sum_{r'=1}^{r-1} V_{r'}^* \sum_{p=1}^{d-1} \sum_{i \in \mathcal{I}} \left( \mathbb{1}\{\mathrm{sgn}(a_{i,p}) \neq \mathrm{sgn}(\theta_{i,p})\} \frac{2r'-r}{r} \binom{r}{r'} \right) \tag{178}$$

$$= \frac{2\Delta}{n(d-1)} \sum_{p=1}^{d-1} \sum_{i=1}^{n} \left( \mathbb{1}\{\mathrm{sgn}(a_{i,p}) \neq \mathrm{sgn}(\theta_{i,p})\} \sum_{r'=1}^{r-1} V_{r'}^* \frac{2r'-r}{r} \binom{r}{r'} \right) \tag{179}$$

where, the second equation follows by re-arranging the order of summations. Now, using the fact that for any function $f$, $\sum_{a \in [m]} f(a) = \frac{1}{2} [\sum_{a \in [m]} f(a) + \sum_{a \in [m]} f(m+1-a)]$, we have

$$\sum_{s' \neq s} [\mathbb{P}(s'|s, a) - \mathbb{P}(s'|s, a^*)]V^*(s') \tag{180}$$

$$= \frac{2\Delta}{n(d-1)} \sum_{p=1}^{d-1} \sum_{i=1}^{n} \Bigg[ \mathbb{1}\{\mathrm{sgn}(a_{i,p}) \neq \mathrm{sgn}(\theta_{i,p})\} \tag{181}$$

$$\sum_{r'=1}^{r-1} \frac{1}{2} \cdot \left( V_{r'}^* \frac{2r'-r}{r} \binom{r}{r'} + V_{r-r'}^* \frac{2(r-r')-r}{r} \binom{r}{r-r'} \right) \Bigg] \tag{182}$$

$$= \frac{\Delta}{n(d-1)} \sum_{p=1}^{d-1} \sum_{i=1}^{n} \left[ \mathbb{1}\{\mathrm{sgn}(a_{i,p}) \neq \mathrm{sgn}(\theta_{i,p})\} \sum_{r'=1}^{r-1} \left( V_{r'}^* \frac{2r'-r}{r} \binom{r}{r'} - V_{r-r'}^* \frac{2r'-r}{r} \binom{r}{r'} \right) \right] \tag{183}$$

$$= \frac{\Delta}{n(d-1)} \sum_{p=1}^{d-1} \sum_{i=1}^{n} \left[ \mathbb{1}\{\mathrm{sgn}(a_{i,p}) \neq \mathrm{sgn}(\theta_{i,p})\} \sum_{r'=1}^{r-1} \left( V_{r'}^* - V_{r-r'}^* \right) \frac{2r'-r}{r} \binom{r}{r'} \right] \tag{184}$$

$$\geq 0. \tag{185}$$

where the last step holds because $V_{r'}^* \geq V_{r-r'}^*$ when $r' \geq r/2$ and vice-versa due to third claim in Theorem 1. This completes the proof. $\qquad \square$

## C.7 Proof of Lemma 6

**Lemma 6.** *For any instance $(n, \delta, \Delta, \theta)$, algorithm $\pi$, and $T \geq 2KV_1^*$, we have $\mathbb{E}_{\theta,\pi}[N_i^-] \geq \frac{KV_1^*}{4}$ for any $i \in [n]$.*

*Proof.* Recall the capped process for any agent from Theorem 2 proof. If capped process for agent $i \in [n]$ finishes before $T$, then $N_i^- = N_i$. Else, number of visits to node $g$ for that agent is less than $K$ due to which $N_i^- \geq T - K$. Therefore, we have for any $i \in [n]$,

$$\mathbb{E}_{\theta,\pi}[N_i^-] \geq \mathbb{E}_{\theta,\pi}[\min\{T - K, N_i\}] \geq \mathbb{E}_{\theta,\pi}\left[\sum_{i=1}^{K} \min\{T/K - 1, N_{i,k}^\pi\}\right] \quad (186)$$

In the above, $N_{i,k}^\pi$ is the random variable representing the number of time steps in episode $k$ that the agent $i$ stayed at node $s$ when the global policy being followed was $\pi$. Since $T \geq 2KV_1^*$, and $V_1^* \geq 1$, we have $T/K - 1 > V_1^*$. Using this, from Equation (186), we observe that the statement of lemma holds for any algorithm $\pi$ if for any agent $i$, $N_{i,k}^\pi \geq V_1^*$ with probability at least $1/4$. So, it suffices to prove that for any $\pi$, the probability that $N_{i,k}^\pi \geq V_1^*$ is at least $1/4$.

Firstly, we note from the proof of Theorem 1 that $V_1^* = \frac{2n}{n-1+2(\delta+\Delta)} > 2$ since

$$\frac{1}{V_1^*} = \frac{n-1}{2n} + \frac{2(\delta + \Delta)}{2n} \quad (187)$$

$$< \frac{n-1}{2n} + \frac{2(\delta + 2^{-n} \cdot \frac{1-2\delta}{1+n+n^2})}{2n} \quad (188)$$

$$< \frac{n-1}{2n} + \frac{2(\delta + \frac{1-2\delta}{6})}{2n} \quad (189)$$

$$= \frac{n-1}{2n} + \frac{(\frac{1+4\delta}{3})}{2n} \quad (190)$$

$$< 1/2 \quad (191)$$

Above, the first inequality arises because $\Delta < \frac{1-2\delta}{1+n+n^2}$ for our instances, the second inequality holds because $n \geq 1$ and the final step is due to $\delta < 1/2$.

Next, we state the following lemma whose proof is available in Appendix C.9.

**Lemma 8.** *Take any agent $\tilde{i} \in [n]$ currently at node $s$. Then under any global action $\boldsymbol{a} \in \mathcal{A}$, the probability that agent $\tilde{i}$ stays in node $s$ in the next step is lower bounded by $1/2$. Formally, if $\boldsymbol{s}$ is such that $s_i = s$, then $\sum_{\boldsymbol{s}'|s_i'=s} \mathbb{P}[\boldsymbol{s}'|\boldsymbol{s}, \boldsymbol{a}] > 1/2$ for any $\boldsymbol{a} \in \mathcal{A}$.*

Now, we observe that, we can express $\mathbb{P}[N_{i,k}^\pi \geq x] = \mathbb{P}[\{(s^1, s^2, \ldots, s^{\lceil x \rceil})|s_i^t = s \,\forall\, t \in [\lceil x \rceil], s^1 = s_{\text{init}}\}]$ i.e., starting at initial state, the probability of observing a $\lceil x \rceil$ length sequence with agent $i$ always at node $s$. Let the set of all states $\boldsymbol{s}$ with $s_i = s$ be $\mathcal{S}(i)$. We further evaluate the LHS below

$$\sum_{\{s^1,\ldots,s^{\lceil x \rceil}\} \in \mathcal{S}(i)^{\times \lceil x \rceil}} \mathbb{P}[s^1, \ldots, s^{\lceil x \rceil}] \quad (192)$$

$$= \sum_{\{s^1,\ldots,s^{\lceil x \rceil - 1}\} \in \mathcal{S}(i)^{\times \lceil x \rceil - 1}} \mathbb{P}[s^1, \ldots, s^{\lceil x \rceil - 1}] \cdot \mathbb{P}[s^{\lceil x \rceil} \in \mathcal{S}(i) \mid \{s^1, \ldots, s^{\lceil x \rceil - 1}\} \in \mathcal{S}(i)^{\lceil x \rceil - 1}] \quad (193)$$

$$= \sum_{\{s^1,\ldots,s^{\lceil x \rceil - 1}\} \in \mathcal{S}(i)^{\times \lceil x \rceil - 1}} \mathbb{P}[s^1, \ldots, s^{\lceil x \rceil - 1}] \cdot \mathbb{P}[s^{\lceil x \rceil} \in \mathcal{S}(i) \mid s^{\lceil x \rceil - 1} \in \mathcal{S}(i)] \quad (194)$$

$$\geq \sum_{\{s^1,\ldots,s^{\lceil x \rceil - 1}\} \in \mathcal{S}(i)^{\times \lceil x \rceil - 1}} \mathbb{P}[s^1, \ldots, s^{\lceil x \rceil - 1}] \cdot (1/2) \quad (195)$$

$$\vdots \quad (196)$$

$$\geq \mathbb{P}[s^1 \in \mathcal{S}(i)] \cdot (1/2)^{\lceil x \rceil - 1} > (1/2)^x \quad (197)$$

Thus, we obtain, $\mathbb{P}[N_{i,k}^\pi \geq x] > (1/2)^x$.

Further, we use $\frac{1}{2} > \frac{1}{2n} - \frac{\delta+\Delta}{n} = p_{1,1}^* = 1 - p_{1,0}^* = 1 - 1/V_1^*$ to obtain,

$$\mathbb{P}[N_{i,k}^\pi \geq x] > (1 - 1/V_1^*)^x \quad (198)$$

Putting $x = V_1^*$ and using that $V_1^* > 2$, $\mathbb{P}[N_{i,k}^\pi \geq V_1^*] > 1/4$, noting that for any $\alpha \geq 2$, we have $(1 - \frac{1}{\alpha})^\alpha \geq 1/4$.

□

## C.8 Proof of Lemma 7

**Lemma 7.** *For any instance $(n, \delta, \Delta, \theta)$ and algorithm $\pi$, for any $j \in [d-1]$, we have,*

$$\mathrm{KL}(\mathbb{P}_\theta^\pi || \mathbb{P}_{\theta^j}^\pi) \leq 3 \cdot 2^{2n} \cdot \frac{\Delta^2}{\delta(d-1)^2} \cdot \mathbb{E}_{\theta,\pi}[N^-]. \tag{64}$$

*Proof.* We let the $t$-length tuple consisting of the sequence of states from first to $t^{th}$ time step be denoted by $\bar{s}_t \in \mathcal{S}^{\times t}$, i.e., $t$ times cross product of state space. Recall that $s^t$ represents the state at time $t$.

By chain rule of KL divergence for any $\theta \in \Theta$, and $j \in [d-1]$,

$$\mathrm{KL}(\mathbb{P}_\theta^\pi || \mathbb{P}_{\theta^j}^\pi) = \sum_{t=1}^{T-1} \mathrm{KL}[\mathbb{P}_\theta^\pi(s^{t+1}|\bar{s}_t)||\mathbb{P}_{\theta^j}^\pi(s^{t+1}|\bar{s}_t)] \tag{199}$$

where,

$$\mathrm{KL}[\mathbb{P}_\theta^\pi(s^{t+1}|\bar{s}_t)||\mathbb{P}_{\theta^j}^\pi(s^{t+1}|\bar{s}_t)]$$

$$= \sum_{\bar{s}_t \in \mathcal{S}^{\times t}} \mathbb{P}_\theta^\pi(\bar{s}_t) \sum_{s^{t+1} \in \mathcal{S}} \mathbb{P}_\theta^\pi(s^{t+1}|\bar{s}_t) \cdot \ln \left[ \frac{\mathbb{P}_\theta^\pi(s^{t+1}|\bar{s}_t)}{\mathbb{P}_{\theta^j}^\pi(s^{t+1}|\bar{s}_t)} \right] \tag{200}$$

$$= \sum_{\bar{s}_t \in \mathcal{S}^{\times t}} \mathbb{P}_\theta^\pi(\bar{s}_t) \cdot \sum_{s' \in \mathcal{S}} \mathbb{P}_\theta^\pi(s^{t+1} = s'|\bar{s}_t) \cdot \ln \left[ \frac{\mathbb{P}_\theta^\pi(s^{t+1} = s'|\bar{s}_t)}{\mathbb{P}_{\theta^j}^\pi(s^{t+1} = s'|\bar{s}_t)} \right] \tag{201}$$

$$= \sum_{\bar{s}_{t-1} \in \mathcal{S}^{\times t-1}} \mathbb{P}_\theta^\pi(\bar{s}_{t-1}) \cdot \sum_{s \in \mathcal{S}, a \in \mathcal{A}} \mathbb{P}_\theta^\pi(s^t = s, a^t = a|\bar{s}_{t-1})$$

$$\times \sum_{s' \in \mathcal{S}} \mathbb{P}_\theta^\pi(s^{t+1} = s'|\bar{s}_{t-1}, s^t = s, a^t = a) \cdot \ln \left[ \frac{\mathbb{P}_\theta^\pi(s^{t+1} = s'|\bar{s}_{t-1}, s^t = s, a^t = a)}{\mathbb{P}_{\theta^j}^\pi(s^{t+1} = s'|\bar{s}_{t-1}, s^t = s, a^t = a)} \right] \tag{202}$$

In the above, we just use the laws of conditional divergences and conditional probabilities. Notice that in the above, for any fixed $(s, a)$, the inner summation (last line) is itself a KL-divergence for fixed parameters from outer summations. We will simplify this inner KL term now.

$$\sum_{s' \in \mathcal{S}} \mathbb{P}_\theta^\pi(s^{t+1} = s'|\bar{s}_{t-1}, s^t = s, a^t = a) \cdot \ln \left[ \frac{\mathbb{P}_\theta^\pi(s^{t+1} = s'|\bar{s}_{t-1}, s^t = s, a^t = a)}{\mathbb{P}_{\theta^j}^\pi(s^{t+1} = s'|\bar{s}_{t-1}, s^t = s, a^t = a)} \right]$$

$$= \sum_{s' \in \mathcal{S}} \mathbb{P}_\theta^\pi(s^{t+1} = s'|s^t = s, a^t = a) \cdot \ln \left[ \frac{\mathbb{P}_\theta^\pi(s^{t+1} = s'|s^t = s, a^t = a)}{\mathbb{P}_{\theta^j}^\pi(s^{t+1} = s'|s^t = s, a^t = a)} \right] \tag{203}$$

where the above equation holds due to Markovian Transitions. We'll upper bound the inner KL for any $(s, a) \in \mathcal{S} \times \mathcal{A}$.

For any probability distributions $A, B$, due to non-negativity of $\mathrm{KL}(.||.)$, we have $\mathrm{KL}(A||B) \leq \mathrm{KL}(A||B) + \mathrm{KL}(B||A)$. Thus,

$$\sum_{s' \in \mathcal{S}} \mathbb{P}_\theta^\pi(s^{t+1} = s'|s^t = s, a^t = a) \cdot \ln \left[ \frac{\mathbb{P}_\theta^\pi(s^{t+1} = s'|s^t = s, a^t = a)}{\mathbb{P}_{\theta^j}^\pi(s^{t+1} = s'|s^t = s, a^t = a)} \right]$$

$$\leq \sum_{s' \in \mathcal{S}} \mathbb{P}_\theta^\pi(s^{t+1} = s'|s^t = s, a^t = a) \cdot \ln \left[ \frac{\mathbb{P}_\theta^\pi(s^{t+1} = s'|s^t = s, a^t = a)}{\mathbb{P}_{\theta^j}^\pi(s^{t+1} = s'|s^t = s, a^t = a)} \right]$$

$$+ \sum_{s' \in \mathcal{S}} \mathbb{P}_{\theta^j}^\pi(s^{t+1} = s'|s^t = s, a^t = a) \cdot \ln \left[ \frac{\mathbb{P}_{\theta^j}^\pi(s^{t+1} = s'|s^t = s, a^t = a)}{\mathbb{P}_\theta^\pi(s^{t+1} = s'|s^t = s, a^t = a)} \right] \tag{204}$$

$$= \sum_{s' \in \mathcal{S}} \left[ \mathbb{P}_\theta^\pi (s^{t+1} = s' | s^t = s, a^t = a) - \mathbb{P}_{\theta^j}^\pi (s^{t+1} = s' | s^t = s, a^t = a) \right]$$

$$\times \ln \left[ \frac{\mathbb{P}_\theta^\pi (s^{t+1} = s' | s^t = s, a^t = a)}{\mathbb{P}_{\theta^j}^\pi (s^{t+1} = s' | s^t = s, a^t = a)} \right] \quad (205)$$

$$= \sum_{s' \in \mathcal{S}} \left[ \mathbb{P}_\theta^\pi (s^{t+1} = s' | s^t = s, a^t = a) - \mathbb{P}_{\theta^j}^\pi (s^{t+1} = s' | s^t = s, a^t = a) \right]$$

$$\times \ln \left[ 1 + \frac{\mathbb{P}_\theta^\pi (s^{t+1} = s' | s^t = s, a^t = a) - \mathbb{P}_{\theta^j}^\pi (s^{t+1} = s' | s^t = s, a^t = a)}{\mathbb{P}_{\theta^j}^\pi (s^{t+1} = s' | s^t = s, a^t = a)} \right] \quad (206)$$

Since $\ln(1 + x) < x$ for any $x \in \mathbb{R}$, from above, we have:

$$\sum_{s' \in \mathcal{S}} \mathbb{P}_\theta^\pi (s^{t+1} = s' | s^t = s, a^t = a) \cdot \ln \left[ \frac{\mathbb{P}_\theta^\pi (s^{t+1} = s' | s^t = s, a^t = a)}{\mathbb{P}_{\theta^j}^\pi (s^{t+1} = s' | s^t = s, a^t = a)} \right]$$

$$\leq \sum_{s' \in \mathcal{S}} \frac{[\mathbb{P}_\theta^\pi (s^{t+1} = s' | s^t = s, a^t = a) - \mathbb{P}_{\theta^j}^\pi (s^{t+1} = s' | s^t = s, a^t = a)]^2}{\mathbb{P}_{\theta^j}^\pi (s^{t+1} = s' | s^t = s, a^t = a)} \quad (207)$$

$$\leq 2^n \cdot \frac{\max_{s,a,s'} (\langle \phi(s'|s, a), \theta - \theta^j \rangle)^2}{\min_{s,a,s'} \mathbb{P}_{\theta^j}^\pi (s^{t+1} = s' | s^t = s, a^t = a)} \leq 2^n \cdot \left( \frac{2\Delta}{d-1} \right)^2 \bigg/ \left( \frac{\delta}{2^{n-1}} - \Delta \right) \quad (208)$$

In the last step, we upper bound every term in the summation by taking the maximum for the numerator and minimum for the denominator. The $2^n$ appears due to there being maximum of $2^n$ such terms for any $s \in \mathcal{S}$. Hence, the $2^n$ appears. Note that the difference $\theta - \theta^j$ is $nd$−dimensional and has non-zero components only at $(j + (i \cdot d))^{th}$ index and are equal to $2 \times \theta_{i,j}$ where $i = \{0, 1, \ldots, n-1\}$. The inner product maximizes when each component produces either $\frac{2\Delta}{n(d-1)}$ throughout or $-\frac{2\Delta}{n(d-1)}$. Either way, the maximum that the numerator can be is $\left\{ \frac{2\Delta}{(d-1)} \right\}^2$. For the minimum of the denominator, one can verify from general transition probability equation in Lemma 2 that it would be minimum when $s' = s = s_{\text{init}}$ and $a = a_\theta$.

Since $2/5 < \delta < 1/2$ and $\Delta < 2^{-n} (\frac{1-2\delta}{1+n+n^2})$, we have $\Delta < 2^{-n} \cdot \frac{2\delta}{3}$. Thus

$$\sum_{s' \in \mathcal{S}} \mathbb{P}_\theta^\pi (s^{t+1} = s' | s^t = s, a^t = a) \cdot \ln \left[ \frac{\mathbb{P}_\theta^\pi (s^{t+1} = s' | s^t = s, a^t = a)}{\mathbb{P}_{\theta^j}^\pi (s^{t+1} = s' | s^t = s, a^t = a)} \right]$$

$$< 2^n \cdot \frac{4\Delta^2}{(d-1)^2} \bigg/ \frac{2\delta}{3 \cdot 2^{n-1}} \quad (209)$$

$$= 3 \cdot 2^{2n} \frac{\Delta^2}{\delta(d-1)^2} \quad (210)$$

Hence, from Equations (203) and (210) we have

$$\sum_{s' \in \mathcal{S}} \mathbb{P}_\theta^\pi (s^{t+1} = s' | \bar{s}_{t-1}, s^t = s, a^t = a) \cdot \ln \left[ \frac{\mathbb{P}_\theta^\pi (s^{t+1} = s' | \bar{s}_{t-1}, s^t = s, a^t = a)}{\mathbb{P}_{\theta^j}^\pi (s^{t+1} = s' | \bar{s}_{t-1}, s^t = s, a^t = a)} \right]$$

$$= \sum_{s' \in \mathcal{S}} \mathbb{P}_\theta^\pi (s^{t+1} = s' | s^t = s, a^t = a) \cdot \ln \left[ \frac{\mathbb{P}_\theta^\pi (s^{t+1} = s' | s^t = s, a^t = a)}{\mathbb{P}_{\theta^j}^\pi (s^{t+1} = s' | s^t = s, a^t = a)} \right] \quad (211)$$

$$< 3 \cdot 2^{2n} \frac{\Delta^2}{\delta(d-1)^2}. \quad (212)$$

Further, note that we must consider only those cases where $s^t \neq g$ since otherwise $s^{t+1} = g$ with probability 1 for both instances $\theta$ and $\theta^j$ and corresponding $\ln$ terms are all 0. Using this and Equation (212) in Equation (202)

$$\text{KL}[\mathbb{P}_\theta^\pi (s^{t+1} | \bar{s}_t) || \mathbb{P}_{\theta^j}^\pi (s^{t+1} | \bar{s}_t)]$$

$$= \sum_{\bar{s}_{t-1} \in \mathcal{S}^{\times t-1}} \mathbb{P}_\theta^\pi (\bar{s}_{t-1}) \cdot \sum_{s \in \mathcal{S}, a \in \mathcal{A}} \mathbb{P}_\theta^\pi (s^t = s, a^t = a | \bar{s}_{t-1})$$

$$\times \left[ \sum_{s' \in \mathcal{S}} \mathbb{P}_\theta^\pi(s^{t+1} = s' | \bar{s}_{t-1}, s^t = s, a^t = a) \cdot \ln\left[ \frac{\mathbb{P}_\theta^\pi(s^{t+1} = s' | \bar{s}_{t-1}, s^t = s, a^t = a)}{\mathbb{P}_{\theta^j}^\pi(s^{t+1} = s' | \bar{s}_{t-1}, s^t = s, a^t = a)} \right] \right] \tag{213}$$

$$< \sum_{\bar{s}_{t-1} \in \mathcal{S} \times t-1} \mathbb{P}_\theta^\pi(\bar{s}_{t-1}) \cdot \sum_{s \in \mathcal{S}, a \in \mathcal{A}} \mathbb{P}_\theta^\pi(s^t = s, a^t = a | \bar{s}_{t-1}) \cdot 3 \cdot 2^{2n} \frac{\Delta^2}{\delta(d-1)^2} \tag{214}$$

$$= \mathbb{P}_\theta^\pi(s^t \neq g) \cdot 3 \cdot 2^{2n} \frac{\Delta^2}{\delta(d-1)^2} \tag{215}$$

Using above result in Equation (199), we have

$$\mathrm{KL}(\mathbb{P}_\theta^\pi || \mathbb{P}_{\theta^j}^\pi) = \sum_{t=1}^{T-1} \mathrm{KL}[\mathbb{P}_\theta^\pi(s^{t+1}|\bar{s}_t) || \mathbb{P}_{\theta^j}^\pi(s^{t+1}||\bar{s}_t)] \tag{216}$$

$$< \sum_{t=1}^{T-1} \mathbb{P}_\theta^\pi(s^t \neq g) \cdot 3 \cdot 2^{2n} \frac{\Delta^2}{\delta(d-1)^2} \tag{217}$$

$$\leq 3 \cdot 2^{2n} \frac{\Delta^2}{\delta(d-1)^2} \cdot \sum_{t=1}^{T} \mathbb{P}_\theta^\pi(s^t \neq g) \tag{218}$$

$$= 3 \cdot 2^{2n} \cdot \frac{\Delta^2}{\delta(d-1)^2} \cdot \mathbb{E}_{\theta,\pi}[N^-] \tag{219}$$

where, the final step follows from the definition of $N^-$. This completes the proof. $\qquad \square$

## C.9 Proof of Lemma 8

**Lemma 8.** *Take any agent $\tilde{i} \in [n]$ currently at node $s$. Then under any global action $a \in \mathcal{A}$, the probability that agent $\tilde{i}$ stays in node $s$ in the next step is lower bounded by $1/2$. Formally, if $s$ is such that $s_i = s$, then $\sum_{s' | s'_{\tilde{i}} = s} \mathbb{P}[s' | s, a] > 1/2$ for any $a \in \mathcal{A}$.*

*Proof.* Similar to Equation (90), for any $\tilde{i} \in [n]$, $s$ and such that $s_{\tilde{i}} = s$, then for any $a \in \mathcal{A}$ we have

$$\sum_{s' \in \mathcal{S}(s) | s'_{\tilde{i}} = s} \mathbb{P}(s' \mid s, a) \tag{220}$$

$$= \sum_{s' \in \mathcal{S}(s) | s'_{\tilde{i}} = s} \left( \sum_{i \in \mathcal{I}} \langle \phi_i(s'_i \mid s_i, a_i), (\theta_i, 1) \rangle + \sum_{j \in \mathcal{J}} \langle \phi_j(s'_j \mid s_j, a_j), (\theta_j, 1) \rangle \right) \tag{221}$$

$$= \sum_{s' \in \mathcal{S}(s) | s'_{\tilde{i}} = s} \left( \langle (-a_{\tilde{i}}, \frac{1-\delta}{n \cdot 2^{r-1}}), (\theta_{\tilde{i}}, 1) \rangle + \sum_{i \in \mathcal{I} \backslash \{\tilde{i}\}} \langle \phi_i(s'_i \mid s_i, a_i), (\theta_i, 1) \rangle + \frac{n-r}{n \cdot 2^r} \right) \tag{222}$$

$$= \sum_{s' \in \mathcal{S}(s) | s'_{\tilde{i}} = s} \left( \langle -a_{\tilde{i}}, \theta_{\tilde{i}} \rangle + \frac{1-\delta}{n \cdot 2^{r-1}} + \sum_{i \in \mathcal{I} \backslash \{\tilde{i}\}} \langle \phi_i(s'_i \mid s_i, a_i), (\theta_i, 1) \rangle + \frac{n-r}{n \cdot 2^r} \right) \tag{223}$$

$$= \left( 2^{r-1} \cdot \langle -a_{\tilde{i}}, \theta_{\tilde{i}} \rangle + \frac{1-\delta}{n} + \frac{2^{r-1}}{2} \cdot (r-1) \sum_{s'_i \in \{s, g\}} \langle \phi_i(s'_i \mid s_i, a_i), (\theta_i, 1) \rangle + 2^{r-1} \cdot \frac{n-r}{n \cdot 2^r} \right) \tag{224}$$

$$= \left( 2^{r-1} \cdot \langle -a_{\tilde{i}}, \theta_{\tilde{i}} \rangle + \frac{1-\delta}{n} + \frac{2^{r-1}}{2} \cdot (r-1) \cdot \frac{1}{n \cdot (2^{r-1})} + \frac{n-r}{2n} \right) \tag{225}$$

$$= \left( 2^{r-1} \cdot \langle -a_{\tilde{i}}, \theta_{\tilde{i}} \rangle + \frac{1-\delta}{n} + \frac{n-1}{2n} \right) \tag{226}$$

In Equation (222), we use the features from Equations (3) and (4). In the next step, we simplify the dot product. To obtain Equation (224), we observe that there are $2^{r-1}$ next states in which agent $\tilde{i}$

stays at node $s$. To simplify the middle summation, we observe that for any agent $i \in \mathcal{I} \setminus \{\tilde{i}\}$, it will appear in half of the next states of interest in node $s$ and half times in node $g$. In rest of the steps, we only rearrange terms and simplify.

Note that the above probability is minimized when optimal action $\boldsymbol{a}^*$ is taken, and hence $\langle -a_{\tilde{i}}^*, \theta_{\tilde{i}} \rangle = -\frac{\Delta}{n}$. Thus,

$$\sum_{\boldsymbol{s}' \in \mathcal{S}(\boldsymbol{s}) | s_{\tilde{i}}' = s} \mathbb{P}(\boldsymbol{s}' \mid \boldsymbol{s}, \boldsymbol{a}) \geq \sum_{\boldsymbol{s}' \in \mathcal{S}(\boldsymbol{s}) | s_{\tilde{i}}' = s} \mathbb{P}(\boldsymbol{s}' \mid \boldsymbol{s}, \boldsymbol{a}^*) \tag{227}$$

$$= \left( -2^{r-1} \cdot \frac{\Delta}{n} + \frac{1 - \delta}{n} + \frac{n - 1}{2n} \right) \tag{228}$$

$$\geq \left( -2^{n-1} \cdot \frac{\Delta}{n} + \frac{1 - \delta}{n} + \frac{n - 1}{2n} \right) \tag{229}$$

$$\geq \left( \frac{-1 + 2\delta}{6n} + \frac{6 - 6\delta}{6n} + \frac{3n - 3}{6n} \right) \tag{230}$$

$$= \left( \frac{3n + 2 - 4\delta}{6n} \right) \tag{231}$$

$$> 1/2 \tag{232}$$

In the above, we use our restriction on $\Delta$ and $\delta$. $\qquad\square$

## C.10 Proof of Corollary 2

**Corollary 2.** *For any non-goal state $\boldsymbol{s} \in \mathcal{S}_r$ of type $r$, the self-transition probability under action $\boldsymbol{a}$ is*

$$\mathbb{P}(\boldsymbol{s}|\boldsymbol{s}, \boldsymbol{a}) = \frac{r(1 - \delta)}{n \cdot 2^{r-1}} + \frac{n - r}{n \cdot 2^r} - \frac{r \cdot \Delta}{n} + \frac{2\Delta}{n(d - 1)} \sum_{i \in \mathcal{I}} \sum_{p=1}^{d-1} \mathbb{1}\{sgn(a_{i,p}) \neq sgn(\theta_{i,p})\}. \tag{16}$$

*This is minimized when $\boldsymbol{a} = \boldsymbol{a}_\theta$ and the resulting probability only depends on $r$.*

*Proof.* Putting $r' = r$ in the equation from Lemma 2 and using $\mathcal{T} = \phi$ gives the desired expression:

$$\mathbb{P}(\boldsymbol{s}|\boldsymbol{s}, \boldsymbol{a}) = \frac{r(1 - \delta)}{n \cdot 2^{r-1}} + \frac{n - r}{n.2^r} - r.\frac{\Delta}{n} + \frac{2\Delta}{n(d - 1)} \sum_{p \in [d-1]} \sum_{i \in \mathcal{I}} \mathbb{1}\{sgn(a_{i,p}) \neq sgn(\theta_{i,p})\} \tag{233}$$

Fixing any $\boldsymbol{s} \in \mathcal{S} \setminus \{\boldsymbol{g}\}$, fixes the first 3 terms. Only the summation over indicators varies for various actions $\boldsymbol{a}$. Since the indicator function can least be 0, the minimum probability for the above transition for any $\boldsymbol{s} \in \mathcal{S}$ is achieved when for each agent $i \in \mathcal{I}$, the signs of each component of its action $a_i$, matches with respective components of $\theta_i$. This definitely happens at $\boldsymbol{a} = \boldsymbol{a}_\theta$. $\qquad\square$

## C.11 Proof for $V_1^* > B^*/n$

By standard evaluation argument,

$$V_r^* = 1 + p_{r,r}^* . V_r^* + \sum_{r'=1}^{r-1} \binom{r}{r'} p_{r,r'}^* . V_{r'}^* \tag{234}$$

$$< 1 + p_{r,r}^* V_r^* + (1 - p_{r,r}^*) V_{r-1}^* \tag{235}$$

$$\Leftrightarrow V_r^* - V_{r-1}^* < \frac{1}{1 - p_{r,r}^*} \tag{236}$$

Thus,

$$V_n^* - V_{n-1}^* < \frac{1}{1 - p_{n,n}^*} \tag{237}$$

$$V_{n-1}^* - V_{n-2}^* < \frac{1}{1 - p_{n-1,n-1}^*} \tag{238}$$

$$\vdots$$

$$V_2^* - V_1^* < \frac{1}{1 - p_{2,2}^*} \tag{239}$$

$$V_1^* - V_0^* = \frac{1}{1 - p_{1,1}^*} \tag{240}$$

Adding the above inequalities,

$$V_n^* - V_0^* \leq \sum_{i=1}^{n} \frac{1}{1 - p_{i,i}^*} \leq \frac{n}{1 - p_{1,1}^*} = nV_1^* \implies B^*/n \leq V_1^* \tag{241}$$

The second inequality above is due to $p_{1,1}^*$ being the maximum over all $p_{i,i}^*$ (Using the Equation in Corollary 2, it's easy to show that $p_{r,r}^*$ decreases with $r$ for our instances). Observe that the final equality in Equation (241) holds for $n = 1$.

