# OpenReview forum: "Regret Lower Bounds for Decentralized Multi-Agent Stochastic Shortest Path Problems"
_NeurIPS.cc/2025/Conference — NeurIPS 2025 poster_

### Official Review · Reviewer_MLjP · 2025-06-15

**Clarity:** 4
**Significance:** 3
**Originality:** 4
**Rating:** 5
**Confidence:** 3

**Summary:**

The authors study decentralized multi-agent SSPs (Dec-MASSPs) under linear function approximation, where the transition dynamics and costs are represented using linear models.
Their main contribution is the first regret lower bound for this setting based on the construction of hard-to-learn instances for any number of agents.
I find this work very interesting!

**Questions:**

Page 1:
Can you please expand the discussion on SSP so that a non-expert can better relate to the problem you are studying?

Page 2:
Can you please elaborate on (iii)?
I think that this explanation is very handwavy :)

Page 3:
Can you please elaborate on the caption of Table 1?

Is there any related work from the field of computational complexity?

Page 5:
In Equation (2), why is there a negative term?
Why are you subtracting $K \cdot V^*(s_{\rm init})$?

Page 6:
Line 238:
This explanation is nice but I am a bit confused.
Why are there exponentially many policies that differ in just two components of the global action in a single state, etc.?

Page 7:
I do not understand Equation (6).
Why do you need to sum over all $r$?

Page 8:
Can you please explain how use Pinsker's inequality to bound the regret?

Why is $N_{i, j}(\theta)$ unbounded?

Page 9:
Can you please expand the future work paragraph? :)

**Ethical Concerns:**

["NO or VERY MINOR ethics concerns only"]

**Final Justification:**

The authors have effectively addressed my comments and concerns.

**Limitations:**

None.

**Paper Formatting Concerns:**

None.

**Quality:**

3

**Strengths And Weaknesses:**

Strengths:
The problem studied is central in decentralized RL.

Weaknesses:
See Questions.

---

> ### Author Rebuttal · Authors · 2025-07-31
>
> *We thank Reviewer for their detailed and constructive feedback. We greatly appreciate the positive assessment of our work. Your thoughtful comments have helped us improve both the clarity and completeness of the paper. Below, we respond to each of your points.*
>
> **Page 1: Discussion on SSP for non-experts:** Sure we will add some more discussion on SSP for the non-expert.
>
> **Page 2: elaboration of (iii):** Unlike the single-agent case with two nodes (Min et al. (2022)), where a closed-form expression for optimal value function can be derived exactly for the two states, extending this to the multi-agent setting is significantly more challenging due to the exponential growth of the state space. While we initially attempted to derive a closed-form expression for the optimal value function, it turned out to be complex to work with analytically. A key insight that enabled progress was partitioning the state space based on the number of agents at each node. This structure helped us establishing the  monotonicity property of the optimal value function with respect to the number of agents at node $s$. This property played a central role in our lower bound analysis and allowed us to proceed without requiring an explicit optimal value function. The details are provided in Theorem 1 and proof of Lemma 5.
>
> **Page 3: caption of Table 1** Sure, we will add more details in the table to make it self-contained. For example, we will consider including the role of $n, d$ and $B^*$ in the table in final version. We are not aware of any related work in the field of computational complexity for the MASSP settings. We would appreciate it if you could point out any aspects we may have missed in this regard.
>
> **Page 5: regret definition, equation (2):** This term arises naturally from the definition of regret in the cost setting (Bandit Algorithms. Tor Lattimore and Csaba Szepesvari). We aim to study the expected cumulative cost an algorithm incurs in excess to that of the optimal policy over $K$ episodes, and its scaling with $K$. In the regret definition (Equation 2) $V^*(s_{init})$
>
> denotes the expected cost per episode under the optimal policy. Therefore, the total expected cost of the optimal policy over $K$ episodes is $K \cdot V^*  (s_{init})$.
>
> **Page 6: line 238, exponentially many near-optimal policies:** We agree that this point could have been made more explicitly. Our intent was to highlight that the number of policies differing from the optimal one in the specified manner grows faster than exponentially in $n$, the number of agents. Specifically, there are $2^n - 1$ non-goal states where a policy may deviate from the optimal. In each such state, there are  $n(d-1)$ components of the global action, and a deviation at any two of these components yields a different policy of this kind. This results in $(2^n-1) \times \binom{n(d-1)}{2}$ such policies, which scales as $n^2 \times 2^n$ in number of agents, $n$. Thus, the number of such policies grows super-exponentially with $n$.
>
> **Page 7: equation (6), summation over $r'$ :** Please note that the summation is over $r'$ not $r$. We recall $r$ is the number of agents at node $s$ in global state $\textbf{\textit{s}}$, whereas $r'$ is a running index representing the similar number for the reachable states from $s$. The summation over $r'$ is used to take care of all the possible types of next states. The goal state and the current state are explicitly included in the equation (6), while the double summation ensures that all other reachable states are also considered. Specifically, the outer summation over all $r'$ captures all other reachable state types, and for each fixed $r'$, the inner summation over all $\textbf{\textit{s}}' \in \mathcal{S}_{r'}(\textbf{\textit{s}})$ ensures that every state of type $r'$ that is reachable from $\textbf{\textit{s}}$ is also included in the summation. This structure guarantees completeness in covering all valid successor states in the transition dynamics.
>
> **Page 8: Pinsker's inequality:** Lemma E.3 (on page 25) is directly borrowed from the related work of Min et al.(2022), where it is obtained using Pinsker’s inequality. In our work, we apply Pinsker’s inequality to bound the KL divergence between probability distributions over sample paths of two closely related instances, denoted by $\theta$ and $\theta^j$, as defined in lines 323–324. Further details of this KL divergence and its role in the proof can be found on page 25.
>
> **Regarding unboundness of $N_{i,j}(\theta)$:** While it is not always unbounded, we must account for the possibility of it being unbounded as shown below. In the proof of Theorem 2, we consider an arbitrary decentralized learning algorithm and it is possible that such an algorithm selects actions for which $sgn(a^t_{i,j}) \neq sgn(\theta_{i,j})$ for some $(i,j)$ and for all $t$ (e.g., this can occur in trivial or poorly designed algorithms). Furthermore, certain sample paths may not terminate, for instance, the trajectory  $s_{init}, s_{init}, s_{init}, ...$ which would lead to unbounded $N_{i,j}(\theta)$. To maintain generality and correctness, we carefully consider such cases in our analysis by defining $N^-_{i,j}(\theta)$,
>
> a truncated version of $N_{i,j}(\theta)$.
>
> We also note that similar techniques to apply Pinsker's inequality and handling the unboundedness of the analogous quantity have been applied in the single-agent setting in Min et al. (2022).
>
> **Page 9: future work paragraph:** Sure, we will add a few more things like capacitated SSP, and multi-commodity problems which posses additional learning challenges.
>
> *We hope our responses have addressed your concerns and clarified the key contributions. We would be glad to engage further during the discussion phase if needed. Given the significance of the results and the improvements made, we would appreciate it if you could consider raising your score.*

---

### Official Review · Reviewer_YLTg · 2025-06-23

**Clarity:** 1
**Significance:** 2
**Originality:** 3
**Rating:** 4
**Confidence:** 3

**Summary:**

The paper examines a regret lower bound for a class of multi-agent stochastic shortest path problems. The general model assumes that $n$ agents are either in the starting or the goal state, and a linearly defined transition function stochastically allows this transition based on an action which is a binary vector of dimensionality $d$ over $\{-1,1\}$. This implies that the optimal policy is given by letting each agent select the action minimizing the likelihood of staying in the starting state. Though the cost is uniform for each action, the transition function is dependent on how many agents are still in the starting state.
The authors prove two main theorems for their setting: The first basically shows that the optimal policy should constantly select the action for each agent which is maximizing the likelihood of transitioning to the goal. Furthermore, due to the homogenous and independent behaviour of agents, states having the same number of agents positioned on the starting state share the same values. Furthermore, as agents may eventually transit to the goal, the cumulated expected future cost decreases.
In the second theorem, the authors compute a lower bound for the expected regret based on this optimal behaviour.

**Questions:**

Q1. Your main claim seems to be Theorem 2, presenting the lower bound. What are the assumptions on the algorithm $\pi$ that is lower bounded? There are some hints in the proofs, but even from the sketch in Section 5, it remains unclear why the algorithm cannot estimate the transition likelihood of actions any quicker.

Q2. To what extent are LM-MASSP problems indicative of general MASSP problems as defined in the related work?

Q3. Could your results easily be extended to a larger number of nodes beyond start and goal?

**Ethical Concerns:**

["NO or VERY MINOR ethics concerns only"]

**Final Justification:**

The authors adequately addressed my concerns in their rebuttal. Thus, I raised my score and recommend accepting this paper.

**Limitations:**

The authors describe the main assumptions of their model and stress that the linear transition function is a limiting factor.

**Paper Formatting Concerns:**

I did not find any formatting issues.

**Quality:**

3

**Strengths And Weaknesses:**

The paper examines whether, in decentralized multi-agent stochastic shortest path problems with linear cost and transition functions, the already proposed methods might already have an optimal complexity. This is interesting as it answers the question of how well-known algorithms are performing compared to optimal policies and if further research in this direction is required.
The idea of finding instances of this problem setting where the optimal policy can be derived and then showing that no algorithm can achieve faster convergence towards the optimal policy also seems feasible.

A drawback of the paper is that it rarely tries to give any intuition behind the assumptions, the design of the problem, or even about the assumption on the actual algorithms, which are lower bounded in the end. Thus, trying to get behind the setting requires a lot of time, and it feels that the authors could have made it far easier for the reader to get an understanding of what is proposed in the paper.

Another drawback of the paper is that the contribution feels somewhat limited as the setting is rather specific. Thus, it is unclear why the proposed setting is indicative of the broader settings from the related work.
Though the authors define a problem that allows deriving a lower bound, the authors do not advertise the practical relevance of the setting. Also, it is not clear to what extent the results are transferable to a larger set of MASSP problems. Thus, the results may not be interesting to the broader machine learning community.

To conclude, the paper examines a very complicated setting from a theoretical point of view by finding and analyzing an instance that allows tractable results. Concerns are that the paper is hard to follow due to a lack of providing intuitions and that the assumptions required to derive  the results are too restrictive to raise attention in the broader machine learning community.

---

> ### Author Rebuttal · Authors · 2025-07-31
>
> *We sincerely thank the reviewer for their thoughtful comments and feedback. We are encouraged by their engagement with our work and provide clarifications below to address the points raised.*
>
> **Q1: Assumptions on algorithm $\pi$:** Thank you for the question. Our goal is to characterize the fundamental limits of LM-MASSPs, and as such, we make no assumptions on the algorithm. Our lower bound is universal in the sense that it applies to any algorithm, including those that estimate transition dynamics. Theorem 2 is proven using information-theoretic techniques to show that no algorithm can reliably distinguish between carefully constructed, hard-to-learn instances better than $O(\sqrt{K})$ in number of episodes. We elaborate on this intuition in the paragraph titled "Why do we expect tight lower bounds for these instances?" on page 6. We also emphasize that our only structural assumption is the linear function approximation. No additional assumptions about the learning algorithm are required for the lower bound to hold.
>
> **Q2: LM-MASSP vs. MASSP:** We focus on LM-MASSP to stay consistent with prior work on regret upper bounds, which also assumes linear function approximation (Trivedi and Hemachandra 2023). LM-MASSP are representative of MASSP as far as linear function approximations are concerned. To emphasize this linear function approximation assumption and its role we explicitly refer to the setting as LM-MASSP.
>
> **Q3: Extension to three or more nodes:** We construct hard-to-learn instances using a two-node setup and establish valid regret lower bounds. Despite the minimal structure, the state space grows exponentially with the number of agents, making the problem highly non-trivial. A key contribution of our work is demonstrating that this simple setup suffices to capture the core difficulty of decentralized learning and to derive tight lower bounds.
> While we focus on two nodes for clarity and tractability, we do not rule out the possibility of extending the construction to three or more nodes. Such extensions would introduce additional technical challenges, particularly in identifying suitable hard instances and characterizing the optimal policy and value function while preserving the lower bound structure. We expect the regret lower bound to scale similarly with $K$, with possible changes only to constant factors. We leave this as a promising direction for future work.
>
> We will now address the points raised in the **Strengths And Weaknesses** section of the review.
>
> **Intuition/restrictiveness of assumption:** We emphasize that the *only* assumption made throughout our work is the linear function approximation of the transition dynamics, a standard and widely adopted assumption in both single-agent and multi-agent RL literature (e.g., Min et al., (2022); Zhang et al., (2018)).
>
> Throughout the paper, we provide supporting intuition and background to motivate our approach. For instance, in constructing hard-to-learn instances and establishing the tightness of the regret lower bound, we highlight that these instances are designed to admit exponentially many near-optimal policies which are indistinguishable making the learning fundamentally hard and leading to $O(\sqrt{K})$ regret scaling (see lines 238–246). Additionally, we outline the key proof steps with underlying intuition, which resulted in the technical lemmas. We will expand on these aspects further in the final version.
>
> **Contributions of the paper:** We provided an itemized summary of our contributions on page 2. To reiterate, our key contributions include: (i) identifying a suitable class of hard-to-learn instances, (ii) carefully designing the feature representations, (iii) uncovering structural properties of the optimal value function, and (iv) rigorously proving the regret lower bounds. Each of these steps involved substantial technical challenges, which we addressed in detail. We hope the reviewer recognizes the depth and significance of our contributions.
>
> **Practical relevance of the setting:** The Stochastic Shortest Path (SSP) problem is a foundational and widely applicable model in the MDP literature. As noted in Neuro-Dynamic Programming (Bertsekas & Tsitsiklis, 1996), many MDPs can be cast as SSP problems. Moreover, real-world applications such as board games (e.g., Go, with random play durations) are naturally modeled as SSPs. The learning version of SSPs has received recent attention (Tarbouriech et al., 2019; Cohen et al., 2020).
>
> A more realistic extension is the multi-agent SSP (MASSP) framework, which captures practical scenarios such as multiple vehicles sharing a road network, leading to congestion effects. These settings, where independently acting agents share costs or information, introduce additional complexity. While Trivedi & Hemachandra (2023) propose an algorithm for MASSP and provide upper bounds on regret, the fundamental lower bounds remained unaddressed.
>
> Our work fills this gap by constructing a carefully designed 2-node hard-to-learn MASSP instance and establishing that no multi-agent algorithm can achieve regret better than $O(\sqrt{K})$ under linear transition assumptions. This construction extends the 2-node hard instance used in single-agent SSP lower bounds (Min et al., 2022) to the multi-agent setting.
>
> We believe our results offer meaningful insights into the fundamental limits of learning in MASSP under linear assumptions and may inspire future lower bound analyses in other multi-agent settings. We welcome further feedback to clarify any aspect of our contributions or their relevance.
>
> *We hope our responses adequately address your concerns and help clarify the key contributions and technical depth of our work. If you find the clarifications convincing, we kindly ask you to consider raising your score. We would also be happy to engage further during the author–reviewer discussion phase to elaborate on any remaining questions.*

---

> > ### Comment · Reviewer_YLTg · 2025-08-01
> > **thank you for the clarifications**
> >
> > Thank you for clarifying all my questions. I will reconvene with the other reviewers and rethink my recommendation.

---

> > > ### Author Response · Authors · 2025-08-04
> > > **Thank you for reconsidering the recommendation**
> > >
> > > Thank you very much for your thoughtful review and for taking the time to consider our clarifications. We appreciate your willingness to revisit your recommendation. We hope the further discussions will lead to a positive reconsideration and an increase in the score. Please let us know if there are any additional points we can address during the discussion.
> > >
> > > Thank you.

---

### Official Review · Reviewer_jnXP · 2025-06-28

**Clarity:** 4
**Significance:** 4
**Originality:** 4
**Rating:** 6
**Confidence:** 5

**Summary:**

This paper studies the fundamental learning limits in decentralized multi-agent stochastic shortest path (Dec-MASSP) problems with linear function approximation—a setting relevant to areas like swarm robotics and traffic routing, but still theoretically underexplored. While single-agent SSPs have been well explored in the existing literature, extending regret bounds to decentralized multi-agent cases poses major challenges due to exponential state-action spaces and coupled dynamics.

To this end, the paper establishes the first ever regret lower bound for Dec-MASSPs, proving a bound of Ω(√K) over K episodes. This matches known upper bounds and recovers the single-agent case when the number of agents is one, providing a tight characterization of the problem's learning complexity. Their analysis is based on a technically novel construction of hard-to-learn problem instances with linearly parameterized dynamics and costs.

To overcome key technical barriers, the paper designs feature representations that yield valid transitions, simplify value function analysis using state partitioning and monotonicity, and bound KL divergence via symmetry in agent distributions. These techniques enable valid analysis despite the exponential state space. Overall, the paper closes a critical gap in theory and offers a foundation for evaluating and designing scalable learning algorithms in decentralized multi-agent systems.

**Questions:**

1. The setting is fully decentralized learning without communication. I wonder how limited communication might affect the regret bounds and instance hardness? Or am I missing something here?


2. Just curious but I wonder what might happen if we allow some model misspecification? The model assumes exact linear approximation. How sensitive are the lower bounds or constructions to small model misspecifications?

**Ethical Concerns:**

["NO or VERY MINOR ethics concerns only"]

**Final Justification:**

I thank the authors for their response to my questions.
Overall, I think this is a very good paper.
I definitely vote for an accept.

**Limitations:**

Yes

**Quality:**

3

**Strengths And Weaknesses:**

**Strengths**

**Quality**

1. Technically rigorous: The paper presents a careful and comprehensive construction of hard instances for Dec-MASSP, supported by well-structured theoretical arguments and inductive proofs.

2. Tight lower bound: The regret lower bound matches the known upper bounds up to poly-log factors, demonstrating tightness and validating the authors' analytical framework.

3. The authors skillfully handle several complex issues — exponential state space, coupled agent dynamics, and KL divergence — using symmetry and feature design to make the analysis doable.

**Clarity**

1. Well-motivated research topic: The paper clearly identifies the knowledge gap in decentralized SSPs with function approximation and motivates the need for lower bounds.

2. Each section is logically laid out, with clear transitions from the problem setup to methodology, theory, and discussion.
Notions like state-type partitions, the monotonicity of value functions, and symmetry-based analysis are introduced and explained in a way that highlights their intuitive appeal and theoretical usefulness.

3. A very clear proof sketch: This is a hard paper. But the proof sketch makes it easier for the readers to follow. The proof sketches are clear, well-motivated, and the full proofs (relegated to appendices) build on and extend techniques from prior work thoughtfully.


**Significance**

I think this paper fills in an important gap in the research direction of SSP and multi-agent RL.

1. This is the first known regret lower bound in the decentralized SSP setting with linear function approximation, establishing a benchmark for future algorithms in this research area. The paper recovers the classical single-agent bounds as a special case, and thus provides a generalization of prior SSP results to the decentralized setting.

2. Importantly, the paper provides a critical insight for understanding the complexity of decentralized MARL. Specifically, the paper reveals that increasing the number of agents increases the density of near-optimal policies, making learning fundamentally harder.


**Originality**

1. The hard instances are new and cleverly designed to yield analytical tractability despite exponential state-action space.

2. Novel technique challenges: The use of symmetry to simplify the KL divergence analysis and characterize optimal actions is a clever extension of classical information-theoretic techniques to a more complex setting.

3. First result in the field: The tight lower bound for any number of agents in Dec-MASSP with linear approximation is a good contribution in MARL theory.


**Weaknesses (Minor and Contextual)**

I do not detect any major technical weakness in the paper. Following are just some minor comments.

**Quality**

As a purely theoretical work, the use of very specialized assumptions (e.g., uniform cost, linear transition features, parameter restrictions on $\delta$) may limit the practicability of the lower bounds beyond the theory.

**Significance**

The impact is currently limited to linear function approximation. While the paper acknowledges this, broader significance will depend on whether similar bounds can be extended to more general function classes (e.g., neural networks).

**Overall**, I deem this as a high-quality and technically sound contribution that meaningfully contributes to our understanding of decentralized stochastic shortest path. Its careful lower bound construction and insights into the hardness of multi-agent learning problems position it as a strong candidate for acceptance, particularly in theory-focused tracks or venues.

---

> ### Author Rebuttal · Authors · 2025-07-31
>
> *We thank Reviewer for their detailed and constructive feedback. We greatly appreciate the positive assessment and the high confidence in our work. Your thoughtful comments have helped us improve both the clarity and completeness of the paper. Below, we respond to each of your points.*
>
> **Limited communication:** Thank you for the thoughtful question. We intentionally do not assume any specific communication structure among agents in order to keep our results general. As such, the lower bounds we derive are valid under any communication among agents, including fully decentralized, partially connected, or even centralized settings. This generality allows our results to serve as fundamental performance limits across a broad range of scenarios. That said, studying how different levels of communication might affect instance hardness or improve lower bounds (in factors other than $K$) is indeed a compelling  future direction, and we appreciate the suggestion.
>
> **Model misspecification:** Thank you for the insightful question. Our work, like prior literature Min et. al. (2022), Zhang et. al. (2018) assumes exact linear function approximation (LFA) and focuses on the LM-MASSP setting. The impact of model misspecification is largely unexplored in this context. While we cannot currently comment on the tightness of our bounds under misspecification, it is a promising future direction. Notably, the existing upper bounds also assume exact LFA and would require similar re-evaluation.
>
> *We hope that our clarifications and improvements adequately address your concerns. We would be happy to further engage during the author–reviewer discussion phase to elaborate on any remaining issues. Given the significance of the contributions and the enhancements made in response to your feedback, we kindly ask you to consider increasing your score if you feel it is appropriate.*

---

> ### Comment · Reviewer_jnXP · 2025-08-04
> **Response to rebuttal**
>
> I would like to thank the authors for their detailed rebuttal, regarding model misspecification and agent communication. They addressed my questions. I have also read the comments from the other reviewers and the response from the authors as well. It seems to me that a large proportion of the questions are well addressed. I will take all these into consideration during the discussion period with other reviewers. For now, I maintain my original scoring of 5 for 'acceptance'.
> Thank you!

---

> > ### Author Response · Authors · 2025-08-04
> > **Response to comment**
> >
> > Thank you for taking the time to review our rebuttal. We’re happy to hear that our responses addressed your questions. We will consider including them in the final version. We value your continued engagement and thank you for maintaining your score.
> >
> > Thank you.

---

### Official Review · Reviewer_Z1Bz · 2025-07-03

**Clarity:** 2
**Significance:** 3
**Originality:** 3
**Rating:** 4
**Confidence:** 4

**Summary:**

This paper characterizes the theoretical lower bound on the regret of the decentralized stochastic shortest path learning problem, which is a crucial problem in path planning. The authors consider decentralized learning in multi-agent state and action spaces. There are three major contributions: (1) designing a class of hard-to-learn MASSP problems and theoretical characterization of the optimal policy and value functions; (2) leveraging that class of problems, a tight lower bound is proposed; (3) Some technical intermediate results on the exponential state-action space would be valuable for future MASSP studies. Overall, the paper appears to be the first work to propose tight lower bounds for MASSP problems, and therefore is a valuable contribution to the field.

**Questions:**

This work considers multi-agent cases, so would the derived lower bound recover existing lower bound results for the single-agent shortest path problem? If so, I believe the constructed hard-to-learn example will also have an impact on the single-agent case.

The constructed lower bound relies on the strong assumption that the optimal policy can be easily found. However, for the general case where the optimal policy is hard to find, would the lower bound still be tight?

The constructed lower bound in Theorem 2 depends on B^*, but the reported lower bound of this work in Table 1 has no dependency on B^*. Why is B^* missing in the last line of Table 1?

What is the technical challenge when extending the proposed technique to random policies?

The constructed regret lower bound seems to rely on the assumption that the value function can be estimated accurately. In practice, it may not be the case. So, how does the value function approximation error affect the regret?

In many theoretical RL papers, we assume all states and actions should have some probability mass in the transition dynamics to avoid the poor exploration issue. Do we need similar assumptions in this work? I believe so, because otherwise the denominator in equation (15) can be zero when we have deterministic transition dynamics.

**Ethical Concerns:**

["NO or VERY MINOR ethics concerns only"]

**Final Justification:**

I appreciate the reviewer’s detailed follow-up discussion. I agree with the authors that empirically comparing an algorithm’s actual computational complexity to its theoretical lower bound can be challenging. Nevertheless, if space permits in the camera-ready version, I recommend including a discussion of these challenges and possibly noting this as an avenue for future work. A clear gap between empirical performance and the theoretical lower bound would call into question the theoretical claim that “the derived regret bound is tight.” More broadly, I believe empirical validation of theoretical bounds is valuable, as it provides a practical check on the reasonableness of the assumptions underlying the analysis. Such validation ultimately brings theoretical work closer to practical problems and increases its potential impact on actual algorithm design. Given the explanation and the challenges described by the authors, I am raising my score to borderline accept.

**Limitations:**

Yes, the authors provide some discussion about the limitations, such as only considering linear feature spaces instead of nonlinear feature spaces. However, the reviewer believes that there are two additional limitations to the current manuscript. Having some discussion about them will significantly improve the impact of this work.

No experiment comparing the actual computation time of different algorithms for solving those hard-to-learn examples. The result would be more convincing if a plot could be provided for visualizing that the lower bound regret is indeed tight and valid, compared to the existing algorithms’ regrets.

In many real-world planning problems, we need to consider road capacity and other types of constraints. The current formulation has no practical constraints. So, looking ahead, a natural question is how constraint satisfaction affects the computational lower bound? Some discussion on this in future work would be exciting and make the work more impactful for real-world constraint-aware shortest-path problems.

**Paper Formatting Concerns:**

No issues.

**Quality:**

2

**Strengths And Weaknesses:**

Strengths:
The high-level two-step strategy of proving the theoretical lower-bound result is technically sound and original.

The constructed worst-case MDP is interesting. It can motivate more research on characterizing which class of linear MDP would have an even tighter lower bound than the one proposed in this paper, and therefore can be an impactful contribution for helping the readers understand the interplay between linear MDP structure and computational complexity of learning the optimal policy.

The linear MDP approach generalizes prior works on tabular MDP, with compact and efficient feature representation.

The four technical challenges of the multi-agent shortest path problem are clearly discussed on page 2. The paper employs various strategies to address these challenges.


Weaknesses:
Assumption 1 can be made more informative. In general, the theoretical lower bound of computational complexity relies on the specific information provided for the algorithm. However, the current version of Assumption 1 only specifies the linear MDP setup. A more informative assumption should also include information on how decentralized learning is implemented. What information can different agents share while learning?

Some technical details are unclear. For example, this paper motivates the problem by discussing the shortest path over a graph, which typically includes a set of nodes and a set of edges; however, the paper only introduces the node set. Moreover, it is unclear whether the paper considers a discrete or continuous state space. In other words, do we allow state or action to only take discrete values, or can they take continuous values?

There is a concern that some technical proofs may have a logic gap, and therefore, it questions the quality and credibility of the proofs. In the equation (45) of Lemma 5’s proof in the supplementary material, why can we aggressively take the sum of h from h = 1 to h = infinity?
This part seems not very clearly explained, and it may be a bold step. Technically, I’m not sure whether it’s correct.

There appears to be no discussion about how frequently the constructed MDP would be encountered in practical applications. If it never happens, then the impact of those constructed examples is low.

The technical details are hard to follow, and limited intuition is shared behind the derivation of proofs. So, it is hard to fully justify the technical correctness of the proposed idea. For example, it would greatly improve the readability if the authors could explain how the binomial coefficients help construct the lower bound.

Many typos. For example, line 28, “recent works has”; line 252, “its desirable to”; line 320, “upto”, etc.

---

> ### Author Rebuttal · Authors · 2025-07-31
>
> *We sincerely thank the reviewer for their thoughtful comments and valuable feedback on our submission. We provide clarifications below on the questions raised.*
>
> **Single agent case:**  Yes, our lower bound results and hard-instance construction readily recover the single-agent case studied by Min et al. (2022). While this connection is noted in lines 272–273, it was not made explicit. We will clarify it in the final version and provide the relevant details.
>
> **Role of optimal policy:** The lower bound does not depend on the ability to find the optimal policy. It reflects the fundamental learning hardness in LM-MASSP setting which we show by constructing the hard-to-learn instances based on proposed feature design (Equations 3 and 4). We show that for any algorithm, there exists a hard-to-learn instance for which the regret is at least the stated bound (the bound need not hold for all the instances). So, we do not make any assumption on how easily the optimal policy can be found. Thus, our lower bounds remain tight irrespective of the whether the optimal policy is hard to find or not.
>
> **Absence of $B^*$ in Table 1:** We appreciate this observation. Table 1 highlights the leading-order dependence on the number of episodes $K$, which is the primary focus in characterizing regret scaling. Constants and instance-dependent terms such as $n,d$, and  $B^*$ are omitted for clarity, but they do appear explicitly in the formal statements (e.g., Theorem 2). We will consider including them in the table in final version.
>
> **Randomized policies:** Extending our results to randomized policies is a valuable direction and aligns with standard theoretical frameworks. As noted in lines 181–182, such an extension is conceptually straightforward and primarily introduces notational overhead.
>
> For example, in the deterministic case, expressions like $\pi(s^t)$ (e.g., in Equation 219) directly specify the action taken in state $s^t$. Thus, $\mathbb{P}(g|s,\pi(s^t))$ is well-defined. In the randomized setting, we instead would need to define
>
> $$
> \mathbb{P}(g|s, \pi(s^t)) = \sum_{a\in\mathcal{A}} \mathbb{P}(g|s, a) \cdot \pi(s^t,a),
> $$
> where $\pi(s^t,a)$ denotes the probability of selecting action $a$ in state $s^t$ as specified by the randomized algorithm.
>
> Moreover, by Yao’s minimax principle (Cohen et. al. (2020)), the expected regret of the best randomized algorithm on the worst-case instance is lower bounded by that of the best deterministic algorithm on a randomized instance. Hence, our lower bound results naturally extend to randomized policies as well.
>
> **Role of value function approximation:** Our lower bound analysis does not rely on value function approximation. Instead, we construct hard instances within the LM-MASSP class that yield tight regret lower bounds without explicit approximation. Key to our approach is the monotonicity of the optimal value function with respect to the number of agents at the initial state (Theorem 1) which allows us to bypass approximation entirely. In settings where such structure is absent, approximation errors may become significant and impact regret. Analyzing how these errors scale with the number of episodes is an important direction for future work.
>
> **Equation (15):** Our feature design ensures that the denominator in Equation (15) will not be zero, since for $r=1$ we have $\mathbb{P}(s|s,a) \neq 1$. Thus, the associated transition probabilities are strictly between 0 and 1, ensuring the expression is well-defined.
>
> **Regarding exploration assumptions:** These are indeed important when designing algorithms and analyzing their upper bounds. However, our focus is on lower bounds, where we construct hard instances to show that, for any algorithm regardless of its level of exploration (e.g. greedy to fully exploratory), the regret lower bound still holds. As a result, we do not require exploration assumption.
>
> Next, we address the **weaknesses and limitations** raised in the review.
>
> **Assumption 1 and information sharing:** We intentionally avoid assuming a specific communication structure to ensure broad applicability. As a result, our lower bounds hold across fully decentralized, partially connected, and centralized settings, capturing fundamental performance limits. Exploring how different communication levels affect instance hardness or refine the bounds (beyond the $K$ dependence) is a promising direction for future work.
>
> **Discrete state and action space:** We clarify in Section 2.1 (lines 121–129) that both state and action spaces are discrete and finite, and our analysis is based entirely on this setting. Extending the results to continuous spaces is an interesting future direction.
>
> **Equation (44):** We thank the reviewer for their time and for carefully examining the technical details. We would be happy to discuss any concerns regarding the correctness of the proofs in greater depth.  Specifically we respond to the concerns about the missing details between Equation (44) and (45) in Appendix B. First note that Equation (44) is not the part of Lemma 5, rather it is part of the proof of Theorem 2. Recall that
>
> $\mathbb{E}_{\theta,\pi} [R_1]$
>
> $= V^{\pi}(\textbf{s}^1) - V^{\pi^*_{\theta}} (\textbf{s}^1) $
>
> $\geq \mathbb{E}_{\textbf{a}^1 \sim \pi(\textbf{s}^1), \textbf{s}^2 \sim \mathbb{P}(.|\textbf{s}^1,\textbf{a}^1)} [ V^{\pi}(\textbf{s}^2) - V^*(\textbf{s}^2) ]$
>
> $\quad +\mathbb{E}_{\textbf{a}^1 \sim \pi(\textbf{s}^1)} [ \frac{2 \Delta}{n(d-1)}$
>
> $\quad \times \sum_{i=1}^n \sum_{p=1}^{d-1} \mathbb{1}[\{\textbf{s}^1 \neq \text{goal}\} ] \mathbb{1} [\{sgn(a_{i,p}^1) \neq sgn(\theta_{i,p})\}] V^*(\textbf{s}^1) ]$.
>
> This introduces a recursive structure in the regret: the difference at time 1 is written in terms of the same difference at time 2 plus another term that depends on $s^1$.  Although $\textbf{s}^1$ is deterministic, $\textbf{s}^2$ is stochastic due to transitions under $\pi$ and $P_{\theta}$. Fixing a sample path of length $H$ (i.e., goal is reached at step $H$), and unrolling the recursion across all steps, we obtain
>
> $V^{\pi}(\textbf{s}^1) - V^{\pi^*_{\theta}}(\textbf{s}^1)$
>
> $\geq E_{1} (\sum_{h=1}^H \frac{2\Delta}{n(d-1)} \times \sum_{i=1}^n \sum_{p=1}^{d-1} \mathbb{1} (\{\textbf{s}^h \neq \text{goal} \} ) \mathbb{1} (\{sgn(a^h_{i,p}) \neq sgn(\theta_{i,p})\}) V^*(\textbf{s}^h) $,
>
> where the expectation is over all sample paths in episode 1.
>
> Now observe that, as per our model, once an agent reaches the goal node, it remains there for the rest of the episode. This allows us to safely extend the summation to infinity without altering its value. For all $h>H$, the state is specified (by convention) to be goal for that episode i.e., $s^{h,k} = goal$, making the indicator term $\mathbb{1} (\{\textbf{s}^h \neq \text{goal} \})  = 0$. Specifically, we apply the following identity:
>
> $\mathbb{E} \left[ \sum_{t=1}^{\tau} x_t \right] =  \mathbb{E} \left[  \sum_{t=1}^{\infty} x_t \mathbb{1}(\{t\leq \tau \}) \right] = \sum_{t=1}^{\infty} \mathbb{E} \left[  x_t \mathbb{1}(\{t\leq \tau \}) \right]$
>
> where $\tau$ and $x_t$ are random variables. Using this we get
>
> $V^{\pi}(\textbf{s}^1) - V^{\pi^*_{\theta}}(\textbf{s}^1)$
>
> $\geq E_{1} (\sum_{h=1}^{\infty} \frac{2\Delta}{n(d-1)} \sum_{i=1}^n \sum_{p=1}^{d-1} \mathbb{1} (\{\textbf{s}^h \neq \text{goal} \}) \mathbb{1} (\{sgn(a^h_{i,p}) \neq sgn(\theta_{i,p})\})  V^*(\textbf{s}^h))$
>
> Now using V^* $(s^h) \geq V^*_1$ we get Equation (45).
>
> **Occurance of the instances:**
> Thank you for raising this important point. Our goal is to establish fundamental lower bounds by constructing hard-to-learn instances that illustrate worst-case limitations in MASSP problems. These instances are not meant to reflect typical real-world MDPs but to capture theoretical hardness, as is standard in prior work (e.g., Min et al., 2022; Trivedi & Hemachandra, 2023). Exploring how such hard instances or their variants manifest in practical settings is indeed a valuable future direction.
>
> **Binomial coefficients:** While the binomial coefficients were not explicitly introduced, they arise naturally from the combinatorial structure of the two-node instance. For example, in Lemma 3, they quantify the number of ways in which r+1-r' out of r+1 agents move from the initial node to the goal node. Importantly, Lemma 3 is a key technical insight that underpins our proof strategy. It plays a central role in establishing Lemma 4 (supporting the monotonicity claim in Theorem 1) and Lemma 8 (see page 46), which is used to prove Lemma 6 and ultimately Theorem 2.
>
> **Typos:** Thank you for pointing these. We will take care of these and others, if any.
>
> **Lower Bound Visualization:** The lower bounds are existential in nature, i.e., they assert that no algorithm can perform better than a certain regret scaling in the worst case. Unlike upper bounds, they do not correspond to a specific algorithm or instance. In our setting, identifying the hard instance (among the exponentially many possibilities) for any fixed algorithm is itself non-trivial. Simulating or plotting regret curves against the lower bound would require checking all $2^{n(d-1)}$ instances, which is computationally infeasible. Moreover, lower bounds are not algorithm-specific; they hold universally across all possible algorithms. Even in classical single-agent SSP settings, despite the availability of many algorithms such plots are not shown, for the same reason.
>
> **Future work on constraint aware SSP:** We agree that this is a valuable and interesting future direction, and we will include a brief discussion on it in the final version under future work.
>
> *We hope these clarifications address the reviewer’s concerns. We would be grateful if the reviewer could consider increasing their score in light of these clarifications and the contributions of our work. We would also be happy to engage in further discussion during the  reviewer-author discussion period to clarify any additional questions or suggestions.*

---

> > ### Comment · Reviewer_Z1Bz · 2025-08-06
> >
> > I would like to thank the authors for their detailed rebuttal. They have addressed some of my concerns, and I have also read the comments from the other reviewers as well as the authors’ responses.  However, my primary concern remains unaddressed: there is still no empirical validation comparing the actual computation time of different algorithms on the hard-to-learn examples. I believe such an empirical study is critical to substantiate the claimed advantages.

---

> > > ### Author Response · Authors · 2025-08-07
> > > **Response to Reviewer Z1Bz**
> > >
> > > Thank you for reviewing our work and providing your valuable feedback. Below, we address your concern regarding the comparison of computation times.
> > >
> > > We respectfully disagree with the expectation of including computation-time comparisons. The primary focus of this work is to establish regret lower bounds for the MASSP setting, rather than to propose an algorithm or compare the computational cost of implementing any algorithms on hard instances.  Specifically, we provide the first regret lower bounds in the MASSP framework and also show that these bounds are tight.  We neither propose any algorithm nor  do we make any empirical claims. Also, to date, the only algorithm available in this setting is MACCM (Trivedi et al., 2023), which demonstrates a sub-linear regret upper bound in the number of episodes. Since there is no other algorithm in this setting it is hard to provide any computation-time comparison at this point. We do agree that such a study would be valuable once additional algorithms become available. In fact, this represents a fundamentally different line of research, distinct from the theoretical focus of the present work.
> > >
> > > We would like to highlight that comparisons of computation time are absent even in the single-agent SSP literature on regret lower bounds, including works such as Min et al. (2022), despite the presence of multiple algorithms. This is because, in general, it is not known which of the exponentially many possible problem instances actually achieves the regret lower bound.
> > > In our multi-agent case, the hard instances also depend on the number of agents  $n$, with both the state and action spaces growing exponentially in  $n$. This significantly increases the complexity of any computation-time comparison and falls outside the intended scope of this work.
> > >
> > > In summary, the experiments related to computational complexity represent a valuable direction for future work. So, we do not consider their absence as a limitation of the present work, as our focus is on the theoretical aspects of regret lower bounds.
> > >
> > > We welcome any further questions and are happy to respond.

---

> > > > ### Author Response · Authors · 2025-08-09
> > > > **Request to Reviewer Z1Bz**
> > > >
> > > > Dear Reviewer Z1Bz,
> > > >
> > > > We would like to thank the reviewer again for the feedback and thoughtful questions. We hope that our responses have addressed all your concerns. Since the author-reviewer discussion period is closing soon, we wanted to check if all your doubts are now resolved or if you have any additional questions, suggestions, or comments that we can address at this stage. We would be happy to provide further clarifications.
> > > >
> > > > Given the significance of the contributions and the enhancements made in response to your feedback, we kindly ask you to consider increasing your score if you feel it is appropriate.
> > > >
> > > > Thanks,
> > > > Authors

---

### Note · Authors · 2025-08-14

We sincerely thank the Reviewers for their comments, questions and their interest in this work. We particularly thank the knowledgeable Reviewers who spotted the significance and technical hardness of this work and also who thoughtfully pointed out further possibilities. We thank them all for their encouraging views. We believe that we gave adequate responses to each of their questions and also in a timely manner.

While we carefully responded to every point raised, we would like to highlight a few of the most pressing and significant comments that will be reflected in the final version of the paper, along with some that open up promising avenues for future work.

**Reviewer Z1Bz:** We addressed the major concern regarding the comparison of computation time across various algorithms, extension of our results to randomized policies and the details on exploration issue. We also thank the reviewer for pointing out some of the typos which we will fix in the final version.


**Reviewer jnXP:** We discussed the issue of limited information sharing across agents, as well as the impact of model misspecification in the MARL setting, a largely unexplored area that we now identify as an important future research direction.

**Reviewer YLTg:** We elaborated on the practical relevance of the MASSP problem considered in this work. In addition, we clarified and addressed the comment related to the assumption on decentralized algorithm .

**Reviewer MLjP:** We will incorporate the extended discussion on the SSP and MASSP problems and the discussion regarding the existence of exponentially many near-optimal policies into the final version.

We sincerely thank all reviewers for their insightful feedback and encouragement, as well as the AC for the opportunity to provide this concluding remark. The comments have helped us strengthen the paper, and we are glad that our rebuttal addressed the key concerns raised. We will address and incorporate the necessary updates in the final version.

---

### Decision · Program_Chairs · 2025-09-17

**Decision:**

Accept (poster)

**Comment:**

This paper studies the theoretical foundations of decentralized multi-agent stochastic shortest path problems with linear function approximation. Its main contribution is a regret lower bound in this setting, established through carefully constructed hard-to-learn instances.

The strengths of the paper lie in its novelty and technical depth. Establishing a lower bound for decentralized multi-agent SSPs represents a meaningful theoretical advance for decentralized reinforcement learning. The feature design, monotonicity arguments, and treatment of exponential state-action spaces form the key technical contributions, and the work highlights the inherent difficulty of decentralized learning.

The weaknesses concern presentation, assumptions, and broader impact. Several assumptions (such as linear models and a decentralized setup without communication details) require clarification, and parts of the proofs are difficult to follow. The relevance of the constructed hard instances to real-world decentralized systems is not entirely convincing, limiting practical significance. One reviewer also noted the lack of empirical validation, such as runtime comparisons. While the rebuttal addressed some technical points (e.g., randomized policies and exploration assumptions), it did not fully resolve concerns about applicability.

Overall, the paper makes a theoretical contribution that helps characterize fundamental limits in decentralized multi-agent learning. However, it is narrowly scoped, heavily assumption-driven, and at times difficult to interpret without additional intuition. Its value lies primarily in advancing theoretical understanding rather than practical impact.